# Real-DRL: Teach and Learn at Runtime

**Yanbing Mao**∗
Engineering Technology Division
Wayne State University
Detroit, MI 48202
hm9062@wayne.edu

**Yihao Cai**∗
Department of Electrical and Computer Engineering
Wayne State University
Detroit, MI 48202
yihao.cai@wayne.edu

**Lui Sha**
Siebel School of Computing and Data Science
University of Illinois Urbana-Champaign
Urbana, IL 61801
lrs@illinois.edu

## Abstract

This paper introduces the Real-DRL framework for safety-critical autonomous systems, enabling **runtime learning** of a deep reinforcement learning (DRL) agent to develop safe and high-performance action policies in real plants (i.e., real physical systems to be controlled), while prioritizing safety! The Real-DRL consists of three interactive components: a DRL-Student, a PHY-Teacher, and a Trigger. The DRL-Student is a DRL agent that innovates in the dual self-learning and teaching-to-learn paradigm and the real-time safety-informed batch sampling. On the other hand, PHY-Teacher is a physics-model-based design of action policies that focuses solely on safety-critical functions. PHY-Teacher is novel in its real-time patch for two key missions: i) fostering the teaching-to-learn paradigm for DRL-Student and ii) backing up the safety of real plants. The Trigger manages the interaction between the DRL-Student and the PHY-Teacher. Powered by the three interactive components, the Real-DRL can effectively address safety challenges that arise from the unknown unknowns and the Sim2Real gap. Additionally, Real-DRL notably features i) assured safety, ii) automatic hierarchy learning (i.e., safety-first learning and then high-performance learning), and iii) safety-informed batch sampling to address the learning experience imbalance caused by corner cases. Experiments with a real quadruped robot, a quadruped robot in NVIDIA Isaac Gym, and a cart-pole system, along with comparisons and ablation studies, demonstrate the Real-DRL's effectiveness and unique features.

## 1   Introduction

Deep reinforcement learning (DRL) has been incorporated into many autonomous systems and has shown significant advancements in making sequential and complex decisions in various fields, such as autonomous driving [32, 33] and robot locomotion [28, 34]. However, the AI incident database[2] has revealed that machine learning (ML) techniques, including DRL, can achieve remarkable performance but without a safety guarantee [60]. Therefore, ensuring high-performance DRL with verifiable safety is more crucial than ever, aligning with the growing market demand for safe ML techniques.

---

∗Equal contribution.
[2]AIID: https://incidentdatabase.ai

## 1.1 Safety Challenges

Two significant challenges emerge when developing high-performance DRL with verifiable safety.

**Challenge 1: Unknown Unknowns.** The dynamics of learning-enabled autonomous systems are governed by conjunctive known known (e.g., Newtonian physics), known unknowns (e.g., Gaussian noise without identified mean and variance), and unknown unknowns. One significant example of an unknown unknown arises from DNN's colossal parameter space, intractable activation, and random factors. Moreover, many autonomous systems, such as quadruped robots [19] and autonomous vehicles [44], have system-environments interaction dynamics. Unpredictable and complex environments, such as freezing rain [51], can introduce a new type of unknown unknowns. Safety assurance also requires the resilience to unknown unknowns since they are hard-to-simulate and hard-to-predict, and often lack sufficient historical data for scientific modeling and analysis.

**Challenge 2: Sim2Real Gap.** The prevalent DRL strategies involve training a policy within a simulator and deploying it to a physical platform. However, the difference between the simulated environment and the real world creates the Sim2Real gap. This gap causes a significant drop in performance when using pre-trained DRL in a novel real environment. Numerous approaches have been developed to address the Sim2Real gap [40, 38, 52, 59, 14, 29, 20, 54, 56]. These methods aim to improve the realism of the simulator and can mitigate the Sim2Real gap to varying degrees. However, undisclosed gaps continue to hinder the safety assurance of real plants.

## 1.2 Related Work

Significant efforts have been devoted to addressing safety concerns related to DRL, detailed below.

**Safe DRL.** Since the reward guides DRL's learning of action policies, one research focus of safe DRL is the safety-embedded reward to develop a high-performance action policy with verifiable safety. The control Lyapunov function (CLF) proposed in [42, 5, 15, 62] is a candidate. Meanwhile, the seminal work in [55] revealed that a CLF-like reward could enable DRL with verifiable stability. Enabling verifiable safety is achievable by extending CLF-like rewards with given safety conditions. However, the systematic guidance for constructing such CLF-like rewards remains open. The residual action policy is another shift in safe DRL, which integrates data-driven action policy and physics-model-based action policy. The existing residual diagrams focus on stability guarantee [45, 35, 18, 30], with the exception being [17] on safety guarantee. However, the physics models considered are nonlinear and intractable, which thwarts delivering a verifiable safety guarantee, if not impossible. The recent Phy-DRL (physics-regulated DRL) framework [13, 12] can satisfactorily address the open problems. It exhibits verifiable safety, but it is only theoretically possible due to the underlying assumption of the manageable Sim2Real gap and unknown unknowns.

**Fault-Tolerant DRL**. It is another direction for enhancing safety assurance of DRL in real plants. Recent approaches include neural Simplex [43], runtime assurance [8, 50, 16], and runtime learning machine [10, 11] for deterministic safety definition, and model predictive shielding [4, 3] for probabilistic safety definition. They treat the DRL agent as a high-performance but black-box module that operates in parallel with a high-assurance module. The high-assurance module is a physics-model-based controller, assumed to have assured safety to tolerate the DRL faults. However, this assumption becomes problematic in situations where there are significant model mismatches, particularly due to unknown unknowns such as dynamic and unpredictable operating environments.

## 1.3 Contribution: Real-DRL Framework: From Theory to Implementation

To address Challenges 1 and 2, we propose the **Real-DRL** framework, which enables the runtime learning of a DRL agent to develop safe and high-performance action policies for safety-critical autonomous systems in real physical environments, while prioritizing safety. As illustrated in Figure 1, Real-DRL, building on Simplex principles [49], has three interactive components:

- **DRL-Student** is a DRL agent that can learn from scratch. It is innovative in the dual self-learning and teaching-to-learn paradigm and the safety-informed batch sampling to efficiently develop a safe and high-performance action policy in real plants.
- **PHY-Teacher** is a physics-model-based design with assured safety, focusing on safety-critical functions only. It aims at fostering the teaching-to-learn mechanism for DRL-Student and backing up the safety of real plants, in the face of Challenges 1 and 2.

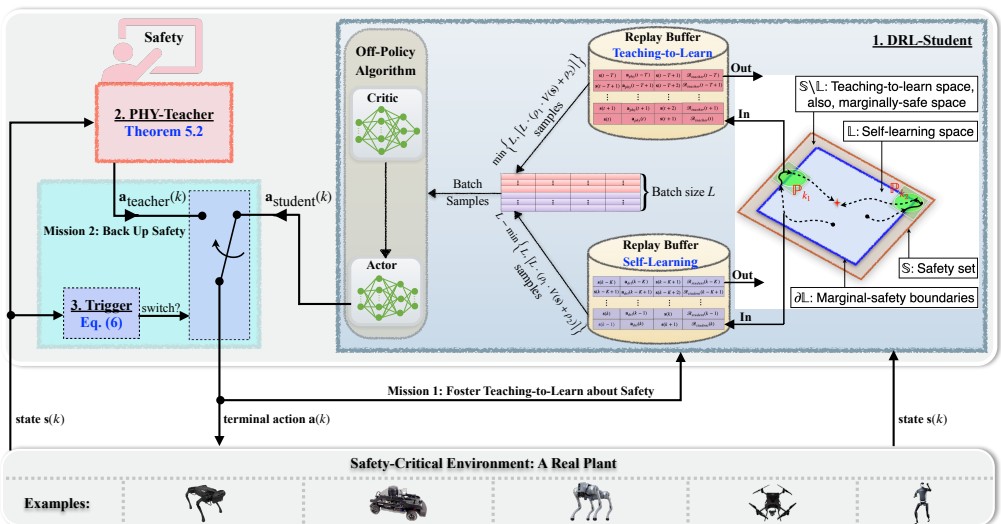

Figure 1: Real-DRL framework, which is also formally described in Algorithm 1 of Appendix B.

- **Trigger** is responsible for monitoring the real-time safety status of the real plant, facilitating interactions between DRL-Student and PHY-Teacher.

Powered by the three interactive components, the Real-DRL notably features i) assured safety in the face of Challenges 1 and 2, ii) automatic hierarchy learning (i.e., safety-first learning and then high-performance learning), and iii) safety-informed batch sampling to address the experience imbalance caused by corner cases.

We present the preliminaries in Section 2 and elaborate on the Real-DRL design in Sections 3 to 5. The experimental results and implementation details of Real-DRL are provided in Sections 6 and 7.

## 2 Preliminaries

### 2.1 Notation

In this paper, we use $\mathbf{P} \succ 0$ to represent matrix $\mathbf{P}$ is positive definite. $\mathbb{R}^n$ is the set of $n$-dimensional real vectors, while $\mathbb{N}$ is the set of natural numbers. The superscript '$\top$' indicates matrix transposition. We define $\mathbf{0}_m$ and $\mathbf{1}_m$ as the $m$-dimensional vectors with entries of all zeros and ones, respectively, and $\mathbf{I}_m$ as the $m$-dimensional identity matrix. The $\mathbf{c} > \mathbf{0}_n$ means all entries of $\mathbf{c} \in \mathbb{R}^n$ are strictly positive. Lastly, the $\mathbf{diag}\{\mathbf{c}\}$ is diagonal matrix whose diagonal entries are vector $\mathbf{c}$'s elements.

### 2.2 Definitions

The dynamics of safety-critical autonomous systems can be described by the following equation:

$$\mathbf{s}(t + 1) = \mathbf{f}(\mathbf{s}(t), \mathbf{a}(t)), \ \ t \in \mathbb{N} \tag{1}$$

where $\mathbf{s}(t) \in \mathbb{R}^n$ represents the real-time state of the system sampled by sensors at time $t$, while $\mathbf{a}(t) \in \mathbb{R}^m$ signifies the real-time action command at time $t$. The equilibrium point of the system is $\mathbf{s}^* = \mathbf{0}_n$. For a safety-critical system (1), there exist the sets that system states and actions must always adhere to:

$$\text{Safety set: } \mathbb{S} \triangleq \left\{ \mathbf{s} \in \mathbb{R}^n | -\mathbf{c} < \mathbf{C} \cdot \mathbf{s} < \mathbf{c}, \ \text{with } \mathbf{c} > \mathbf{0}_p \right\}, \tag{2}$$

$$\text{Admissible action set: } \mathbb{A} \triangleq \left\{ \mathbf{a} \in \mathbb{R}^m \mid -\mathbf{d} < \mathbf{D} \cdot \mathbf{a} < \mathbf{d}, \ \text{with } \mathbf{d} > \mathbf{0}_q \right\}, \tag{3}$$

where $\mathbf{C}$ and $\mathbf{c}$ are provided in advance to describe $p \in \mathbb{N}$ safety conditions, while $\mathbf{D}$ and $\mathbf{d}$ are used to describe $q \in \mathbb{N}$ conditions on admissible action set. The inequalities presented in Equation (2) are sufficiently generic to describe many safety conditions. Examples include yaw control for quadruped robots to prevent collisions [22], and lane keeping and slip regulation in autonomous vehicles to follow traffic regulation and avoid slipping and skidding [44, 37].

In our Real-DRL, the DRL-Student will engage in dual self-learning and teaching-to-learn paradigm to develop a safe and high-performance action policy. To better describe this aspect, we introduce:

$$\text{Self-learning space: } \mathbb{L} \triangleq \left\{ \mathbf{s} \in \mathbb{R}^n \mid -\eta \cdot \mathbf{c} < \mathbf{C} \cdot \mathbf{s} < \eta \cdot \mathbf{c}, \text{ with } 0 < \eta < 1 \right\}. \tag{4}$$

**Remark 2.1** (**Buffer Zone**). *When system states arrive at a safety boundary of safety set $\mathbb{S}$, the action policy of PHY-Teacher usually cannot immediately drive them back to the safety set $\mathbb{S}$ without delay. These delays arise from multiple factors, such as computation time, communication latency, sampling periods, and actuator response times. This consideration motivates the self-learning space $\mathbb{L}$ (4). Specifically, the condition $0 < \eta < 1$ in conjunction with Equation (2) and Equation (4) indicates that $\mathbb{L} \subset \mathbb{S}$, which creates a non-empty buffer zone, i.e., $\mathbb{S} \setminus \mathbb{L}$. The Trigger will reference boundary $\partial \mathbb{L}$ (not $\partial \mathbb{S}$) to trigger PHY-Teacher for assuring safety, providing adequate margins for response time and decision latency.*

Building on the foundation (4), we introduce the concept of the safe action policy of DRL-Student:

**Definition 2.2** (**Safe Action Policy of DRL-Student**). *Consider the self-learning space $\mathbb{L}$ (4), the admissible action set $\mathbb{A}$ (3), and the real plant (1). The action policy of the DRL-Student, denoted as $\pi_{student}(\cdot) : \mathbb{R}^n \to \mathbb{R}^m$, is considered safe if, under its control, the system states satisfy the condition: Given $\mathbf{s}(1) \in \mathbb{L}$, the $\mathbf{s}(t) \in \mathbb{L}$ and the $\mathbf{a}_{student}(t) = \pi_{student}(\mathbf{s}(t)) \in \mathbb{A}$ hold for all time $t \in \mathbb{N}$.*

In our Real-DRL framework, PHY-Teacher's action policy is designed to have assured safety only. We introduce the concept of "assured safety" below, which will guide the design of PHY-Teacher.

**Definition 2.3** (**Safe Action Policy of PHY-Teacher**). *Consider the safety set $\mathbb{S}$ (2), the action set $\mathbb{A}$ (3), the self-learning space $\mathbb{L}$ (4), the real plant (1), and denote:*

$$\text{Activation Period of PHY-Teacher: } \mathbb{T}_k \triangleq \{k+1, \ k+2, \ \ldots, \ k+\tau_k\}, \tag{5}$$

*where $\tau_k \in \mathbb{N}$. The action policy of PHY-Teacher, denoted as $\pi_{teacher}(\cdot) : \mathbb{R}^n \to \mathbb{R}^m$, is said to have assured safety, if $\mathbf{s}(k) \in \mathbb{S} \setminus \mathbb{L}$, then i) the $\mathbf{s}(t) \in \mathbb{S}$ holds for all time $t \in \mathbb{T}_k$, ii) the $\mathbf{a}_{teacher}(t) = \pi_{teacher}(\mathbf{s}(t)) \in \mathbb{A}$ holds for all time $t \in \mathbb{T}_k$, and iii) the $\mathbf{s}(k + \tau_k) \in \mathbb{L}$ holds.*

By Definition 2.3, the PHY-Teacher is deemed trustworthy for guiding the DRL-Student in safety learning and backing up the safety of real plant, only if its action policy simultaneously satisfies the corresponding conditions: i) it must ensure the real plant's real-time states consistently adhere to the safety conditions within $\mathbb{S}$; ii) it must keep its real-time actions within an admissible action set $\mathbb{A}$; and iii) it must ensure the real plant's states return to the self-learning space $\mathbb{L}$ as activation period ends.

With the definitions established, we next detail the design of Real-DRL's three interactive components in Sections 3 to 5, respectively.

## 3 Real-DRL: Trigger Component

Real-DRL's Trigger facilitates interactions between the DRL-Student and PHY-Teacher by monitoring the triggering condition outlined below.

$$\text{Triggering condition of PHY-Teacher: } \mathbf{s}(k-1) \in \mathbb{L} \text{ and } \mathbf{s}(k) \in \partial \mathbb{L}, \tag{6}$$

where $\partial \mathbb{L}$ denotes the boundaries of the self-learning space $\mathbb{L}$ (4). The condition (6), in conjunction with the $\mathbb{T}_k$ defined in (5), can describe the conditional activation periods of PHY-Teacher and DRL-Student as follows:

$$\begin{cases} \text{PHY-Teacher is active for time } t \in \mathbb{T}_k, & \text{if condition (6) holds at time } k, \\ \text{DRL-Student is active for time } t \notin \mathbb{T}_k, & \text{otherwise.} \end{cases} \tag{7}$$

Furthermore, the logic for managing actions applied to the real plant is as follows:

$$\mathbf{a}(t) \leftarrow \begin{cases} \mathbf{a}_{\text{teacher}}(t), & \text{if condition (6) holds at time } k \text{ and } t \in \mathbb{T}_k, \\ \mathbf{a}_{\text{student}}(t), & \text{otherwise,} \end{cases} \tag{8}$$

where $\mathbf{a}(t)$ represents the terminal real-time action applied to the real plant, as illustrated in Figure 1.

The interactions between the DRL-Student and PHY-Teacher, as facilitated by the Trigger, can be summarized from Equations (6) to (8) as follows: When the states of the plant controlled by the DRL-Student approach a marginal-safety boundary (i.e., they are leaving the safe self-learning space $\mathbb{L}$), the Trigger prompts the PHY-Teacher to instruct the DRL-Student on safety learning and back up the safety of real plant. Once system states return to the (safe) self-learning space $\mathbb{L}$, the DRL-Student regains the control of real plant. Next, we detail the design of DRL-Student and PHY-Teacher.

# 4 Real-DRL: DRL-Student Component

As shown in Figure 1, DRL-Student adopts an *actor-critic* architecture [36, 27] with dual replay buffers. DRL-Student operates within a dual self-learning and teaching-to-learn paradigm to develop safe and high-performance action policies in real plants. Specifically, it learns safety from the PHY-Teacher's experience of backing up safety acquired in the teaching-to-learn space, denoted as $\mathbb{S} \setminus \mathbb{L}$. At the same time, it learns to achieve high task-performance from its own (purely data-driven) experience in the self-learning space, represented as $\mathbb{L}$. Meanwhile, DRL-Student employs our proposed safety-informed batch sampling to effectively 1) address the experience imbalance (caused by corner cases) throughout learning, and 2) promote the dual self-learning and teaching-to-learn paradigm. Next, we present the novel design of DRL-Student.

## 4.1 Self-Learning Mechanism

The self-learning objective is to optimize an action policy $\mathbf{a}_{\text{student}}(k) = \pi_{\text{student}}(\mathbf{s}(k))$ which maximizes the expected return from the initial state distribution:

$$\mathcal{Q}^{\pi_{\text{student}}}(\mathbf{s}(k), \mathbf{a}_{\text{student}}(k)) = \mathbf{E}_{\mathbf{s}(k) \sim \mathbb{L}} \left[ \sum_{t=k}^{\infty} \gamma^{t-k} \cdot \mathcal{R}(\mathbf{s}(t), \mathbf{a}_{\text{student}}(t), \mathbf{s}(t+1)) \right], \qquad (9)$$

where $\mathbb{L}$ is the self-learning space defined in Equation (4), $\mathcal{R}(\cdot, \cdot, \cdot)$ is the reward function that maps the state-action-next-state triples to real-value rewards, and $\gamma \in [0, 1]$ is the discount factor. The expected return (9) and action policy $\pi_{\text{student}}(\cdot)$ are parameterized by the critic and actor networks, respectively. The reward function is designed to enhance task-oriented performance through runtime learning. One such example is minimizing travel time and power consumption in the robot navigation.

Referring to Figure 1, we can summarize that during the activation period of DRL-Student, its interactions with real plants generate the purely data-driven experience data as

$$\text{Self-learning experience:} \left[ \mathbf{s}(t) \mid \mathbf{a}_{\text{student}}(t) \mid \mathbf{s}(t+1) \mid \mathcal{R}(\mathbf{s}(t), \mathbf{a}_{\text{student}}(t), \mathbf{s}(t+1)) \right], \qquad (10)$$

which is stored in the self-learning replay buffer, and then batch sampled to update the critic and actor networks using the off-policy algorithm, such as deep deterministic policy gradient (DDPG) [36], twin delayed DDPG [23] or soft actor-critic [27]. We thus conclude that the self-learning mechanism is the learning from data-driven experience through the trial-and-error in self-learning space $\mathbb{L}$ (4). This mechanism is also formally described by Lines 25 to 29 in Algorithm 1 of Appendix B.

## 4.2 Teaching-to-Learn Mechanism

The designed PHY-Teacher has successful (physics-model-based) experience — not grounded in data-driven trial-and-error — in backing up the safety of real plants. As shown in Figure 1, this experience is acquired in the teaching-to-learn (or marginally-safe) space $\mathbb{S} \setminus \mathbb{L}$, where the DRL-Student is deactivated due to safety violations that occur as a result of its actions' failure in assuring safety. If PHY-Teacher shares its successful experiences in backing up the safety of real plants, then DRL-Student can learn about safety from these experiences. This process is known as the teaching-to-learn mechanism. Referring to Figure 1, the teaching-to-learn mechanism can be summarized as that during the activation period of PHY-Teacher, it gains its experience data in backing up safety as

$$\text{Teaching-to-learn experience:} \left[ \mathbf{s}(t) \mid \mathbf{a}_{\text{teacher}}(t) \mid \mathbf{s}(t+1) \mid \mathcal{R}(\mathbf{s}(t), \mathbf{a}_{\text{teacher}}(t), \mathbf{s}(t+1)) \right], \quad (11)$$

where $\mathbf{a}_{\text{teacher}}(t)$ is computed according to PHY-Teacher's action policy (18) and its design is detailed in Theorem 5.2. The experience data (11) is stored in the teaching-to-learn replay buffer, and is then batch sampled to update the critic and actor networks using the off-policy algorithm. This teaching-to-learn mechanism is also formally described by Lines 12 to 16 in Algorithm 1 of Appendix B.

## 4.3 Safety-Informed Batch Sampling

This section proposes a batch sampling approach to facilitate the dual self-learning and teaching-to-learn paradigm. The benefits of the proposed approach go beyond achieving this paradigm. Before proceeding, we recall a real problem that the incidents of learning-enabled autonomous systems, such as self-driving cars, often occur in corner cases [63, 7, 6]. As illustrated in Figure 1, the marginally-safe (or teaching-to-learn) space can represent the space of corner cases, where a slight disturbance

or fault can take a system out of control. Such marginal samples are safety-critical and are not often and hard visited normally, creating an experience imbalance for learning. Intuitively, focusing the learning on such corner-case experience will enable a more robust action policy. However, the current DRL frameworks usually adopt random sampling [45, 35, 18, 30, 17, 47], which cannot guarantee sufficient sampling of such samples, and thus cannot overcome the experience imbalance problem. To address the problem, we propose the safety-informed batch sampling for DRL-Student's learning, which also facilitates the dual self-learning and teaching-to-learn paradigm. To explain the sampling mechanism, we first introduce:

$$\text{Safety-status indicator: } V(\mathbf{s}) \triangleq \mathbf{s}^\top \cdot \mathbf{P} \cdot \mathbf{s}, \tag{12}$$

where $\mathbf{P}$ is computed according to the following:

$$\mathbf{P} = \arg\min_{\widetilde{\mathbf{P}} \succ 0}\{\log\det(\widetilde{\mathbf{P}})\}, \text{ subject to } \mathbf{I}_p - (\mathbf{C} \cdot \mathbf{diag}^{-1}\{\mathbf{c}\}) \cdot \widetilde{\mathbf{P}}^{-1} \cdot (\mathbf{C} \cdot \mathbf{diag}^{-1}\{\mathbf{c}\})^\top \succ 0, \tag{13}$$

with $\mathbf{C}$ and $\mathbf{c}$ given in Equation (2) for defining the safety set $\mathbb{S}$. We can derive the following property related to safety-status indicator, whose proof is given in Appendix C.

**Theorem 4.1.** *Consider the safety set $\mathbb{S}$ (2) and the function $V(\mathbf{s})$ (12) with its matrix $\mathbf{P}$ computed in Equation (13). The ellipsoid set $\{\mathbf{s} \in \mathbb{R}^n \mid V(\mathbf{s}) < 1\} \subseteq \mathbb{S}$ and it has the maximum volume.*

The proposed mechanism of safety-informed batch sampling is based on Theorem 4.1 and the safety-status indicator in Equation (12). As shown in Figure 1, the sampling mechanism is to generate the batch samples to update critic and actor networks, which, thus, can be summarized as:

$$\underbrace{L}_{\text{Batch size}} = \underbrace{\min\{L, \lceil L \cdot (\rho_1 \cdot V(\mathbf{s}(t)) + \rho_2)\rceil\}}_{\text{Number of samples from teaching-to-learn reply buffer}} + \underbrace{L - \min\{L, \lceil L \cdot (\rho_1 \cdot V(\mathbf{s}(t)) + \rho_2)\rceil\}}_{\text{Number of samples from self-learning reply buffer}}, \tag{14}$$

where $\rho_1 > 0$ and $\rho_2 \geq 0$ are given hyperparameters.

An illustration example of the proposed batch sampling is provided in Figure 2. Along with Theorem 4.1 and Equation (14), it reveals two key insights of DRL-Student's learning:

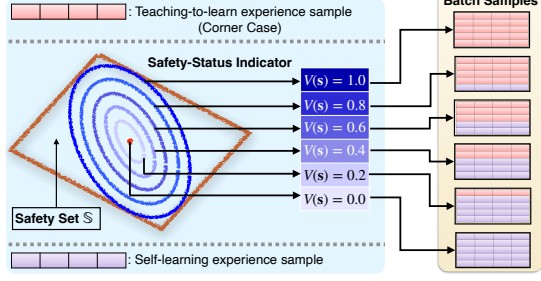

- The batch comprises experience samples from the teaching-to-learn and self-learning replay buffers, with their contribution ratios depending on real-time safety status due to $V(\mathbf{s}(t))$. Consequently, the DRL-Student participates in the dual self-learning and teaching-to-learn paradigm, discussed in Sections 4.1 and 4.2.

Figure 2: An illustration example of safety-informed batch sampling, given batch size $L = 5$, $\rho_1 = 1$, and $\rho_2 = 0$.

- The contribution ratio of teaching-to-learn (also, corner-case) experience samples is increasing with respect to the real-time safety status $V(\mathbf{s}(t))$, thereby addressing the experience imbalance. For example, by Theorem 4.1 and illustrated in Figure 2, an extreme value of $V(\mathbf{s}(t)) = 1$ suggests that the real-time states are approaching safety boundaries. Consequently, according to Equation (14), all batch samples are from the marginally-safe space. This enables the DRL-Student to concentrate on learning from corner-case experience samples (stored in teaching-to-learn replay buffer) about safety, when the real plant is only marginally safe.

## 5  Real-DRL: PHY-Teacher Component

PHY-Teacher is a physics-model-based design to have assured safety by Definition 2.3. One aim of PHY-Teacher is to tolerate real-time unknown unknowns (which result Sim2Real gap) to assure safety. We note many autonomous systems, such as quadruped robots [19] and autonomous vehicles [44], have system-environments interaction dynamics. So, the real-time unknown unknowns (e.g., unpredictable weather conditions and complex wild terrains) can have a significant influence on system dynamics. To tolerate such unknowns, we develop the real-time patch as for PHY-Teacher. Its design is detailed below.

The proposed real-time patch first defines the state space in which PHY-Teacher's action policy has assured safety according to Definition 2.3.

$$\text{Real-time patch: } \mathbb{P}_k \triangleq \left\{ \mathbf{s} \in \mathbb{R}^n \mid -\theta \cdot \mathbf{c} < \mathbf{C} \cdot (\mathbf{s} - \mathbf{s}_k^*) < \theta \cdot \mathbf{c}, \text{ with } 0 < \theta < 1 \right\}, \quad (15)$$

where $\mathbf{s}_k^*$ is the patch center, and also PHY-Teacher's real-time control goal, which is defined below.

$$\mathbf{s}_k^* \triangleq \chi \cdot \mathbf{s}(k), \text{ where } \mathbf{s}(k) \text{ satisfies the triggering condition (6) and } 0 < \chi < 1. \quad (16)$$

With the real-time control goal at hand, we introduce the following dynamics model of tracking errors, which is transformed from the dynamics model described in Equation (1).

$$\mathbf{e}(t+1) = \mathbf{A}(\mathbf{s}(k)) \cdot \mathbf{e}(t) + \mathbf{B}(\mathbf{s}(k)) \cdot \mathbf{a}_{\text{teacher}}(t) + \mathbf{h}(\mathbf{e}(t)), \text{ for } t \in \mathbb{T}_k, \quad (17)$$

where $\mathbb{T}_k$ (as defined in Equation (5)) indicates the PHY-Teacher's activation period, $\mathbf{e}(t) \triangleq \mathbf{s}(t) - \mathbf{s}_k^*$, and $\mathbf{h}(\mathbf{e}(t))$ represents an unknown model mismatch, arising from unknown model uncertainties, parameter uncertainties, unmodeled dynamics, and external disturbances that are not captured by the nominal model knowledge $\{\mathbf{A}(\mathbf{s}(k)), \mathbf{B}(\mathbf{s}(k))\}$. The $\{\mathbf{A}(\mathbf{s}(k)), \mathbf{B}(\mathbf{s}(k))\}$ will be utilized for designing PHY-Teacher's action policy:

$$\mathbf{a}_{\text{teacher}}(t) = \mathbf{F}_k \cdot \mathbf{e}(t) = \mathbf{F}_k \cdot (\mathbf{s}(t) - \mathbf{s}_k^*), \text{ for } t \in \mathbb{T}_k. \quad (18)$$

We can summarize the working mechanism of PHY-Teacher from Equations (15) to (18). When the PHY-Teacher is triggered by the condition (6) at time $k$, it uses the most recent sensor data $\mathbf{s}(k)$ to update the physics-model knowledge $\{\mathbf{A}(\mathbf{s}(k)), \mathbf{B}(\mathbf{s}(k))\}$. This update is then utilized to compute the real-time patch and the associated action policy. Meanwhile, the patch center (16) lies within DRL-Student's self-learning space $\mathbb{L}$ (since of $0 < \chi < 1$), and also represents the real-time control goal. This arrangement defines the teaching task for the PHY-Teacher: to instruct the DRL-Student in learning a safe action policy that can return the system to space $\mathbb{L}$. Before presenting the overall design, we outline a practical assumption regarding the model mismatch.

**Assumption 5.1.** *The model mismatch $\mathbf{h}(\mathbf{e})$ in Equation (17) is bounded in the patch $\mathbb{P}_k$ (15). In other words, there exists $\kappa > 0$ such that $\mathbf{h}^\top(\mathbf{e}) \cdot \mathbf{P_k} \cdot \mathbf{h}(\mathbf{e}) \leq \kappa$ holds for $\mathbf{P_k} \succ 0$ and any $\mathbf{e} \in \mathbb{P}_k$.*

We present the design of PHY-Teacher in the theorem below; its proof can be found in Appendix D.

**Theorem 5.2** (Real-Time Patch). *Recall the sets defined in Equations (2) to (4), and consider the PHY-Teacher's action policy in Equation (18), whose $\mathbf{F}_k$ is computed according to*

$$\mathbf{F}_k = \mathbf{R}_k \cdot \mathbf{Q}_k^{-1} \qquad and \qquad \mathbf{P}_k = \mathbf{Q}_k^{-1}, \quad (19)$$

*with $\mathbf{R}_k$ and $\mathbf{Q}_k$ satisfying the following linear matrix inequalities (LMIs):*

$$\mathbf{I}_p - \overline{\mathbf{C}} \cdot \mathbf{Q}_k \cdot \overline{\mathbf{C}}^\top \succ 0, \text{ and } \mathbf{I}_q - \overline{\mathbf{D}} \cdot \mathbf{T}_k \cdot \overline{\mathbf{D}}^\top \succ 0, \text{ and } \mathbf{Q}_k - n \cdot \boldsymbol{diag}^2(\mathbf{s}(k)) \succ 0, \text{ and } \quad (20)$$

$$\begin{bmatrix} \alpha \cdot \mathbf{Q}_k & \mathbf{Q}_k \cdot \mathbf{A}^\top(\mathbf{s}(k)) + \mathbf{R}_k^\top \cdot \mathbf{B}^\top(\mathbf{s}(k)) \\ \mathbf{A}(\mathbf{s}(k)) \cdot \mathbf{Q}_k + \mathbf{B}(\mathbf{s}(k)) \cdot \mathbf{R}_k & \frac{\mathbf{Q}_k}{1+\phi} \end{bmatrix} \succ 0, \text{ and } \begin{bmatrix} \mathbf{Q}_k & \mathbf{R}_k^\top \\ \mathbf{R}_k & \mathbf{T}_k \end{bmatrix} \succ 0, (21)$$

*where $\overline{\mathbf{C}} \triangleq \mathbf{C} \cdot \boldsymbol{diag}^{-1}\{\theta \cdot \mathbf{c}\}$, $\overline{\mathbf{D}} = \mathbf{D} \cdot \boldsymbol{diag}^{-1}\{\mathbf{d}\}$, $0 < \alpha < 1$, and $\phi > 0$. Meanwhile, the given parameters $\eta$ in Equation (4), $\theta$ in Equation (15), $\chi$ in Equation (16), $\alpha$ and $\phi$ in Equation (21), and $\kappa$ in Assumption 5.1 satisfy*

$$\eta < \theta + \chi \cdot \eta < 1 \text{ and } \frac{(\phi+1) \cdot \kappa}{(1-\alpha) \cdot \phi} < \frac{(1-\chi)^2 \cdot \eta^2}{\theta^2}. \quad (22)$$

*Then, according to Definition 2.3, the PHY-Teacher's action policy (18) has assured safety in the real-time patch $\mathbb{P}_k$.*

**Remark 5.3** (**Policy Computation**). *By using the CVX [26] or LMI [24] toolbox, the matrices $\mathbf{Q}_k$ and $\mathbf{R}_k$ can be computed from Equations (20) and (21) for the matrix $\mathbf{F}_k$. The pseudocode for computing $\mathbf{F}_k$ using CVX and LMI, with a comment, is presented in Appendix E.*

**Remark 5.4** (**Adaptive Mechanism for Assuring Validity of Assumption 5.1**). *PHY-Teacher is designed to have verifiable expertise in safety protocols, as formally stated in Theorem 5.2, but this relies on Assumption 5.1. In real-world applications, particularly in hard-to-predict and hard-to-simulate environments (e.g., natural disasters and snow squalls), we cannot always guarantee that Assumption 5.1 will hold. To address this concern, we introduce an adaptive mechanism, presented in Appendix F, that utilizes real-time state to estimate real-time model mismatch, thereby enabling monitoring of Assumption 5.1's validity. When the assumption is violated, the mechanism computes compensatory actions in real-time to mitigate the effects of model mismatch and restore the validity of Assumption 5.1 timely.*

# 6 Implementation Best Practices: Minimizing Runtime Overhead

The PHY-Teacher and DRL-Student are designed to operate in parallel. The DRL-Student utilizes GPU acceleration and can be deployed on embedded hardware such as the NVIDIA Jetson series, while the PHY-Teacher efficiently handles real-time safety-critical tasks on the CPU. To minimize runtime overhead, the LMI solver — responsible for automatic and rapid computation of the PHY-Teacher's real-time action policies — should be implemented in C to significantly improve both memory efficiency and computational speed. To facilitate this implementation, we provide ECVXCONE (embedded CVX for cone programming), an online solver for embedded conic optimization capable of solving LMIs in real-time using C. The code is available at GitHub: https://github.com/Charlescai123/ecvxcone.

Table 2 in Appendix G presents an experimental evaluation of ECVXCONE's computational resource usage across four platforms with varying CPU frequencies, which demonstrates that ECVXCONE makes the Real-DRL well-suited for edge-AI devices with limited computational resources.

# 7 Experiment

The section presents experiments on a real quadruped robot, a quadruped robot in the NVIDIA Isaac Gym, and a cart-pole system, along with comprehensive comparisons and ablation studies. The complete code and details can be found on GitHub: https://github.com/Charlescai123/Real-DRL.

## 7.1 Real Quadruped Robot: Indoor Environment

The detailed design of this experiment is presented in Appendix I. During pre-training in the simulator, we set $r_{v_x} = 0.6$ m/s and $r_h = 0.24$ m. To better demonstrate Real-DRL, the real robot's velocity command is $r_{v_x} = 0.35$ m/s, which is very different from the one in the simulator. For the runtime learning in the real robot, one episode is defined as "*running robot for 15 seconds*."

We compare our Real-DRL with the state-of-the-art safe DRL, called Phy-DRL, which claims theoretically and experimentally fast training towards safety guarantee [13, 12]. Meanwhile, Phy-DRL integrates the concurrent delay randomization [29] and domain randomization [46] (randomizing friction force) for addressing the Sim2Real gap into its training. Here, we have three models for comparison: 1) '**Continual Phy-DRL**' representing a well-trained Phy-DRL in the simulator, which performed continual learning in the real robot to fine-tune the action policy; 2) '**Phy-DRL**' representing the well-trained Phy-DRL policy directly deployed to the real robot, and 3) our **Real-DRL**. The robot's CoM height and CoM-x velocity are shown in Figure 3, and a comparison video is available at

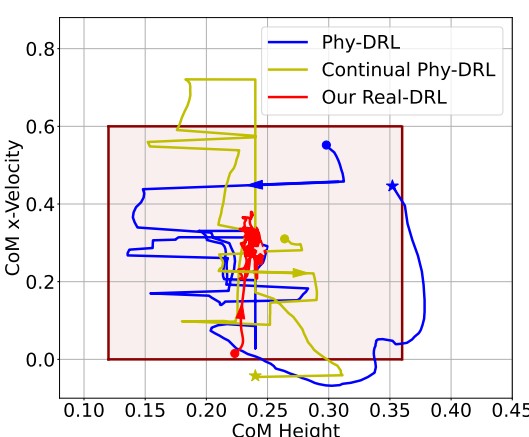

Figure 3: Phase plots.

https://www.youtube.com/watch?v=tQSPcnKEnYU. From the comparison video and Figure 3, we concluded that the Phy-DRL well-trained in simulator cannot guarantee the safety of the real robot due to the Sim2Real gap and unknown unknowns that the delay randomization and force randomization failed to capture. In contrast, our Real-DRL can always guarantee the safety of the real robot. Additionally, we recall that PHY-Teacher fosters DRL-Student's teaching-to-learn about safety. To verify this, we deactivate the PHY-Teacher in episodes 1 and 20, and compare system behavior. The demonstration video is available at https://www.youtube.com/shorts/UIFwlMrDgjA, which illustrates that DRL-Student quickly learned from PHY-Teacher to be safe, within 20 episodes, i.e., 300 seconds.

Finally, we showcase the Real-DRL's ability to tolerate various unknown unknowns. In addition to inherent ones in noisy samplings and complex DRL agents, we add five unknown unknowns that have never occurred in DRL-Student's historical learning. They are 1) **Beta**: Disturbances injected into DRL-Student's actions, generated by a randomized Beta distribution (see Appendix H for an explanation of its representation of unknown unknowns); 2) **PD**: Random and sudden payload

(around 4 lbs) drops on the robot's back; 3) **Kick**: Random kick by a human; 4) **DoS**: A real denial-of-service fault of the platform, which can be caused by task scheduling latency, communication delay, communication block, etc., but is unknown to us; and 5) **SP**: A sudden side push. We consider three combinations of them: i) '**Beta + PD**,' ii) '**Beta + DoS + Kick**,' and iii) '**Beta + SP**.' The demonstration video is available at https://www.youtube.com/shorts/Tl2jEUJMe_Y, which demonstrates that our Real-DRL successfully assures the safety of the real robot by tolerating such complex combined unknowns.

## 7.2 Quadruped Robot in NVIDIA Isaac Gym: Wild Environment (see Figure 8)

The operating environment plays a crucial role in introducing real-time unknown unknowns and Sim2Real gap. So, this benchmark focuses on comprehensive comparisons in complex environments. We utilized NVIDIA Isaac Gym to create a wild environment, assuming a real one. This environment transitions from flat terrain to unstructured and uneven ground, further complicated by unexpected ice that results in low friction. We here can conclude that the robot's operating environment are non-stationary and unforeseen. Besides, our Real-DRL allows for learning from scratch, which is performed here. Meanwhile, we integrate all comparison models with a navigation module, which enable the robot to autonomously traverse the uneven and unstructured terrains, and dynamic obstacles in the wild. The architecture of Real-DRL with integration of navigation is shown in Figure 9. The experiment design appears in Appendix J.

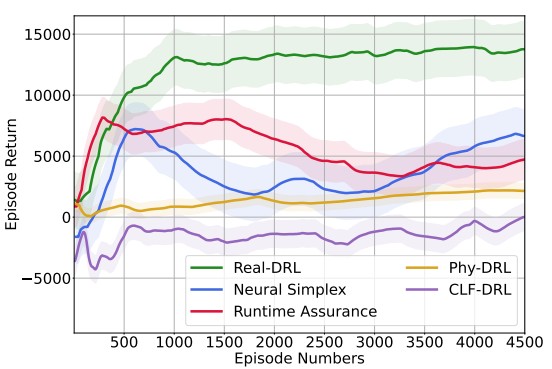

Figure 4: Trajectories of episode return.

Table 1: Safety metrics are highlighted in red. If the robot falls, other task-related metrics are labeled as N/A. If collision is not avoided, travel time and energy consumption are represented as $\infty$.

| Model ID | Navigation Performance | | | | | Energy Efficiency | |
| --- | --- | --- | --- | --- | --- | --- | --- |
| | **Success** | **Is Fall** | **Collision** | **Num (wp)** | **Travel Time (s)** | **Avg Power (W)** | **Total Energy (J)** |
| CLF-DRL | No | Yes | No | 0 | N/A | N/A | N/A |
| Phy-DRL | No | No | Yes | 1 | $\infty$ | 507.9441 | $\infty$ |
| Runtime Assurance | No | Yes | No | 2 | N/A | N/A | N/A |
| Neural Simplex | No | No | Yes | 2 | $\infty$ | 487.9316 | $\infty$ |
| PHY-Teacher | **Yes** | No | No | **4** | 55.5327 | 482.8468 | 26817.68 |
| Our Real-DRL | **Yes** | No | No | **4** | **45.3383** | **479.4638** | **21742.42** |

The experiment compares our Real-DRL with state-of-the-art frameworks: i) safe DRLs: CLF-DRL [55] and Phy-DRL [13, 12]; ii) fault-tolerant DRLs: neural Simplex [43] and runtime assurance [8, 50, 16]; iii) our sole PHY-Teacher. The testing models are chosen after 4500 episodes. Their performance comparisons are summarized in Table 1. Figure 4 presents the episode returns, and the corresponding comparison video is available at https://www.youtube.com/shorts/QqYSHjhq8vg. Additionally, a comparison video of **Real-DRL w/o v.s. w/ PHY-Teacher** during early stage of runtime learning is available at https://www.youtube.com/shorts/xcsZwrwfEOM. These comprehensive comparisons showcase our Real-DRL's notable features: its high-performance learning and assured safety.

## 7.3 Cart-Pole System: Ablation Studies

Our Real-DRL has three learning innovations: i) a teaching-to-learn mechanism focused on safety, ii) safety-informed batch sampling to overcome the experience imbalance, and iii) an automatic hierarchy learning, i.e., safety-first learning and then high-performance learning. In this benchmark, we primarily conduct ablation studies to demonstrate the impact of the three novel designs. The system's state consists of pendulum angle $\theta$, cart position $x$, velocities $\omega = \dot{\theta}$, and $v = \dot{x}$. The detailed experiment design can be found in Appendix K.

■ **Safety-Informed Batch Sampling:** To demonstrate the effect of this batch sampling scheme on addressing experience imbalance, we set up an imbalanced learning environment, as shown in Figure 10. We evaluate two DRL-Students in this environment: one learned in Real-DRL with our proposed batch sampling scheme (denoted as **'w/ SBS'**), and the other one learned in Real-DRL using a single replay buffer without this scheme (denoted as **'w/o SBS'**). The phase plots in Figure 5 and the rewards in Figure 11 illustrate that, after reaching convergence, the DRL-Student learned in Real-DRL with our batch sampling scheme has significantly enhanced safety assurance in corner-case environment, compared to the one learned from Real-DRL without this scheme.

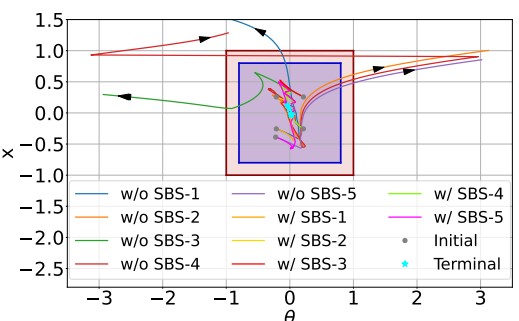

Figure 5: Phase plots, given five random initial states within self-learning space.

■ **Teaching-to-Learn Mechanism:** To evaluate the effect of this learning mechanism on policy learning, we compare two models: **Real-DRL w/ and w/o the teaching-to-learn**. The episode-average rewards (defined in Equation (84)) across five random initial states are shown in Figure 6 (a), demonstrating that the teaching-to-learn mechanism significantly improves sample efficiency, enabling faster convergence, more stable learning, and higher reward performance.

■ **Automatic Hierarchy Learning:** Finally, we showcase the automatic hierarchy learning. To do so, we disable PHY-Teacher in episodes 5 and 50 and observe system behavior under the control of sole DRL-Student. Intuitively, DRL-Student shall automatically become independent of PHY-Teacher and self-learn for a high-performance action policy. This can be seen from PHY-Teacher's activation ratio (defined in Equation (85)) in Figure 6 (b), where after episode 50, PHY-Teacher is rarely activated. Meanwhile, Figure 12 of Appendix K illustrates the high mission performance of action policy in episode 50: much more clustering and closer proximity to the mission goal.

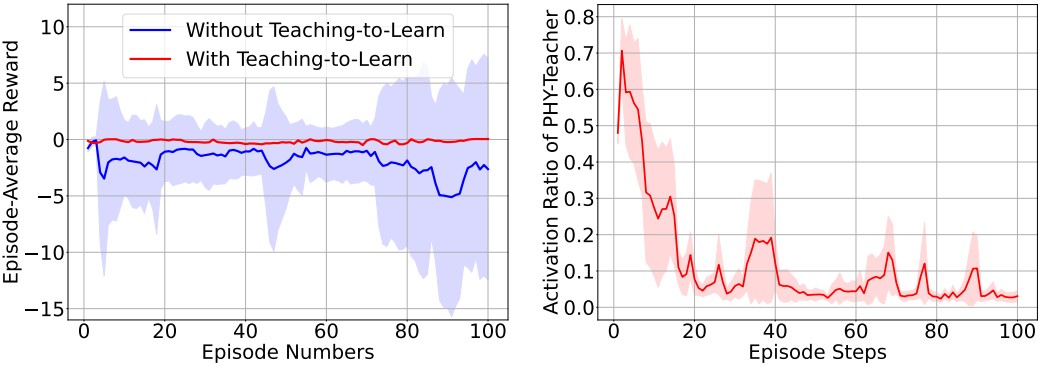

(a) Trajectories of episode-average reward (95% CI).  (b) PHY-Teacher's activation ratio over 100 episodes.

Figure 6: Ablation studies. (a): Teaching-to-learn mechanism, (b): Automatic hierarchy learning.

## 8 Conclusion and Discussion

This paper introduces Real-DRL, a framework specifically designed for safety-critical autonomous systems. The Real-DRL consists of three main components: a DRL-Student, a PHY-Teacher, and a Trigger. The DRL-Student engages in a dual self-learning and teaching-to-learn paradigm. The PHY-Teacher serves two primary functions: fostering the teaching-to-learn mechanism for DRL-Student and backing up safety. Real-DRL notably features i) assured safety in the face of unknown unknowns and Sim2Real gap, ii) teaching-to-learn for safety and self-learning for high performance, and iii) safety-informed batch sampling to address the experience imbalance. Experiments conducted on three benchmarks have successfully shown the effectiveness and unique features of Real-DRL.

In current Real-DRL, the Trigger continuously monitors the system safety status to manage DRL-Student and PHY-Teacher. To alleviate this monitoring burden, Trigger shall have an early warning function of future safety status. Utilizing reachability techniques [2] could provide a viable solution.

## Acknowledgments

This research was partially funded by the National Science Foundation through Grants CPS-2311084 and CPS-2311085, as well as by the NVIDIA Academic Grant Program.

We thank Hongpeng Cao at the Technical University of Munich, Germany, for his assistance with the indoor experiment on the real quadruped robot.

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

# Appendices

# A Auxiliary Lemmas

In this section, we present auxiliary lemmas that will be used in the proofs of the main results of this article.

**Lemma A.1** (Schur Complement [61]). *For any symmetric matrix* $\mathbf{M} = \begin{bmatrix} \mathbf{A} & \mathbf{B} \\ \mathbf{B}^\top & \mathbf{C} \end{bmatrix}$, *the* $\mathbf{M} \succ 0$ *holds if and only if the* $\mathbf{C} \succ 0$ *and* $\mathbf{A} - \mathbf{B} \cdot \mathbf{C}^{-1} \cdot \mathbf{B}^\top \succ 0$ *hold.*

**Lemma A.2.** *[48] We define two state sets:* $\mathbb{S} \triangleq \{ \mathbf{s} \in \mathbb{R}^n | \overline{\mathbf{C}} \cdot \mathbf{s} < \mathbf{1}_m \}$ *and* $\Omega \triangleq \{ \mathbf{s} \in \mathbb{R}^n \mid \mathbf{s}^\top \cdot \mathbf{Q}^{-1} \cdot \mathbf{s} < 1, \ \mathbf{Q} \succ 0 \}$. *We have the* $\Omega \subseteq \mathbb{S}$, *if the* $\mathbf{I}_m - \overline{\mathbf{C}} \cdot \mathbf{Q} \cdot \overline{\mathbf{C}}^\top \succ 0$ *holds.*

**Lemma A.3.** *[13] We define two action sets:* $\mathbb{A} \triangleq \{ \mathbf{a} \in \mathbb{R}^m | \overline{\mathbf{D}} \cdot \mathbf{a} < \mathbf{1}_p \}$ *and* $\Xi \triangleq \{ \mathbf{a} \in \mathbb{R}^m | \mathbf{a}^\top \cdot \mathbf{T}^{-1} \cdot \mathbf{a} < 1, \ \mathbf{T} \succ 0 \}$. *We have the* $\Xi \subseteq \mathbb{A}$, *if the* $\mathbf{I}_p - \overline{\mathbf{D}} \cdot \mathbf{T} \cdot \overline{\mathbf{D}}^\top \succ 0$ *holds.*

**Lemma A.4.** *For two vectors* $\mathbf{x} \in \mathbb{R}^n$, $\mathbf{y} \in \mathbb{R}^n$, *and a matrix* $\mathbf{P} \succ 0$, *we have*

$$2 \cdot \mathbf{x}^\top \cdot \mathbf{P} \cdot \mathbf{y} \leq \phi \cdot \mathbf{x}^\top \cdot \mathbf{P} \cdot \mathbf{x} + \frac{1}{\phi} \cdot \mathbf{y}^\top \cdot \mathbf{P} \cdot \mathbf{y}, \ \ with \ \ \phi > 0.$$

*Proof.* The proof straightforwardly follows inequality of:

$$(\sqrt{\phi} \cdot \mathbf{x} - \frac{1}{\sqrt{\phi}} \cdot \mathbf{y})^\top \cdot \mathbf{P} \cdot (\sqrt{\phi} \cdot \mathbf{x} - \frac{1}{\sqrt{\phi}} \cdot \mathbf{y}) = \phi \cdot \mathbf{x}^\top \cdot \mathbf{P} \cdot \mathbf{x} + \frac{1}{\phi} \cdot \mathbf{y}^\top \cdot \mathbf{P} \cdot \mathbf{y} - 2 \cdot \mathbf{x}^\top \cdot \mathbf{P} \cdot \mathbf{y} \geq 0,$$

which is due to $\mathbf{P} \succ 0$ and $\phi > 0$. $\qquad\square$

**Lemma A.5.** *Given any* $\lambda > 0$ *and matrix* $\mathbf{X} \in \mathbf{R}^{m \times n}$, *the matrix* $\mathbf{X} \cdot \mathbf{X}^\top + \lambda \cdot \mathbf{I}_m$ *is invertible.*

*Proof.* The proof is straightforward by letting $\mathbf{y} = \mathbf{X}^\top \cdot \mathbf{x}$. For any non-zero vector $\mathbf{x}$, we have $\mathbf{x}^\top \cdot (\mathbf{X} \cdot \mathbf{X}^\top + \lambda \cdot \mathbf{I}_m) \cdot \mathbf{x} = \|\mathbf{y}\|_2^2 + \lambda \cdot \|\mathbf{x}\|_2^2 > 0$, where $\|\cdot\|_2$ denotes the $L_2$-norm of a vector. So, $\mathbf{X} \cdot \mathbf{X}^\top + \lambda \cdot \mathbf{I}_m$ is a positive definite matrix, thus always invertible. $\qquad\square$

# B   Pseudocode: Real-DRL Framework

**Algorithm 1** Real-DRL Algorithm

---

1: Real plant operates from the safe self-learning space $\mathbb{L}$ initially, i.e., $\mathbf{s}(t) \in \mathbb{L}$ at initial time $t = 1$;
2: DRL-Student's real-time actor-network outputs the action $\mathbf{a}_{\text{student}}(t)$ given the input $\mathbf{s}(t)$;
3: DRL-Student applies $\mathbf{a}_{\text{student}}(t)$ to real plan to have the state $\mathbf{s}(t+1)$;
4: Update time index: $k \leftarrow t + 1$;
5: **if** $\mathbf{s}(t+1) \notin \mathbb{L}$ **then**
6:     PHY-Teacher uses the real-time state $\mathbf{s}(k)$ to update physics-model knowledge $(\mathbf{A}(\mathbf{s}(k)), \mathbf{B}(\mathbf{s}(k))$;
7:     PHY-Teacher uses the real-time state $\mathbf{s}(k)$ to update control goal $\mathbf{s}_k^*$ in Equation (16);
8:     PHY-Teacher computes $\mathbf{R}_k$ and $\mathbf{Q}_k$ from inequalities in Equations (20) and (21);
9:     PHY-Teacher computes action policy, denoted by $\mathbf{F}_k$, according to Equation (19);
10:     PHY-Teacher computes action $\mathbf{a}_{\text{teacher}}(t+1)$ according to Equation (18), given real-time state $\mathbf{s}(t+1)$;
11:     PHY-Teacher applies $\mathbf{a}_{\text{teacher}}(t+1)$ to real plan to back up safety and have $\mathbf{s}(t+2)$;
12:     PHY-Teacher computes $\mathcal{R}(\mathbf{s}(t+1), \mathbf{a}_{\text{teacher}}(t+1), \mathbf{s}(t+2))$;
13:     PHY-Teacher compiles the teaching-to-learn experience data as described in Equation (11);
14:     PHY-Teacher stores the teaching-to-learn experience data (11) into teaching-to-learning replay buffer;
15:     Generate batch samples according to the safety-informed batch sampling mechanism (14);
16:     Apply off-policy algorithms to update critic and actor networks, using the generated batch samples;
17:     **if** $\mathbf{s}(t+2) \notin \mathbb{L}$ **then**
18:         Update time index: $t + 1 \leftarrow t + 2$;
19:         Go to Line 10;
20:     **else**
21:         Update time index: $t \leftarrow t + 2$;
22:         Go to Line 2;
23:     **end if**
24: **else**
25:     DRL-Student computes the reward $\mathcal{R}(\mathbf{s}(t), \mathbf{a}_{\text{student}}(t), \mathbf{s}(t+1))$;
26:     DRL-Student compiles the self-learning experience data as described in Equation (10);
27:     DRL-Student stores the self-learning experience data (10) into self-learning replay buffer;
28:     Generate batch samples according to the safety-informed batch sampling mechanism (14);
29:     Apply off-policy algorithm to update critic and actor networks, using the generated batch samples;
30:     Update time index: $t \leftarrow t + 1$;
31:     Go to Line 2;
32: **end if**

---

The working mechanism of our proposed Real-DRL is explained in Algorithm 1. Below, we provide additional clarifications. The actions of the activated DRL-Student and PHY-Teacher are highlighted in the yellow and green blocks, respectively. As indicated in Line 1 and Line 5, when the triggering condition described by (6) is met, the PHY-Teacher is activated to back up the safety of the real plant (as detailed in Lines 6 to 11) and to foster the teaching-to-learn paradigm (as described in Lines 12 to 16). Furthermore, Lines 17 to 23 outlines the process in which, once the PHY-Teacher has guided the real plant back to the self-learning space, control is returned to the DRL-Student. At that point, the DRL-Student is activated while the PHY-Teacher is deactivated.

The Lines 15 and 16 in the yellow block (i.e., the activation period of the PHY-Teacher) and Lines 28 and 29 in the green block (i.e., the activation period of the DRL-Student) indicate that during the activation period of either the DRL-Student or the PHY-Teacher, the critic and actor networks continue to update through off-policy algorithms. The generated batch samples are based on safety-informed batch sampling, which includes samples drawn from both the teach-to-learn and self-learning replay buffers. Their contribution ratios depend on the real-time safety status, represented as $V(\mathbf{s}(k))$. Therefore, we can conclude that Real-DRL operates within a dual paradigm of self-learning and teaching-to-learn during its runtime operation.

## C Proof of Theorem 4.1

The overall proof consists of the following two steps.

**Step 1: Conclude: Set $\{\mathbf{s} \in \mathbb{R}^n \mid V(\mathbf{s}) < 1\} \subseteq \mathbb{S}$.**

The safety set defined in Equation (2) can be transformed into an equivalent representation as follows:

$$\mathbb{S} = \left\{ \mathbf{s} \in \mathbb{R}^n \mid (\mathbf{C} \cdot \mathbf{diag}^{-1}\{\mathbf{c}\}) \cdot \mathbf{s} \leq \mathbf{1}_p \right\}. \tag{23}$$

Meanwhile, letting $\mathbf{Q} = \mathbf{P}^{-1}$, the conditions (i.e., $\mathbf{I}_p - (\mathbf{C} \cdot \mathbf{diag}^{-1}\{\mathbf{c}\}) \cdot \widetilde{\mathbf{P}}^{-1} \cdot (\mathbf{C} \cdot \mathbf{diag}^{-1}\{\mathbf{c}\})^\top \succ 0$ and $\widetilde{\mathbf{P}} \succ 0$) included in Equation (13) imply that

$$\mathbf{I}_p - (\mathbf{C} \cdot \mathbf{diag}^{-1}\{\mathbf{c}\}) \cdot \mathbf{P}^{-1} \cdot (\mathbf{C} \cdot \mathbf{diag}^{-1}\{\mathbf{c}\})^\top \succ 0 \text{ and } \mathbf{Q} \succ 0. \tag{24}$$

Then, by applying Lemma A.2 from Appendix A and taking into account the $V(\mathbf{s})$ defined in Equation (12), we conclude from Equations (23) and (24) that:

$$\{\mathbf{s} \in \mathbb{R}^n \mid \mathbf{s}^\top \cdot \mathbf{Q}^{-1} \cdot \mathbf{s} \leq 1\} = \{\mathbf{s} \in \mathbb{R}^n \mid \mathbf{s}^\top \cdot \mathbf{P} \cdot \mathbf{s} \leq 1\} = \{\mathbf{s} \in \mathbb{R}^n \mid V(\mathbf{s}) \leq 1\} \subseteq \mathbb{S} \tag{25}$$

**Step 2: Proof of maximum volume.**

We recall that the volume of an ellipsoid, defined by the set $\{\mathbf{s} \in \mathbb{R}^n \mid \mathbf{s}^\top \cdot \mathbf{P} \cdot \mathbf{s} \leq 1\} = \{\mathbf{s} \in \mathbb{R}^n \mid V(\mathbf{s}) \leq 1\}$, is proportional to $\sqrt{\det(\mathbf{P}^{-1})}$ [48, 53]. Therefore, maximizing the volume is equivalent to minimizing the determinant: $\det(\mathbf{P}^{-1})$. This leads to the optimal problem outlined in Equation (13). As a result, the ellipsoid set $\{\mathbf{s} \in \mathbb{R}^n \mid V(\mathbf{s}) \leq 1\}$ achieves the maximum volume. Thus, the proof is completed.

# D Proof of Theorem 5.2

According to Definition 2.3, to prove the PHY-Teacher – designed by Theorem 5.2 – has assured safety, we shall prove the PHY-Teacher's real-time actions and the resulted system states simultaneously satisfy three requirements. They are when $\mathbf{s}(k) \in \partial\mathbb{L}$, then i) the $\mathbf{s}(t) \in \mathbb{S}$ holds for any time $t \in \mathbb{T}_k$, ii) $\mathbf{a}_{\text{teacher}}(t) = \pi_{\text{teacher}}(\mathbf{s}(t)) \in \mathbb{A}$ holds for any time $t \in \mathbb{T}_k$, and iii) there exists a $\tau_k \in \mathbb{N}$, such that $\mathbf{s}(k + \tau_k) \in \mathbb{L}$. The overall proof will separated into seven steps to conclude the three conditions i)–iii) hold in **Steps 4, 5, and 7**, respectively. These steps are detailed below.

## Step 1: Derivations from the first inequality in Equation (21).

Recalling the Schur Complement in Lemma A.1 of Appendix A, and considering that $\mathbf{Q}_k \succ 0$, we conclude that the first inequality in Equation (21) is equivalent to

$$\alpha \cdot \mathbf{Q}_k - (1+\phi) \cdot (\mathbf{Q}_k \cdot \mathbf{A}^\top(\mathbf{s}(k)) + \mathbf{R}_k^\top \cdot \mathbf{B}^\top(\mathbf{s}(k))) \cdot \mathbf{Q}_k^{-1} \cdot (\mathbf{A}(\mathbf{s}(k)) \cdot \mathbf{Q}_k + \mathbf{B}(\mathbf{s}(k)) \cdot \mathbf{R}_k) \succ 0,$$

multiplying both the left-hand side and the right-hand side of which by $\mathbf{Q}_k^{-1}$ yields:

$$\alpha \cdot \mathbf{Q}_k^{-1} - (1+\phi) \cdot (\mathbf{A}^\top(\mathbf{s}(k)) + \mathbf{Q}_k^{-1} \cdot \mathbf{R}_k^\top \cdot \mathbf{B}^\top(\mathbf{s}(k))) \cdot \mathbf{Q}_k^{-1} \cdot (\mathbf{A}(\mathbf{s}(k)) + \mathbf{B}(\mathbf{s}(k)) \cdot \mathbf{R}_k \cdot \mathbf{Q}_k^{-1})$$
$$\succ 0,$$

substituting the definitions in Equation (19) into which, we obtain:

$$\alpha \cdot \mathbf{P}_k - (1 + \phi) \cdot (\mathbf{A}^\top(\mathbf{s}(k)) + \mathbf{F}_k^\top \cdot \mathbf{B}^\top(\mathbf{s}(k))) \cdot \mathbf{P}_k \cdot (\mathbf{A}(\mathbf{s}(k)) + \mathbf{B}(\mathbf{s}(k)) \cdot \mathbf{F}_k) \succ 0, \quad (26)$$

which is equivalent to

$$\alpha \cdot \mathbf{P}_k - (1 + \phi) \cdot \overline{\mathbf{A}}^\top(\mathbf{s}(k)) \cdot \mathbf{P}_k \cdot \overline{\mathbf{A}}(\mathbf{s}(k)) \succ 0, \quad (27)$$

where we define:

$$\overline{\mathbf{A}}(\mathbf{s}(k)) \triangleq \mathbf{A}(\mathbf{s}(k)) + \mathbf{B}(\mathbf{s}(k)) \cdot \mathbf{F}_k. \quad (28)$$

## Step 2: Derivations from the first inequality in Equation (20).

We define the set:

$$\Omega_k \triangleq \{ \mathbf{s} \mid (\mathbf{s} - \mathbf{s}_k^*)^\top \cdot \mathbf{Q}_k^{-1} \cdot (\mathbf{s} - \mathbf{s}_k^*) < 1, \quad \mathbf{Q}_k \succ 0 \}. \quad (29)$$

Considering the definition $\mathbf{e}(t) \triangleq \mathbf{s}(t) - \mathbf{s}_k^*$ and the transformation $\overline{\mathbf{C}} \triangleq \mathbf{C} \cdot \text{diag}^{-1}\{\theta \cdot \mathbf{c}\}$, the patch defined in Equation (15) can be equivalently expressed as $\mathbb{P}_k = \{ \mathbf{s} \in \mathbb{R}^n | \overline{\mathbf{C}} \cdot (\mathbf{s} - \mathbf{s}_k^*) < \mathbf{1} \}$. Meanwhile, by considering the first inequality in Equation (20) and applying Lemma A.2 of Appendix A, we conclude:

$$\Omega_k \subseteq \mathbb{P}_k \text{ because } \mathbf{I}_p - \overline{\mathbf{C}} \cdot \mathbf{Q}_k \cdot \overline{\mathbf{C}}^\top \succ 0. \quad (30)$$

We note that the patch in Equation (15) can also be rewritten as follows:

$$\mathbb{P}_k = \{ \mathbf{s} \in \mathbb{R}^n | -\theta \cdot \mathbf{c} + \mathbf{C} \cdot \mathbf{s}_k^* < \mathbf{C} \cdot \mathbf{s} < \theta \cdot \mathbf{c} + \mathbf{C} \cdot \mathbf{s}_k^* \}. \quad (31)$$

From Equations (4), (6) and (16), we obtain that $\mathbf{C} \cdot \mathbf{s}_k^* = \chi \cdot \mathbf{C} \cdot \mathbf{s}(k) = \chi \cdot \eta \cdot \mathbf{c}$ or $-\chi \cdot \eta \cdot \mathbf{c}$. Substituting this into Equation (31) yields:

$$\mathbb{P}_k = \{ \mathbf{s} \in \mathbb{R}^n | -(\theta - \chi \cdot \eta) \cdot \mathbf{c} < \mathbf{C} \cdot \mathbf{s} < (\theta + \chi \cdot \eta) \cdot \mathbf{c} \}$$
$$or \{ \mathbf{s} \in \mathbb{R}^n | -(\theta + \chi \cdot \eta) \cdot \mathbf{c} < \mathbf{C} \cdot \mathbf{s} < (\theta - \chi \cdot \eta) \cdot \mathbf{c} \}. \quad (32)$$

The first item in Equation (22) shows that $\theta + \chi \cdot \eta < 1$, which is equivalent to $-\theta - \chi \cdot \eta > -1$. Since both $\theta > 0$ and $\chi \cdot \eta > 0$, we can also deduce that $\theta - \chi \cdot \eta < 1$ and $-\theta + \chi \cdot \eta > -1$. Therefore, we conclude from Equations (4) and (32) that $\mathbb{P}_k \subseteq \mathbb{S}$. This, in combination with Equation (30), leads to the following result:

$$\Omega_k \subseteq \mathbb{P}_k \subseteq \mathbb{S}. \quad (33)$$

**Step 3: Derivations from the third inequality in Equation (20).**

The vector $\mathbf{s}(k)$ can have zero entries, for which we define:

$$[\bar{\mathbf{s}}(k,\delta)]_i \triangleq \begin{cases} [\bar{\mathbf{s}}(k)]_i + \delta, & \text{if } [\bar{\mathbf{s}}(k)]_i = 0 \\ [\mathbf{s}(k)]_i, & \text{otherwise} \end{cases}, \quad i = 1, 2, \ldots, n,$$

where $\delta \neq 0$ and $[\bar{\mathbf{s}}(k)]_i$ represents the $i$-th entry of the vector $\bar{\mathbf{s}}(k)$. Thus, the vector $\bar{\mathbf{s}}(k)$ contains no zero entries. We therefore conclude:

$$\lim_{\delta \to 0} \bar{\mathbf{s}}(k,\delta) = \mathbf{s}(k), \tag{34}$$

in light of which and the third inequality in Equation (20), we can conclude the following:

$$\lim_{\delta \to 0} \{\mathbf{Q}_k - n \cdot \mathbf{diag}^2(\bar{\mathbf{s}}(k,\delta))\} = \mathbf{Q}_k - n \cdot \mathbf{diag}^2(\mathbf{s}(k)) \succ 0. \tag{35}$$

The $\lim_{\delta \to 0}\{\mathbf{Q}_k - n \cdot \mathbf{diag}^2(\bar{\mathbf{s}}(k,\delta))\} \succ 0$ is equivalent to $\mathbf{Q}_k - \lim_{\delta \to 0}\{n \cdot \mathbf{diag}^2(\bar{\mathbf{s}}(k,\delta))\} \succ 0$. Given that $\bar{\mathbf{s}}(k,\delta)$ does not have zero entries and the $\mathbf{Q}_k \succ 0$, we can further conclude that $\lim_{\delta \to 0}\{n^{-1} \cdot \mathbf{diag}^{-2}(\bar{\mathbf{s}}(k,\delta))\} - \mathbf{Q}_k^{-1} \succ 0$, which means

$$n^{-1} \cdot \mathbf{I}_n - \lim_{\delta \to 0}\{\mathbf{diag}(\bar{\mathbf{s}}(k,\delta)) \cdot \mathbf{Q}_k^{-1} \cdot \mathbf{diag}(\bar{\mathbf{s}}(k,\delta))\} \succ 0,$$

from which, the $\mathbf{diag}(\bar{\mathbf{s}}(k)) \cdot \mathbf{1}_n = \bar{\mathbf{s}}(k)$, and the $\mathbf{Q}_k^{-1} = \mathbf{P}_k$, we have:

$$\lim_{\delta \to 0}\{\mathbf{1}_n^\top \cdot \mathbf{diag}(\bar{\mathbf{s}}(k,\delta)) \cdot \mathbf{Q}_k^{-1} \cdot \mathbf{diag}(\bar{\mathbf{s}}(k,\delta)) \cdot \mathbf{1}_n = \bar{\mathbf{s}}^\top(k,\delta) \cdot \mathbf{P}_k \cdot \bar{\mathbf{s}}(k,\delta)\} < n^{-1} \cdot \mathbf{1}_n^\top \cdot \mathbf{1}_n = 1.$$

In conjunction with Equation (34), this leads to the following conclusion:

$$\mathbf{s}^\top(k) \cdot \mathbf{P}_k \cdot \mathbf{s}(k) = \lim_{\delta \to 0}\{\bar{\mathbf{s}}^\top(k,\delta) \cdot \mathbf{P}_k \cdot \bar{\mathbf{s}}(k,\delta)\} < 1. \tag{36}$$

**Step 4: Conclude: $\mathbf{s}(t) \in \mathbb{S}, \forall t \in \mathbb{T}_k$.**

Considering the dynamics outlined in Equation (17), the action policy described in Equation (18), and the definition provided in Equation (28), we obtain:

$$\mathbf{e}^\top(t+1) \cdot \mathbf{P}_k \cdot \mathbf{e}(t+1) - \alpha \cdot \mathbf{e}^\top(t) \cdot \mathbf{P}_k \cdot \mathbf{e}(t)$$
$$= \mathbf{e}^\top(t) \cdot \left(\overline{\mathbf{A}}^\top(\mathbf{s}(k)) \cdot \mathbf{P}_k \cdot \overline{\mathbf{A}}(\mathbf{s}(k)) - \alpha \cdot \mathbf{P}_k\right) \cdot \mathbf{e}(t) + \mathbf{h}^\top(\mathbf{e}(t)) \cdot \mathbf{P}_k \cdot \mathbf{h}(\mathbf{e}(t))$$
$$+ 2 \cdot \mathbf{e}^\top(t) \cdot \overline{\mathbf{A}}^\top(\mathbf{s}(k)) \cdot \mathbf{P}_k \cdot \mathbf{h}(\mathbf{e}(t)), \quad t \in \mathbb{T}_k. \tag{37}$$

In light of Lemma A.4 of Appendix A, we have:

$$2 \cdot \mathbf{e}^\top(t) \cdot \overline{\mathbf{A}}^\top(\mathbf{s}(k)) \cdot \mathbf{P}_k \cdot \mathbf{h}(\mathbf{e}(t)) \leq \phi \cdot \mathbf{e}^\top(t) \cdot \overline{\mathbf{A}}^\top(\mathbf{s}(k)) \cdot \mathbf{P}_k \cdot \overline{\mathbf{A}}(\mathbf{s}(k)) \cdot \mathbf{e}(t)$$
$$+ \frac{1}{\phi} \cdot \mathbf{h}^\top(\mathbf{e}(t)) \cdot \mathbf{P}_k \cdot \mathbf{h}(\mathbf{e}(t)). \tag{38}$$

We note that Assumption 5.1 means:

$$\mathbf{h}^\top(\mathbf{e}(t)) \cdot \mathbf{P}_k \cdot \mathbf{h}(\mathbf{e}(t)) \leq \kappa, \quad \forall \mathbf{e}(t) \in \mathbb{P}_k. \tag{39}$$

Substituting inequalities in Equations (38) and (39) into Equation (37) yields:

$$\mathbf{e}^\top(t+1) \cdot \mathbf{P}_k \cdot \mathbf{e}(t+1) - \alpha \cdot \mathbf{e}^\top(t) \cdot \mathbf{P}_k \cdot \mathbf{e}(t)$$
$$\leq \mathbf{e}^\top(t) \cdot \left((1+\phi) \cdot \overline{\mathbf{A}}^\top(\mathbf{s}(k)) \cdot \mathbf{P}_k \cdot \overline{\mathbf{A}}(\mathbf{s}(k)) - \alpha \cdot \mathbf{P}_k\right) \cdot \mathbf{e}(t) + \frac{(\phi+1) \cdot \kappa}{\phi}, \quad t \in \mathbb{T}_k. \tag{40}$$

We now are able to conclude from Equations (27) and (40) that

$$\mathbf{e}^\top(t+1) \cdot \mathbf{P}_k \cdot \mathbf{e}(t+1) < \alpha \cdot \mathbf{e}^\top(t) \cdot \mathbf{P}_k \cdot \mathbf{e}(t) + \frac{(\phi+1) \cdot \kappa}{\phi}, \quad t \in \mathbb{T}_k. \tag{41}$$

The left-hand inequality in the first item of Equation (22) is equivalent to $(1 - \chi) \cdot \eta < \theta$. Given that $0 < \chi < 1$, this leads to the conclusion that $\frac{(1-\chi)^2 \cdot \eta^2}{\theta^2} < 1$. Considering this, we can derive from the second item in Equation (22) that $\frac{(\phi+1)\cdot\kappa}{(1-\alpha)\cdot\phi} < 1$, which can be rewritten as $\alpha + \frac{(\phi+1)\cdot\kappa}{\phi} < 1$. From this inequality and Equation (36), we obtain:

$$\mathbf{e}^\top (t+1) \cdot \mathbf{P}_k \cdot \mathbf{e} (t+1) < 1, \text{ for any } \mathbf{e}^\top (t) \cdot \mathbf{P}_k \cdot \mathbf{e} (t) < 1. \tag{42}$$

Additionally, recalling Equation (16), we have: $\mathbf{e}(k) = \mathbf{s}(k) - \mathbf{s}_k^* = \mathbf{s}(k) - \chi \cdot \mathbf{s}(k) = (1 - \chi) \cdot \mathbf{s}(k)$, where $0 < \chi < 1$ This expression, along with Equation (36), leads to the result: $\mathbf{e}^\top (k) \cdot \mathbf{P}_k \cdot \mathbf{e} (k) < 1$ Combining this with Equation (41) and considering the process of iteration, we can draw the following conclusion:

$$\mathbf{e}^\top (t) \cdot \mathbf{P}_k \cdot \mathbf{e} (t) \leq 1, \quad \forall t \in \mathbb{T}_k. \tag{43}$$

Furthermore, referring to $\mathbf{e}(t) \triangleq \mathbf{s}(t) - \mathbf{s}_k^*$ and Equation (29) with $\mathbf{Q}_k^{-1} = \mathbf{P}_k$, the result in Equation (43) means that $\mathbf{s}(t) \in \Omega_k, \forall t \in \mathbb{T}_k$. Finally, considering Equation (33), we reach the final conclusion:

$$\mathbf{s}(t) \in \Omega_k \subseteq \mathbb{P}_k \subseteq \mathbb{S}, \ \forall t \in \mathbb{T}_k. \tag{44}$$

**Step 5: Conclude: $\mathbf{a}_{\text{teacher}}(t) \in \mathbb{A}, \forall t \in \mathbb{T}_k$.**

According to Lemma A.1 in Appendix A, the second inequality in (21) implies the following:

$$\mathbf{Q}_k - \mathbf{R}_k^\top \cdot \mathbf{T}_k^{-1} \cdot \mathbf{R}_k \succ 0. \tag{45}$$

By substituting $\mathbf{F}_k \cdot \mathbf{Q}_k = \mathbf{R}_k$ into Equation (45), we obtain:

$$\mathbf{Q}_k - (\mathbf{F}_k \cdot \mathbf{Q}_k)^\top \cdot \mathbf{T}_k^{-1} \cdot (\mathbf{F}_k \cdot \mathbf{Q}_k) \succ 0. \tag{46}$$

multiplying both left-hand and right-hand sides of which by $\mathbf{Q}_k^{-1}$ yields:

$$\mathbf{Q}_k^{-1} - \mathbf{F}_k^\top \cdot \mathbf{T}_k^{-1} \cdot \mathbf{F}_k \succ 0,$$

from which and Equation (18) we thus have

$$\mathbf{e}^\top (t) \cdot \mathbf{Q}_k^{-1} \cdot \mathbf{e}(t) - \mathbf{e}^\top (t) \cdot \mathbf{F}_k^\top \cdot \mathbf{T}_k^{-1} \cdot \mathbf{F}_k \cdot \mathbf{e}(t) = \mathbf{e}^\top (t) \cdot \mathbf{Q}_k^{-1} \cdot \mathbf{e}(t)$$
$$- \mathbf{a}_{\text{teacher}}^\top (t) \cdot \mathbf{T}_k^{-1} \cdot \mathbf{a}_{\text{teacher}}(t) > 0. \tag{47}$$

We observe that the patch set in Equation (29) can be expressed in an equivalent form as follows:

$$\Omega_k \triangleq \{ \mathbf{e} \mid \mathbf{e}^\top \cdot \mathbf{Q}_k^{-1} \cdot \mathbf{e} < 1, \text{ where } \mathbf{Q}_k \succ 0 \text{ and } \mathbf{e} \triangleq \mathbf{s} - \mathbf{s}_k^* \}, \tag{48}$$

from which, we conclude that $\mathbf{e}(t) \in \Omega_k$ means $\mathbf{e}^\top (t) \cdot \mathbf{Q}_k^{-1} \cdot \mathbf{e}(t) < 1$. So, we can obtain from Equation (47) that

$$\mathbf{a}_{\text{teacher}}^\top (t) \cdot \mathbf{T}_k^{-1} \cdot \mathbf{a}_{\text{teacher}}(t) < \mathbf{e}^\top (t) \cdot \mathbf{Q}_k^{-1} \cdot \mathbf{e}(t) < 1, \text{ for } \mathbf{e}(t) \in \Omega_k. \tag{49}$$

Additionally, based on Equation (48), the conclusion in Equation (44) indicates that $\mathbf{e}(t) \in \Omega_k, \forall t \in \mathbb{T}_k$. This, in light of Equation (49), results in

$$\mathbf{a}_{\text{teacher}}^\top (t) \cdot \mathbf{T}_k^{-1} \cdot \mathbf{a}_{\text{teacher}}(t) < 1, \ \forall t \in \mathbb{T}_k. \tag{50}$$

We define a set: $\Xi_k \triangleq \{ \mathbf{a} \in \mathbb{R}^m | \mathbf{a}^\top \cdot \mathbf{T}_k^{-1} \cdot \mathbf{a} < 1, \mathbf{T}_k \succ 0 \}$. In light of Equation (50), we obtain:

$$\mathbf{a}_{\text{teacher}}(t) \in \Xi_k, \ \forall t \in \mathbb{T}_k. \tag{51}$$

Noting $\overline{\mathbf{D}} = \mathbf{D} \cdot \mathbf{diag}^{-1}\{\mathbf{d}\}$, the definition of action set $\mathbb{A}$ in Equation (3) can be equivalently expressed as $\mathbb{A} \triangleq \{ \mathbf{a} \in \mathbb{R}^m | \overline{\mathbf{D}} \cdot \mathbf{a} \leq \mathbf{1} \}$. Besides, recalling the second inequality (i.e., $\mathbf{I}_q - \overline{\mathbf{D}} \cdot \mathbf{T}_k \cdot \overline{\mathbf{D}}^\top \succ 0$) from Equation (20) and applying Lemma A.3 with consideration of Equation (51), we can conclude that:

$$\mathbf{a}_{\text{teacher}}(t) \in \Xi_k \subseteq \mathbb{A}, \ \forall t \in \mathbb{T}_k. \tag{52}$$

**Step 6: Further derivations from the first inequality in Equation (20).**

Referring to Equation (15), we define an auxiliary real-time set as follows:

$$\mathbb{G}_k \triangleq \left\{ \mathbf{s} \in \mathbb{R}^n \,|\, -\gamma \cdot \mathbf{c} \leq \mathbf{C} \cdot (\mathbf{s} - \mathbf{s}_k^*) \leq \gamma \cdot \mathbf{c}, \text{ with } \gamma = (1 - \chi) \cdot \eta < \theta \right\}, \tag{53}$$

where $\chi$ is given in Equation (16), $\eta$ is given in Equation (4), and $\eta$ is given in Equation (15). The $\mathbb{G}_k$ in Equation (53) can be equivalently expressed as

$$\mathbb{G}_k = \left\{ \mathbf{s} \in \mathbb{R}^n \,|\, \mathbf{C} \cdot \mathbf{s}_k^* - \gamma \cdot \mathbf{c} \leq \mathbf{C} \cdot \mathbf{s} \leq \mathbf{C} \cdot \mathbf{s}_k^* + \gamma \cdot \mathbf{c}, \text{ with } \gamma = (1 - \chi) \cdot \eta < \theta \right\}. \tag{54}$$

We recall from Equation (6) that $\mathbf{s}(k)$ denotes a system state that approaches the boundary of safe self-learning space $\mathbb{L}$ defined in Equation (4). Hereto, by considering $\mathbf{s}_k^* = \chi \cdot \mathbf{s}(k)$ and Equation (15), we have $-\chi \cdot \eta \cdot \mathbf{c} \leq \mathbf{C} \cdot \mathbf{s}_k^* \leq \chi \cdot \eta \cdot \mathbf{c}$, substituting which into (54) yields:

$$\mathbb{G}_k = \left\{ \mathbf{s} \in \mathbb{R}^n \,|\, (\chi \cdot \eta - \gamma) \cdot \mathbf{c} < \mathbf{C} \cdot \mathbf{s} < (\chi \cdot \eta + \gamma) \cdot \mathbf{c}, \text{ with } \gamma = (1 - \chi) \cdot \eta < \theta \right\}. \tag{55}$$

We note that the relationship $\gamma = (1 - \chi) \cdot \eta$ is equivalent to the equation $\chi \cdot \eta + \gamma = \eta$. This implies that $\chi \cdot \eta - \gamma > -\eta$, considering that $\chi \cdot \eta > 0$. In summary, we establish both $\chi \cdot \eta + \gamma = \eta$ and $\chi \cdot \eta - \gamma > -\eta$. Considering them, we can derive result from Equations (4) and (55) as follows:

$$\mathbb{G}_k \subset \mathbb{L}. \tag{56}$$

Considering that $\mathbf{P}_k = \mathbf{Q}_k^{-1}$ and the patch defined in Equation (15), we can derive from Equations (29) and (30) that:

$$\left\{ \mathbf{s} \in \mathbb{R}^n \,|\, (\mathbf{s} - \mathbf{s}_k^*)^\top \cdot \mathbf{P}_k \cdot (\mathbf{s} - \mathbf{s}_k^*) \leq 1 \right\} \subseteq \left\{ \mathbf{s} \in \mathbb{R}^n \,|\, -\theta \cdot \mathbf{c} \leq \mathbf{C} \cdot (\mathbf{s} - \mathbf{s}_k^*) \leq \theta \cdot \mathbf{c} \right\},$$
$$\text{if } \mathbf{I}_p - \overline{\mathbf{C}} \cdot \mathbf{P}_k^{-1} \cdot \overline{\mathbf{C}}^\top \succ 0 \tag{57}$$

where $\overline{\mathbf{C}} \triangleq \mathbf{C} \cdot \text{diag}^{-1}\{\theta \cdot \mathbf{c}\}$. Letting $\widehat{\mathbf{C}} \triangleq \mathbf{C} \cdot \text{diag}^{-1}\{\gamma \cdot \mathbf{c}\}$ and considering $\gamma > 0$, we can obtain from Equation (57) that

$$\left\{ \mathbf{s} \in \mathbb{R}^n \,|\, (\mathbf{s} - \mathbf{s}_k^*)^\top \cdot (\frac{\theta^2}{\gamma^2} \cdot \mathbf{P}_k) \cdot (\mathbf{s} - \mathbf{s}_k^*) \leq 1 \right\} \subseteq \left\{ \mathbf{s} \in \mathbb{R}^n \,|\, -\gamma \cdot \mathbf{c} \leq \mathbf{C} \cdot (\mathbf{s} - \mathbf{s}_k^*) \leq \gamma \cdot \mathbf{c} \right\},$$
$$\text{if } \mathbf{I}_p - \widehat{\mathbf{C}} \cdot (\frac{\gamma^2}{\theta^2} \cdot \mathbf{P}_k^{-1}) \cdot \widehat{\mathbf{C}}^\top \succ 0. \tag{58}$$

Moreover, it is straightforward to verify from $\overline{\mathbf{C}} \triangleq \mathbf{C} \cdot \text{diag}^{-1}\{\theta \cdot \mathbf{c}\}$ and $\widehat{\mathbf{C}} \triangleq \mathbf{C} \cdot \text{diag}^{-1}\{\gamma \cdot \mathbf{c}\}$ that $\widehat{\mathbf{C}} \cdot (\frac{\gamma^2}{\theta^2} \cdot \mathbf{P}_k^{-1}) \cdot \widehat{\mathbf{C}}^\top = \overline{\mathbf{C}} \cdot \mathbf{P}_k^{-1} \cdot \overline{\mathbf{C}}^\top$. So, with the consideration of Equation (53), we can further conclude from Equations (57) and (58) that:

$$\left\{ \mathbf{s} \in \mathbb{R}^n \,|\, (\mathbf{s} - \mathbf{s}_k^*)^\top \cdot \mathbf{P}_k \cdot (\mathbf{s} - \mathbf{s}_k^*)^\top \leq \frac{\gamma^2}{\theta^2} \right\} \subseteq \mathbb{G}_k, \text{ if } \mathbf{I}_p - \overline{\mathbf{C}} \cdot \mathbf{P}_k^{-1} \cdot \overline{\mathbf{C}}^\top \succ 0. \tag{59}$$

Furthermore, recalling the first inequality (i.e., $\mathbf{I}_p - \overline{\mathbf{C}} \cdot \mathbf{Q}_k \cdot \overline{\mathbf{C}}^\top \succ 0$) from Equation (20), we arrive at the conclusion $\left\{ \mathbf{s} \in \mathbb{R}^n \,|\, (\mathbf{s} - \mathbf{s}_k^*)^\top \cdot \mathbf{P}_k \cdot (\mathbf{s} - \mathbf{s}_k^*) \leq \frac{\gamma^2}{\theta^2} \right\} \subseteq \mathbb{G}_k$. This, combined with Equation (56), leads to the following result:

$$\left\{ \mathbf{s} \in \mathbb{R}^n \,|\, (\mathbf{s} - \mathbf{s}_k^*)^\top \cdot \mathbf{P}_k \cdot (\mathbf{s} - \mathbf{s}_k^*) \leq \frac{\gamma^2}{\theta^2} \right\} \subset \mathbb{L},$$

substituting $\gamma = (1 - \chi) \cdot \eta$ into which yields:

$$\left\{ \mathbf{s} \in \mathbb{R}^n \,|\, (\mathbf{s} - \mathbf{s}_k^*)^\top \cdot \mathbf{P}_k \cdot (\mathbf{s} - \mathbf{s}_k^*) \leq \frac{(1 - \chi)^2 \cdot \eta^2}{\theta^2} \right\} \subset \mathbb{L}. \tag{60}$$

**Step 7: Conclude: $\mathbf{s}(k + \tau_k) \in \mathbb{L}$ for a $\tau_k \in \mathbb{N}$.**

We can derive from (41) that

$$\mathbf{e}^\top (k + \tau) \cdot \mathbf{P}_k \cdot \mathbf{e}(k + \tau) \leq \alpha^\tau \cdot \mathbf{e}^\top (k) \cdot \mathbf{P}_k \cdot \mathbf{e}(k) + \frac{(\phi + 1) \cdot \kappa}{\phi} \cdot \frac{1 - \alpha^{\tau - 1}}{1 - \alpha},$$

where $0 < \alpha < 1$. This leads to the following result:

$$\mathbf{e}^\top (\infty) \cdot \mathbf{P}_k \cdot \mathbf{e}(\infty) \leq \frac{(\phi + 1) \cdot \kappa}{(1 - \alpha) \cdot \phi},$$

which with the second item in Equation (22) leads to

$$\mathbf{e}^\top (\infty) \cdot \mathbf{P}_k \cdot \mathbf{e}(\infty) \leq \frac{(\phi + 1) \cdot \kappa}{(1 - \alpha) \cdot \phi} < \frac{(1 - \chi)^2 \cdot \eta^2}{\theta^2}. \tag{61}$$

Given that $\mathbf{e}(t) \triangleq \mathbf{s}(t) - \mathbf{s}_k^*$, we can equivalently express the result in Equation (61) as:

$$(\mathbf{s}(\infty) - \mathbf{s}_k^*)^\top \cdot \mathbf{P}_k \cdot (\mathbf{s}(\infty) - \mathbf{s}_k^*) \leq \frac{(\phi + 1) \cdot \kappa}{(1 - \alpha) \cdot \phi} < \frac{(1 - \chi)^2 \cdot \eta^2}{\theta^2}. \tag{62}$$

Finally, we can conclude from Equations (60) and (62) that $\mathbf{s}(\infty) \in \mathbb{L}$. This indicates that there exists a $\tau_k \in \mathbb{N}$ such that $\mathbf{s}(k + \tau_k) \in \mathbb{L}$. Thus, the proof is completed.

# E   Pseudocode: Computation of PHY-Teacher's Action Policy

Referring to Equation (18), the computation of the PHY-Teacher's action policy involves calculating $\mathbf{F}_k$ based on the inequalities presented in Equations (20) and (21). To compute $\mathbf{F}_k$, we can utilize the Python CVXPY toolbox [26] or the Matlab LMI toolbox [24], as described in Algorithm 2 and Algorithm 3, respectively. In both Algorithm 2 and Algorithm 3, the A, B, s, C, D, and F correspond to $\mathbf{A}(\mathbf{s}(k))$, $\mathbf{B}(\mathbf{s}(k))$, $\mathbf{s}(k)$, $\overline{\mathbf{C}}$, $\overline{\mathbf{D}}$, and $\mathbf{F}_k$, respectively.

Finally, we recall that the ECVXCONE toolbox described in Section 6 can solve the LMIs using C

---

**Algorithm 2** Python CVXPY Toolbox for Computing $\mathbf{F}_k$ from Equations (20) and (21)

1: **Input:** A, B, s, C, D, m, n, p, q, and $0 < \alpha < 1$ and $\phi > 0$ satisfying conditions in Equation (22).
2: Q = cp.Variable((n, n), PSD=True);
3: T = cp.Variable((m, m), PSD=True);
4: R = cp.Variable((m, n));
5: constraints = [cp.bmat([[$\alpha$ * Q, Q @ A.T + R.T @ B.T], [A @ Q + B @ R, Q / (1 + $\phi$)]]) » 0;
6: Q - n * np.diag(s) @ np.diag(s) » 0;
7: np.identity(p) - C @ T @ C.T» 0;
8: np.identity(q) - D @ Q @ D.T » 0;
9: problem = cp.Problem(cp.Minimize(0), constraints);
10: problem.solve(solver=cp.CVXOPT);
11: optimal_Q = Q.value;
12: optimal_R = R.value;
13: P = np.linalg.inv(optimal_Q);
14: F = optimal_R @ P.

---

**Algorithm 3** Matlab LMI Toolbox for Computing $\mathbf{F}_k$ from Equations (20) and (21)

1: **Input:** A, B, s, C, D, m, n, p, q, and $0 < \alpha < 1$ and $\phi > 0$ satisfying conditions in Equation (22).
2: setlmis([]) ;
3: Q = lmivar(1,[n 1]);
4: T = lmivar(1,[m 1]);
5: R = lmivar(2,[m n]);
6: lmiterm([-1 1 1 Q],1,$\alpha$);
7: lmiterm([-1 2 1 Q],A,1);
8: lmiterm([-1 2 1 R],B,1);
9: lmiterm([-1 2 2 Q],1,1/(1 + $\phi$));
10: lmiterm([-2 1 1 Q],1,1);
11: lmiterm([-2 2 1 R],1,1);
12: lmiterm([-2 2 2 T],1,1);
13: lmiterm([-3 1 1 Q],1,1);
14: lmiterm([-3 1 1 0],-n*np.diag(s)*np.diag(s));
15: lmiterm([-4 1 1 Q],-D,D');
16: lmiterm([-4 1 1 0],eye(p));
17: lmiterm([-5 1 1 T],-C,C');
18: lmiterm([-5 1 1 0],eye(q));
19: mylmi = getlmis;
20: [tmin, psol] = feasp(mylmi);
21: Q = dec2mat(mylmi, psol, Q);
22: R = dec2mat(mylmi, psol, R);
23: P = inv(Q)
24: F = R * P.

---

**Remark E.1.** *We observed that the computation time for the methods described in Algorithm 2 and Algorithm 3 is quite short, ranging from 0.01 to 0.04 seconds (see evaluation in Table 2); therefore, its impact can be considered negligible. However, we found that the default CVX solver, SCS, demonstrated instability and inconsistent accuracy across different computing platforms. As a result, we opted to use the CVXOPT solver [1] in our experiments to achieve more stable and accurate results. This issue, however, does not arise when using the Matlab LMI toolbox.*

# F    Compensation for Large Model Mismatch

When the actual dynamics described by Equation (17) experiences significant model mismatch, the physics-model knowledge $(\mathbf{A}(\mathbf{s}(k)), \mathbf{B}(\mathbf{s}(k))$ may be inadequate. In such situations, the PHY-Teacher should implement a strategy to address this considerable model mismatch. This section outlines the technique used to compensate for the discrepancies in the model.

## F.1    Computation of Compensation Action

Given the state sample at the current time step $t$ (denoted as $\mathbf{e}(t)$), we can obtain the current control command $\mathbf{a}_{\text{teacher}}(t)$, calculated according to the method described in Equation (18). In addition to this information, we also have the previous state $\mathbf{e}(t-1)$ and the physics-model knowledge represented by $(\mathbf{A}(\mathbf{s}(k)), \mathbf{B}(\mathbf{s}(k))$. We can use the dynamics outlined in Equation (17) to predict the current state as follows:

$$\widehat{\mathbf{e}}(t) = \mathbf{A}(\mathbf{s}(k)) \cdot \mathbf{e}(t-1) + \mathbf{B}(\mathbf{s}(k)) \cdot \mathbf{a}_{\text{teacher}}(t), \tag{63}$$

where $\widehat{\mathbf{e}}(t)$ denotes the predicted state at time $t$.

In the context of compensating for model mismatch, the dynamics at the current time, as described in (17), can be expressed in an equivalent manner:

$$\mathbf{e}(t) = \mathbf{A}(\mathbf{s}(k)) \cdot \mathbf{e}(t-1) + \mathbf{B}(\mathbf{s}(k)) \cdot (\mathbf{a}_{\text{teacher}}(t) + \mathbf{a}_{\text{compensation}}(t)) + \widetilde{\mathbf{h}}(\mathbf{e}(t)), \tag{64}$$

where the term $\mathbf{a}_{\text{compensation}}(t)$ signifies the real-time actions taken to address the model mismatch $\widetilde{\mathbf{h}}(\mathbf{e}(t))$.

We can further derive from Equations (63) and (64) that

$$\mathbf{B}(\mathbf{s}(k)) \cdot \mathbf{a}_{\text{compensation}}(t)) + \widetilde{\mathbf{h}}(\mathbf{e}(t) = \mathbf{e}(t) - \widehat{\mathbf{e}}(t), \tag{65}$$

where $\mathbf{e}(t)$ and $\widehat{\mathbf{e}}(t)$ are known. From Equation (65), we find that if the compensation action meets the condition outlined in Equation (66), the model mismatch $\widetilde{\mathbf{h}}(\mathbf{e}(t))$ in Equation (64) can be effectively compensated by $\mathbf{a}_{\text{compensation}}(t)$) (i.e., $\widetilde{\mathbf{h}}(\mathbf{e}(t) = \mathbf{0}_n$).

$$\mathbf{B}(\mathbf{s}(k)) \cdot \mathbf{a}_{\text{compensation}}(t)) = \mathbf{e}(t) - \widehat{\mathbf{e}}(t), \tag{66}$$

We next explain how to calculate the compensation action, denoted as $\mathbf{a}_{\text{compensation}}(t)$, based on the conditions outlined in Equation (66).

By multiplying the left-hand side of Equation (66) by $\mathbf{B}^{\top}(\mathbf{s}(k))$, we obtain

$$(\mathbf{B}^{\top}(\mathbf{s}(k)) \cdot \mathbf{B}(\mathbf{s}(k))) \cdot \mathbf{a}_{\text{compensation}}(t) = \mathbf{B}^{\top}(\mathbf{s}(k)) \cdot (\mathbf{e}(t) - \widehat{\mathbf{e}}(t)). \tag{67}$$

By incorporating a sufficiently small value of $\lambda > 0$ in Equation (67), we achieve the following outcome:

$$(\mathbf{B}^{\top}(\mathbf{s}(k)) \cdot \mathbf{B}(\mathbf{s}(k)) + \lambda \cdot \mathbf{I}_m) \cdot \mathbf{a}_{\text{compensation}}(t) \approx \mathbf{B}^{\top}(\mathbf{s}(k)) \cdot (\mathbf{e}(t) - \widehat{\mathbf{e}}(t)),$$

which, in light of Lemma A.5 in Appendix A, the solution for $\mathbf{a}_{\text{compensation}}(t)$ is yielded as follows:

$$\mathbf{a}_{\text{compensation}}(t) \approx (\mathbf{B}^{\top}(\mathbf{s}(k)) \cdot \mathbf{B}(\mathbf{s}(k)) + \lambda \cdot \mathbf{I}_m)^{-1} \cdot \mathbf{B}^{\top}(\mathbf{s}(k)) \cdot (\mathbf{e}(t) - \widehat{\mathbf{e}}(t)). \tag{68}$$

## F.2    Conditional Usage of Compensation Action

At the current time $t$, we possess the following information: the current state $\mathbf{e}(t)$, the previous state $\mathbf{e}(t-1)$, the previous action $\mathbf{a}(t-1)$, and the physics model knowledge denoted by $(\mathbf{A}(\mathbf{s}(k)), \mathbf{B}(\mathbf{s}(k)))$. With this data, we can estimate the real-time value of model mismatch at time $t$ using the dynamics described in Equation (17). The estimation of model mismatch can be expressed as follows:

$$\mathbf{h}(\mathbf{e}(t)) \approx \mathbf{h}(\mathbf{e}(t-1)) = \mathbf{e}(t) - \mathbf{A}(\mathbf{s}(k)) \cdot \mathbf{e}(t-1) + \mathbf{B}(\mathbf{s}(k)) \cdot \mathbf{a}_{\text{teacher}}(t-1),$$

where $\mathbf{h}(\mathbf{e}(t)) \approx \mathbf{h}(\mathbf{e}(t-1))$ holds true due to the sufficiently small sampling period of sensors. Given the estimated model mismatch, we can carry out a real-time evaluation of Assumption 5.1 to decide whether to implement the compensation action. The process is outlined as follows:

$$\mathbf{a}_{\text{teacher}}(t) \leftarrow \begin{cases} \mathbf{a}_{\text{teacher}}(t) + \mathbf{a}_{\text{compensation}}(t), & \text{if } \mathbf{h}^{\top}(\mathbf{e}) \cdot \mathbf{P_k} \cdot \mathbf{h}(\mathbf{e}) > \kappa, \\ \mathbf{a}_{\text{teacher}}(t), & \text{if } \mathbf{h}^{\top}(\mathbf{e}) \cdot \mathbf{P_k} \cdot \mathbf{h}(\mathbf{e}) \leq \kappa. \end{cases}$$

# G   Experimental Evaluation: Python CVXPY v.s. C ECVXCONE

Table 2: Python CVXPY v.s. C ECVXCONE

| Hardware Platforms | CPU Configurations | | | Runtime Memory Usage | | LMIs Solve Time | |
|---|---|---|---|---|---|---|---|
| | Arch | Core | Frequency | CVXPY | ECVXCONE | CVXPY | ECVXCONE |
| Dell XPS 8960 Desktop | x86/64 | 32 | 5.4 GHz | 485 MB | 9.87 MB | 49.15 ms | 13.81 ms |
| Intel GEEKOM XT 13 Pro Mini | x86/64 | 20 | 4.7 GHz | 443 MB | 7.32 MB | 61.76 ms | 33.26 ms |
| NVIDIA Jetson AGX Orin | ARM64 | 12 | 2.2 GHz | 423 MB | 8.16 MB | 137.54 ms | 35.73 ms |
| Raspberry Pi 4 Model B | ARM64 | 4 | 1.5 GHz | 436 MB | 8.21 MB | 509.41 ms | 149.87 ms |

We summarize the experimental evaluation comparing Python CVXPY and C ECVXCONE regarding computational resource usage across four platforms with varying CPU frequencies in Table 2. Both CVXPY and ECVXCONE solve the same LMIs for the experiment 'Quadruped Robot in NVIDIA Isaac Gym: Wild Environment' in Section 7.2.

# H    Unknown Unknown: Randomized Beta Distribution

In the real world, plants can encounter a multitude of unknown variables, each with unique characteristics. To tackle this challenge, we propose utilizing a variant of the Beta distribution [31] to effectively model one type of these unknowns. This approach holds promise in mathematically defining and addressing these uncertainties.

**Definition H.1** (Randomized Beta Distribution). *The disturbance, noise, or fault, denoted by* $\mathbf{d}(k)$, *is considered to be a bounded unknown if (i)* $\mathbf{d}(k) \sim Beta(\alpha(k), \beta(k), c, a)$, *and (ii)* $\alpha(k)$ *and* $\beta(k)$ *are random parameters. In other words, the disturbance* $\mathbf{d}(k)$ *is within the range of [a, c], and its probability density function (pdf) is given by*

$$f\left(\mathbf{d}(k);\ \alpha(k),\ \beta(k),\ a,\ c\right) = \frac{(\mathbf{d}(k) - a)^{\alpha - 1}(c - \mathbf{d}(k))^{\alpha(k)-1}\Gamma\left(\alpha(k) + \beta(k)\right)}{(c - a)^{\alpha(k)+\beta(k)-1}\Gamma\left(\alpha(k)\right)\Gamma\left(\beta(k)\right)}, \quad (69)$$

*where* $\Gamma\left(\alpha(k)\right) = \int_0^\infty t^{\alpha(k)-1}e^{-t}dt,\ \mathrm{Re}\left(\alpha(k)\right) > 0,\ \alpha(k)$ *and* $\beta(k)$ *are randomly given at every k.*

The randomized Beta distribution, as defined in Definition H.1, is essential for describing a particular type of unknown unknown. This is important for two main reasons.

First, the nature of unknown unknowns is characterized by a lack of historical data, making them highly unpredictable in terms of time and distribution. As a result, existing models for scientific discoveries and understanding are often insufficient.

Second, as the examples shown in Figure 7, the parameters $\alpha$ and $\beta$ directly influence the probability density function (pdf) of the distribution, and consequently, the mean and variance. Suppose $\alpha$ and $\beta$ are randomized (expressed as $\alpha(k)$ and $\beta(k)$). In that case, the distribution of $\mathbf{d}(k)$ can take the form of a uniform distribution, exponential distribution, truncated Gaussian distribution, or a combination of these. However, the specific distribution is unknown. Therefore, the randomized $\alpha(k)$ and $\beta(k)$, which result in a randomized Beta distribution, can effectively capture the characteristics of "unavailable model" and "unforeseen" traits associated with unknown unknowns in both time and distribution. Furthermore, the randomized Beta distributions are bounded, with the bounds denoted as $a$ and $c$. This is motivated by the fact that, in general, there are no probabilistic solutions for handling unbounded unknowns, such as earthquakes and volcanic eruptions.

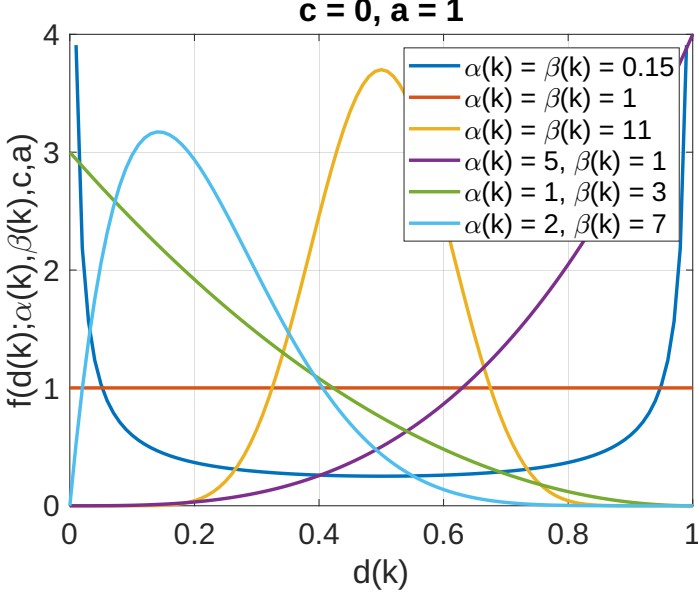

Figure 7: $\alpha(k)$ and $\beta(k)$ control the probability density function of the distribution.

# I  Experiment Design: Real Quadruped Robot: Indoor Environment

In our real quadruped robot experiment, we used a Python-based framework developed for the Unitree A1 robot, which was released on GitHub by [57]. This framework includes a PyBullet-based simulator, an interface for direct simulation-to-real transfer, and an implementation of the Convex Model Predictive Controller for essential motion control.

## I.1  Learning Configurations and Computation Resources

The runtime learning machine and Phy-DRL are designed to accomplish the safe mission detailed in Section 7.2. The policy observation consists of a 10-dimensional tracking error vector that represents the difference between the robot's state vector and the mission vector. Both systems utilize off-policy algorithm DDPG.

The actor and critic networks are implemented as MLPs with four fully connected layers. The output dimensions for the critic network are 256, 128, 64, and 1, while those for the actor network are 256, 128, 64, and 6. The input to the critic network comprises the tracking error vector and the action vector, while the input to the actor network is solely the tracking error vector. The activation functions for the first three layers of the neural networks are ReLU, while the output layer of the actor network uses the Tanh function and the critic network uses a Linear function. Additionally, the discount factor $\gamma$ is set to 0.9, and the learning rates for both the critic and actor networks are set at 0.0003. Finally, the batch size is configured to 512.

For computation resources, we utilized a desktop running Ubuntu 22.04, equipped with a 12th Gen Intel(R) Core(TM) i9-12900K 16-core processor, 64 GB of RAM, and an NVIDIA GeForce GTX 3090 GPU. The algorithm was implemented in Python, utilizing the TensorFlow framework alongside the Python CVXPY toolbox for solving real-time patches.

## I.2  Safety Conditions

The robot's learning process involves controlling its center of mass (CoM) height, CoM x-velocity, and other states in order to track the commands $r_{v_x}$, $r_h$, and zeros, respectively, while maintaining system states within a safety set, denoted as $\mathbb{S}$. This safety set is defined as: $\mathbb{S} = \left\{ \mathbf{s} \mid |\text{CoM x-velocity} - r_{v_x}| \leq 0.6 \text{ m/s}, \ |\text{CoM z-height} - r_h| \leq 0.12 \text{ m} \right\}$. Additionally, PHY-Teacher's action space is defined as: $\mathbb{A} = \left\{ \mathbf{a} \in \mathbb{R}^6 \mid |\mathbf{a}| \leq [10, 10, 10, 20, 20, 20]^\top \right\}$. This ensures that the robot operates within specified limits for safety and performance.

## I.3  PHY-Teacher Design

The robot's physics-model knowledge used in the design relies on its dynamics, which focuses on a single rigid body subject to forces at the contact points [19]. Our model of the robot's dynamics includes several key parameters: the CoM height (h), the CoM velocity ($\mathbf{v}$), represented as a 3D vector [CoM x-velocity, CoM y-velocity, CoM z-velocity]$^\top$, the Euler angles ($\widetilde{\mathbf{e}}$), described as a 3D vector [roll, pitch, yaw]$^\top$, and the angular velocity in world coordinates. According to [19], this model below effectively characterizes the dynamics of quadruped robots.

$$\frac{\mathbf{d}}{\mathbf{d}t} \underbrace{\begin{bmatrix} h \\ \widetilde{\mathbf{e}} \\ \mathbf{v} \\ \mathbf{w} \end{bmatrix}}_{\triangleq \widehat{\mathbf{s}}} = \underbrace{\begin{bmatrix} \mathbf{O}_{1\times 1} & \mathbf{O}_{1\times 5} & 1 & \mathbf{O}_{1\times 3} \\ \mathbf{O}_{3\times 3} & \mathbf{O}_{3\times 3} & \mathbf{O}_{3\times 3} & \mathbf{R}(\phi,\theta,\psi) \\ \mathbf{O}_{3\times 3} & \mathbf{O}_{3\times 3} & \mathbf{O}_{3\times 3} & \mathbf{O}_{3\times 3} \\ \mathbf{O}_{3\times 3} & \mathbf{O}_{3\times 3} & \mathbf{O}_{3\times 3} & \mathbf{O}_{3\times 3} \end{bmatrix}}_{\triangleq \widehat{\mathbf{A}}(\phi,\theta,\psi)} \cdot \begin{bmatrix} h \\ \widetilde{\mathbf{e}} \\ \mathbf{v} \\ \mathbf{w} \end{bmatrix} + \widehat{\mathbf{B}} \cdot \widehat{a} + \begin{bmatrix} 0 \\ \mathbf{O}_{3\times 1} \\ \mathbf{O}_{3\times 1} \\ \widetilde{\mathbf{g}} \end{bmatrix}$$
$$+ \ \mathbf{f}(\widehat{\mathbf{s}}), \quad (70)$$

where $\widetilde{\mathbf{g}} = [0, 0, -g]^\top \in \mathbb{R}^3$, with $g$ being the gravitational acceleration. $\mathbf{f}(\widehat{\mathbf{s}})$ denotes unknown model mismatch, $\widehat{\mathbf{B}} = [\mathbf{O}_{6\times 4}, \ \mathbf{I}_6]^\top$, and $\mathbf{R}(\phi,\theta,\psi) = \mathbf{R}_z(\psi) \cdot \mathbf{R}_y(\theta) \cdot \mathbf{R}_x(\phi) \in \mathbb{R}^{3\times 3}$ is the rotation matrix, with

$$\mathbf{R}_x(\phi) = \begin{bmatrix} 1 & 0 & 0 \\ 0 & \cos\phi & -\sin\phi \\ 0 & \sin\phi & \cos\phi \end{bmatrix}, \mathbf{R}_y(\theta) = \begin{bmatrix} \cos\theta & 0 & \sin\theta \\ 0 & 1 & 0 \\ -\sin\theta & 0 & \cos\theta \end{bmatrix}, \mathbf{R}_z(\psi) = \begin{bmatrix} \cos\psi & -\sin\psi & 0 \\ \sin\psi & \cos\psi & 0 \\ 0 & 0 & 1 \end{bmatrix}.$$

### I.3.1 Physics-Model Knowledge

Given the equilibrium point (control goal) $\mathbf{s}^*$, we define $\mathbf{s} \triangleq \widetilde{\mathbf{s}} - \mathbf{s}^*$. From this, we can obtain the following state-space model derived from the dynamics model in Equation (70).

$$\frac{d}{dt} \underbrace{\begin{bmatrix} h \\ \widetilde{\mathbf{e}} \\ \mathbf{v} \\ \mathbf{w} \end{bmatrix}}_{\mathbf{s}} = \underbrace{\begin{bmatrix} \mathbf{O}_{1\times1} & \mathbf{O}_{1\times3} & 1 & \mathbf{O}_{1\times5} \\ \mathbf{O}_{3\times1} & \mathbf{O}_{3\times3} & \mathbf{O}_{3\times3} & \mathbf{R}(\phi,\theta,\psi) \\ \mathbf{O}_{3\times1} & \mathbf{O}_{3\times3} & \mathbf{O}_{3\times3} & \mathbf{O}_{3\times3} \\ \mathbf{O}_{3\times1} & \mathbf{O}_{3\times3} & \mathbf{O}_{3\times3} & \mathbf{O}_{3\times3} \end{bmatrix}}_{\widehat{\mathbf{A}}(\mathbf{s})} \cdot \begin{bmatrix} h \\ \widetilde{\mathbf{e}} \\ \mathbf{v} \\ \mathbf{w} \end{bmatrix}$$

$$+ \underbrace{\begin{bmatrix} \mathbf{O}_{1\times3} & \mathbf{O}_{1\times3} & \mathbf{O}_{1\times3} & \mathbf{O}_{1\times3} \\ \mathbf{O}_{3\times3} & \mathbf{O}_{3\times3} & \mathbf{O}_3 & \mathbf{O}_{3\times3} \\ \mathbf{O}_{3\times3} & \mathbf{O}_{3\times3} & \mathbf{I}_3 & \mathbf{O}_{3\times3} \\ \mathbf{O}_{3\times3} & \mathbf{O}_{3\times3} & \mathbf{O}_3 & \mathbf{I}_3 \end{bmatrix}}_{\widehat{\mathbf{B}}(\mathbf{s})} \cdot \mathbf{a} + \mathbf{g}(\mathbf{s}), \quad (71)$$

where $\mathbf{g}(\mathbf{s})$ represents the model mismatch in the updated model. The sampling technique changes the continuous-time dynamics model in Equation (71) into a discrete-time model:

$$\mathbf{s}(k+1) = (\mathbf{I}_{10} + T \cdot \widehat{\mathbf{A}}(\mathbf{s})) \cdot \mathbf{s}(k) + T \cdot \widehat{\mathbf{B}}(\mathbf{s}) \cdot \mathbf{a}(k) + T \cdot \mathbf{g}(\mathbf{s}),$$

from which we obtain the knowledge of $\mathbf{A}(\mathbf{s}(k))$ and $\mathbf{B}(\mathbf{s}(k))$ in Equation (17) as follows:

$$\mathbf{A}(\mathbf{s}(k)) = \mathbf{I}_{10} + T \cdot \widehat{\mathbf{A}}(\mathbf{s}(k)) \ \text{ and } \ \mathbf{B}(\mathbf{s}(k)) = T \cdot \widehat{\mathbf{B}}(\mathbf{s}(k)), \quad (72)$$

where $T = \frac{1}{30}$ second, i.e., the sampling frequency is 30 Hz.

### I.3.2 Parameters for Real-Time Patch Computing

Given that $T = \frac{1}{30}$ seconds, to satisfy Assumption 5.1, we let $\kappa = 0.01$. For other parameters, we let $\chi = 0.3, \alpha = 0.9, \eta = 0.6, \theta = 0.3$, and $\phi = 0.15$. This ensures that the inequalities presented in Equation (22) are satisfied.

### I.4 DRL-Student Design

In this experiment, our DRL-Students build upon two state-of-the-art safe DRLs: CLF-DRL, as proposed in [55], and Phy-DRL outlined in [13, 12]. A critical difference is CLF-DRL has purely data-driven action policy, while Phy-DRL has a residual action policy:

$$\mathbf{a}_{\text{student}}(k) = \underbrace{\mathbf{a}_{\text{drl}}(k)}_{\text{data-driven}} + \underbrace{\mathbf{a}_{\text{phy}}(k) \, (= \mathbf{F} \cdot \mathbf{s}(k))}_{\text{model-based}},$$

where

$$\mathbf{F} = \begin{bmatrix} 0 & 0 & 0 & 0 & -23.65 & 0 & 0 & 0 & 0 & 0 \\ 0 & 0 & 0 & 0 & 0 & -20 & 0 & 0 & 0 & 0 \\ -63.11 & 0 & 0 & 0 & 0 & 0 & -20 & 0 & 0 & 0 \\ 0 & -32.51 & 0 & 0 & 0 & 0 & 0 & -21.88 & 0 & 0 \\ 0 & 0 & -32.51 & 0 & 0 & 0 & 0 & 0 & -21.88 & 0 \\ 0 & 0 & 0 & -30.95 & 0 & 0 & 0 & 0 & 0 & -22.28 \end{bmatrix}. \quad (73)$$

A comparison of these models will be conducted, with all other configurations remaining the same. In all models, the parameter for defining DRL-Student's self-learning space in Equation (4) is the same as $\eta = 0.3$. Both CLF-DRL and Phy-DRL adopt the safety-embedded reward:

$$\mathcal{R}\left(\mathbf{s}(t), \mathbf{a}_{\text{student}}(t)\right) = \mathbf{s}^\top(t) \cdot \overline{\mathbf{P}} \cdot \mathbf{s}(t) - \mathbf{s}^\top(t+1) \cdot \overline{\mathbf{P}} \cdot \mathbf{s}(t+1), \quad (74)$$

where

$$\overline{\mathbf{P}} = \begin{bmatrix} 122.1647861 & 0 & 0 & 0 & 2.487166 & 0 & 0 \\ 0 & 1.5e-5 & 0 & 0 & 0 & 0 & 0 \\ 0 & 0 & 1.5e-5 & 0 & 0 & 0 & 0 \\ 0 & 0 & 0 & 480.6210753 & 0 & 0 & 0 \\ 2.487166 & 0 & 0 & 0 & 3.2176033 & 0 & 0 \\ 0 & 0 & 0 & 0 & 0 & 1.3e-6 & 0 \\ 0 & 0 & 0 & 0 & 0 & 0 & 1.2e-6 \\ 0 & 9e-5 & 0 & 0 & 0 & 0 & 0 \\ 0 & 0 & 9e-5 & 0 & -0 & 0 & 0 \\ 0 & 0 & 0 & 155.2954559 & 0 & 0 & \end{bmatrix}$$

$$\begin{bmatrix} 0 & 0 & 0 \\ 9e-5 & 0 & 0 \\ 0 & 9e-5 & 0 \\ 0 & 0 & 155.2954559 \\ 0 & 0 & 0 \\ 0 & 0 & 0 \\ 0 & 0 & 0 \\ 7e-5 & 0 & 0 \\ 0 & 7e-5 & 0 \\ 0 & 0 & -0, 156.3068079 \end{bmatrix}. \quad (75)$$

# J   Experiment Design: Quadruped Robot: Wild Environment

We utilize NVIDIA Isaac Gym [39] and the Unitree Go2 robot to evaluate our Real-DRL approach. In Isaac Gym, we create dynamic wild environments that include various natural elements, such as unstructured terrain, movable stones, and obstacles like trees and large rocks. These environments are further complicated by unexpected patches of ice that result in low friction. We can conclude that the Go2 robot operates in non-stationary and unpredictable conditions.

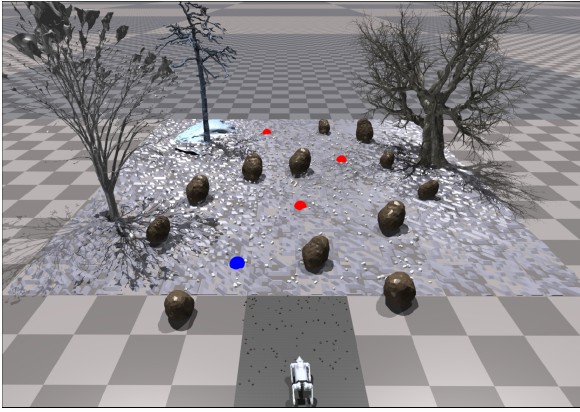

Additionally, we arrange multiple waypoints at reasonable intervals across the terrain to simulate real-world tasks for the robot, such as outdoor exploration and search-and-rescue operations. The goal is to guide the robot to its destination by sequentially following these waypoints while minimizing traversal costs and adhering to safety constraints. The established wild environments are detailed in Figure 8.

Figure 8: An overview of wild environments. The quadruped robot navigates to its destination by following waypoints with minimal costs through runtime learning. The next waypoint is highlighted in blue, while the remaining waypoints are marked in red.

Our Real-DRL must integrate perception and planning to complete tasks safely in real-time wild environments. The integrated Real-DRL framework is illustrated in Figure 9, where the perception and planning components form the navigation module.

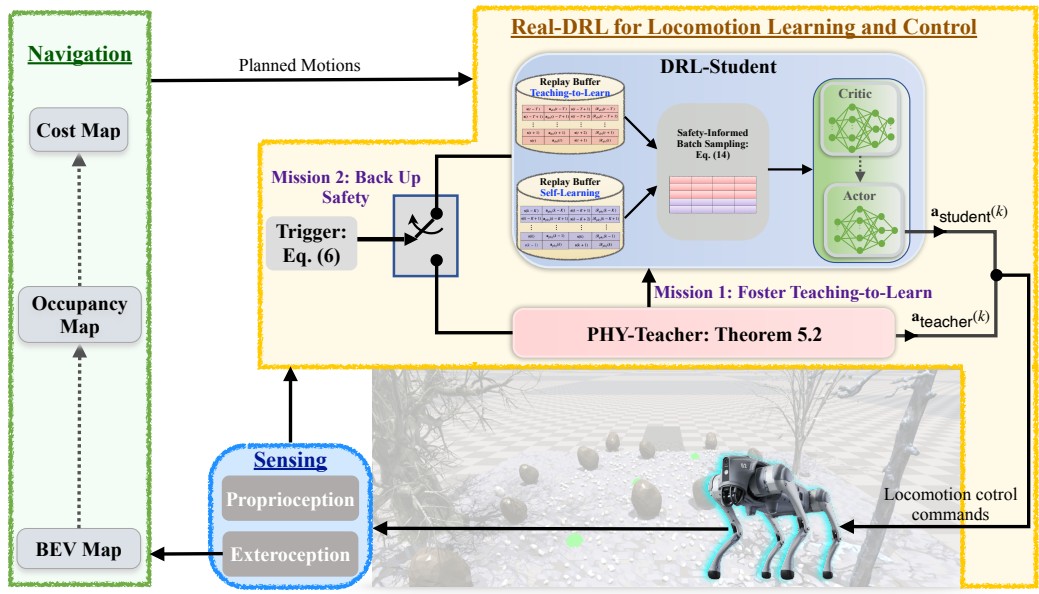

Figure 9: **Integration of Real-DRL with Navigation**: A seamless integration of perception, planning, learning, and control is essential for robotic operation in complex environments. The **sensing** module collects both proprioceptive and exteroceptive data from the robot as it navigates challenging surroundings. The **navigation** module constructs a Bird's Eye View (BEV) map by filtering Regions of Interest, generates an occupancy map using height thresholding, and computes a cost map with the Fast Marching method. Using this cost map, a planner generates planned motions for the locomotion controller. In the **locomotion learning and control** module, the trigger facilitates interaction between the PHY-Teacher and the DRL-Student to ensure the robot's safety. This setup allows the DRL-Student to learn from the PHY-Teacher, while also engaging in self-learning to develop a safe and high-performance locomotion policy.

### J.1 Learning Configurations and Computation Resources

The observation space is defined as $\mathbf{o} = [\mathbf{z}, \mathbf{s}]^\top$, where $\mathbf{z}$ represents the exteroceptive observation and $\mathbf{s}$ indicates the proprioceptive tracking-error vector. A waypoint is considered successfully reached when the Euclidean distance between the robot's center of mass and the target waypoint is less than 0.3 meters.

The actor and critic networks are implemented as Multi-Layer Perceptrons with four fully connected layers. The output dimensions are structured as follows: the critic network has output sizes of 256, 128, 64, and 1, while the actor network has output sizes of 256, 128, 64, and 6. The input for the critic network consists of both the observation vector and the action vector, while the actor network takes only the observation vector as input. The activation functions used in the first three layers of both networks are ReLU, while the output layer of the actor network employs the Tanh function, and the critic network uses a linear activation function. Additionally, the discount factor $\gamma$ is set to 0.9, and both the critic and actor networks have learning rates of 0.0003. Finally, the batch size is configured to be 512.

For the computational resources, we utilized a desktop running Ubuntu 22.04, equipped with a 13th Gen Intel® Core™ i9-13900K 16-core processor, 64 GB of RAM, and an NVIDIA GeForce GTX 4090 GPU. The algorithm was implemented in Python using the PyTorch framework, and the MATLAB LMI toolbox was employed for solving real-time patches.

### J.2 Safety Conditions

The Real-DRL is designed to track the planned movements generated by the navigation module while ensuring that the system's states remain within a predefined safety set. This is crucial for avoiding falls and collisions with obstacles. The safety set is defined as follows: $\mathbb{S} = \{\mathbf{s} \in \mathbb{R}^{10} \mid |\text{CoM x-velocity} - r_{v_x}| \leq 1.5 \text{ m/s}, |\text{CoM z-height} - r_z| \leq 0.6 \text{ m}, |\text{raw} - r_{\text{raw}}| \leq 0.8 \text{ rad}, |\text{pitch} - r_{\text{pitch}}| \leq 1 \text{ rad}\}$. The action space is defined as: $\mathbb{A} = \{\mathbf{a} \in \mathbb{R}^6 \mid |\mathbf{a}| \leq [25, 25, 25, 50, 50, 50]^\top\}$. These formulations ensure that the system operates within safe parameters while effectively following the intended navigation paths.

### J.3 PHY-Teacher Design

The physics model knowledge $(\mathbf{A}(\mathbf{s}(k)), \mathbf{B}(\mathbf{s}(k))$ used for the design of PHY-Teacher is described in Equation (72). Given sampling period $T = \frac{1}{30}$ second, we set $\kappa = 0.0008$ to satisfy the conditions outlined in Assumption 5.1. Additionally, the following parameters are defined: $\chi = 0.15$, $\alpha = 0.9$, $\eta = 0.6$, $\theta = 0.3$, and $\phi = 0.2$, ensuring that the inequalities in Equation (22) are upheld.

### J.4 DRL-Student Design

In this experiment, our DRL-Students build upon two state-of-the-art safe DRLs: CLF-DRL, as proposed in [55], and Phy-DRL, outlined in [13, 12]. A critical difference is CLF-DRL has purely data-driven action policy, while Phy-DRL has a residual action policy:

$$\mathbf{a}_{\text{student}}(k) = \underbrace{\mathbf{a}_{\text{drl}}(k)}_{\text{data-driven}} + \underbrace{\mathbf{a}_{\text{phy}}(k) \, (= \mathbf{F} \cdot \mathbf{s}(k))}_{\text{model-based}},$$

where $\mathbf{F}$ is given in Equation (73). A comparison of these models will be conducted, with all other configurations remaining the same, as detailed below.

#### J.4.1 Reward

The reward for the DRL-Student primarily focuses on three key aspects: robot safety, travel time, and energy efficiency.

■ **Safety**. We consider a Control-Lyapunov-like reward in [55], which incorporates both safety and stability considerations:

$$\mathcal{R}_{\text{safety}} = \mathbf{e}^\top(t) \cdot \overline{\mathbf{P}} \cdot \mathbf{e}(t) - \mathbf{e}^\top(t+1) \cdot \overline{\mathbf{P}} \cdot \mathbf{e}(t+1), \tag{76}$$

where $\overline{\mathbf{P}}$ is given in Equation (75).

■ **Travel Cost**. The Euclidean distance between the robot and the waypoint is defined as:

$$d(t) = \|\hat{\mathbf{x}}_b(t) - \hat{\mathbf{p}}_{wp}\|_2 \tag{77}$$

where $\hat{\mathbf{p}}_{wp}$ is the location of next waypoint, and $\hat{\mathbf{x}}_b$ is the position of the robot base in the world frame. Inspired by [25] and [41], the navigation reward is defined as:

$$\mathcal{R}_{\text{navigation}} = c_1 \cdot \hat{r}_{dis}(t) + c_2 \cdot \hat{r}_{wp} + \hat{r}_{obs} \tag{78}$$

where $\hat{r}_{dis}(t) = d(t) - d(t-1)$ is the reward for forwarding the waypoint. $\hat{r}_{wp} = e^{-\lambda \cdot T_{reach}}$ rewards the robot as it reaches the waypoint. $\lambda \in (0,1)$ is a time decay factor and $T_{reach}$ denotes the time step when the waypoint is reached. The $\hat{r}_{obs}$ serves as a penalty when the quadruped collides with the obstacles. $c_1$ and $c_2$ are used hyperparamters. This reward structure incentivizes the robot to achieve an efficient and collision-free navigation through the waypoints.

■ **Energy Consumption**. Power efficiency remains a major challenge for robots in outdoor settings. We model the motor as a non-regenerative braking system [58]:

$$p_{motor} = \max\{ \underbrace{\tau_m \cdot \omega_m}_{output\ power} + \underbrace{L_{copper} \cdot \tau_m^2}_{heat\ dissipation} , 0\} \tag{79}$$

where $\tau_m$ and $\omega_m$ are motor's torque and angular velocity respectively. $L_{copper}$ is copper loss coefficient. Taking $c_m$ as a hyperparameter, we define the energy consumption reward:

$$\mathcal{R}_{\text{energy}} = -c_m \cdot p_{motor} \tag{80}$$

The ultimate reward function, designed to guide the DRL-Student in learning a safe and high-performance policy as defined in Equation (9), is given by integrating the components from above Equations (76), (78) and (80), i.e., $\mathcal{R}(\mathbf{s}(t), \mathbf{a}_{\text{student}}(t)) = \mathcal{R}_{\text{safety}} + \mathcal{R}_{\text{navigation}} + \mathcal{R}_{\text{energy}} + \hat{c} \cdot \mathcal{R}_{\text{auxiliary}}$, where $\mathcal{R}_{\text{auxiliary}}$ denotes the auxiliary reward for tracking velocity and orientation, with a small coefficient $\hat{c}$ to promote smooth locomotion. In experiment, we choose $c_1 = 5, c_2 = 10, c_m = 0.02$ and $\hat{c} = 0.5$.

### J.4.2   Self-Learning Space

The parameter used to define the self-learning space for DRL-Student, as mentioned in Equation (4), is set at $\eta = 0.6$.

### J.4.3   Safety-Informed Batch Sampling

For the safety-informed batch sampling in Equation (14), we let $\rho_1 = 0.01$ and $\rho_2 = 0.05$

### J.5   Ablation Experiment

We recall that the model knowledge of PHY-Teacher, along with its resulting action policy and control goal, is dynamic and responsive to complex and dynamic operating environments in real-time. In contrast, the physics or dynamics models used in fault-tolerant DRLs, such as neural Simplex [43], runtime assurance [8, 50, 16], and model predictive shielding [4, 3], are static. This difference suggests that current fault-tolerant DRLs struggle to guarantee safety in complex and highly dynamic environments. To illustrate this point, we compare the PHY-Teachers with dynamic (real-time) and static models in an unstructured terrain filled with random stones. The comparison video can be found at https://www.youtube.com/shorts/Ct_xXlohSiM. This video demonstrates that when using the static model, the PHY-Teacher cannot guarantee safety in the environment. In contrast, with the real-time dynamic model, our PHY-Teacher can not only guarantee safety but also withstand unexpected sudden kicks.

# K   Experiment Design: Cart-Pole System

We use the cart-pole simulator available in OpenAI Gym [9] to emulate the real system, allowing the deployed Real-DRL to learn from scratch. The unknown unknowns introduced in this scenario consist of disturbances that affect the DRL-Student's action commands and the friction force. These disturbances are generated using a randomized Beta distribution. In Appendix H, we explain why the randomized Beta distribution serves as a form of unknown unknown.

## K.1   Learning Configurations and Computation Resources

The actor and critic networks are designed as Multi-Layer Perceptrons consisting of four fully connected layers. The output dimensions for the critic and actor networks are 256, 128, 64, and 1, respectively. The activation functions for the first three layers are ReLU, while the final layer of the actor network uses the Tanh function and the critic network employs a linear activation function. The input to the critic network is $[\mathbf{s}; \mathbf{a}]$, which combines the state and action, while the actor network takes only the state $\mathbf{s}$ as its input. We have set the discount factor $\gamma$ to 0.9, and both the critic and actor networks have the same learning rate of 0.0003. The batch size is $L = 512$, and each episode consists of 1,000 steps, with a sampling frequency of 50 Hz.

For the computational resources, we utilized a desktop equipped with Ubuntu 22.04, a 12th Gen Intel® Core™ i9-12900K 16-core processor, 64 GB of RAM, and an NVIDIA GeForce GTX 3090 GPU. The algorithm was implemented in Python using the TensorFlow framework, and we employed the Matlab LMI toolbox for solving real-time patches.

## K.2   Safety Conditions

The objective of Real-DRL is to stabilize the system at the equilibrium point $\mathbf{s}^* = [0, 0, 0, 0]^\top$ while ensuring that the system states remain within the safety set $\mathbb{S} = \left\{ \mathbf{s} \in \mathbb{R}^4 \mid |x| \leq 1, \ |\theta| < 1.0 \right\}$. The action space is defined as $\mathbb{A} = \left\{ \mathbf{a} \in \mathbb{R} \mid |\mathbf{a}| \leq 50 \right\}$.

## K.3   PHY-Teacher Design

The physics-model knowledge about the dynamics of cart-pole systems for the design of PHY-Teacher is from the following dynamics model presented in [21]:

$$\ddot{\theta} = \frac{g \sin\theta + \cos\theta \left( \frac{-F - m_p l \dot{\theta}^2 \sin\theta}{m_c + m_p} \right)}{l \left( \frac{4}{3} - \frac{m_p \cos^2\theta}{m_c + m_p} \right)}, \tag{81a}$$

$$\ddot{x} = \frac{F + m_p l \left( \dot{\theta}^2 \sin\theta - \ddot{\theta} \cos\theta \right)}{m_c + m_p}, \tag{81b}$$

whose physical representations of the model parameters can be found in following Table 3.

Table 3: Notation table

| | Notation | Value | Unit |
|---|---|---|---|
| $m_c$ | mass of cart | 0.94 | $kg$ |
| $m_p$ | mass of pole | 0.23 | $kg$ |
| $g$ | gravitational acceleration | 9.8 | $m \cdot s^{-2}$ |
| $l$ | half length of pole | 0.32 | $m$ |
| $F$ | actuator input | [-50, 50] | $N$ |
| $x$ | position of cart | [-1, 1] | $m$ |
| $\dot{x}$ | velocity of cart | [-3, 3] | $m \cdot s^{-1}$ |
| $\theta$ | angle of pole | [-1, 1] | $rad$ |
| $\dot{\theta}$ | angular velocity of pole | [-4.5, 4.5] | $rad \cdot s^{-1}$ |

### K.3.1 Physics-Model Knowledge

The PHY-Teacher converts the dynamics model (81) into a state-space model as follows:

$$\frac{d}{dt}\underbrace{\begin{bmatrix} x \\ \dot{x} \\ \theta \\ \dot{\theta} \end{bmatrix}}_{\mathbf{s}} = \underbrace{\begin{bmatrix} 0 & 1 & 0 & 0 \\ 0 & 0 & \frac{-m_p g \sin\theta\cos\theta}{\theta[\frac{4}{3}(m_c+m_p)-m_p\cos^2\theta]} & \frac{\frac{4}{3}m_p l \sin\theta\dot{\theta}}{\frac{4}{3}(m_c+m_p)-m_p\cos^2\theta} \\ 0 & 0 & 0 & 1 \\ 0 & 0 & \frac{g\sin\theta(m_c+m_p)}{l\theta[\frac{4}{3}(m_c+m_p)-m_p\cos^2\theta]} & \frac{-m_p\sin\theta\cos\theta\dot{\theta}}{\frac{4}{3}(m_c+m_p)-m_p\cos^2\theta} \end{bmatrix}}_{\widehat{\mathbf{A}}(\mathbf{s})} \cdot \begin{bmatrix} x \\ \dot{x} \\ \theta \\ \dot{\theta} \end{bmatrix}$$

$$+ \underbrace{\begin{bmatrix} 0 \\ \frac{\frac{4}{3}}{\frac{4}{3}(m_c+m_p)-m_p\cos^2\theta} \\ 0 \\ \frac{-\cos\theta}{l[\frac{4}{3}(m_c+m_p)-m_p\cos^2\theta]} \end{bmatrix}}_{\widehat{\mathbf{B}}(\mathbf{s})} \cdot \underbrace{F}_{\mathbf{a}}, \quad (82)$$

where $\widehat{\mathbf{A}}(\mathbf{s})$ and $\widehat{\mathbf{B}}(\mathbf{s})$ are known to the PHY-Teacher. The sampling technique transforms the continuous-time dynamics model (82) to the discrete-time one:

$$\mathbf{s}(k+1) = (\mathbf{I}_4 + T \cdot \widehat{\mathbf{A}}(\mathbf{s})) \cdot \mathbf{s}(k) + T \cdot \widehat{\mathbf{B}}(\mathbf{s}) \cdot \mathbf{a}(k),$$

from which we obtain the model knowledge $\mathbf{A}(\mathbf{s}(k))$ and $\mathbf{B}(\mathbf{s}(k))$ in Equation (17) as

$$\mathbf{A}(\mathbf{s}(k)) = \mathbf{I}_4 + T \cdot \widehat{\mathbf{A}}(\mathbf{s}(k)), \qquad \mathbf{B}(\mathbf{s}(k)) = T \cdot \widehat{\mathbf{B}}(\mathbf{s}(k)), \quad (83)$$

where $T = \frac{1}{50}$ second, i.e., the sampling frequency is 50 Hz.

### K.3.2 Parameters for Real-Time Patch Computing

Given that $T = \frac{1}{50}$ second, to satisfy Assumption 5.1, we let $\kappa = 0.0008$. Additionally, the parameters are defined as follows: $\chi = 0.3, \alpha = 0.9, \eta = 0.7, \theta = 0.5$, and $\phi = 0.01$. These values are chosen to ensure that the inequalities stated in Equation (22) hold true.

### K.4 DRL-Student Design

In the ablation studies, the DRL-Student builds on the CLF-DRL proposed in [55]. Its CLF reward in this experiment is

$$\mathcal{R}(\mathbf{s}(k), \mathbf{a}_{\text{student}}(k)) = (\mathbf{s}^\top(k) \cdot \overline{\mathbf{P}} \cdot \mathbf{s}(k) - \mathbf{s}^\top(k+1) \cdot \overline{\mathbf{P}} \cdot \mathbf{s}(k+1)) \cdot \gamma_1 - \mathbf{a}_{\text{student}}^2(k) \cdot \gamma_2,$$

where we let $\gamma_1 = 1, \gamma_2 = 0.015$, and

$$\overline{\mathbf{P}} = \begin{bmatrix} 54.1134178606985 & 26.2600592637275 & 61.7975412804215 & 12.9959418258126 \\ 26.2600592637275 & 14.3613985149923 & 34.6710819094179 & 7.27321583818861 \\ 61.7975412804215 & 34.6710819094179 & 88.7394386456256 & 18.0856894519164 \\ 12.9959418258126 & 7.27321583818861 & 18.0856894519164 & 3.83961074325448 \end{bmatrix}.$$

The parameter for defining DRL-Student's self-learning space in Equation (4) is $\eta = 0.7$.

The ablation studies involve experimental comparisons of our Real-DRL models with and without the teaching-to-learn mechanism. When the teaching-to-learn mechanism is used, the DRL-Student utilizes both a self-learning replay buffer and a teaching-to-learn replay buffer, with each buffer configured to hold a maximum of 100,000 instances. Conversely, when the teaching-to-learn mechanism is not employed, the DRL-Student relies on a single replay buffer to store self-learning experience data. To ensure a fair comparison, the size of this single buffer is set to match the combined capacity of the two buffers, totaling 200,000.

### K.5 Auxiliary Experiment

This section provides additional experimental results that complement the main ablation study findings.

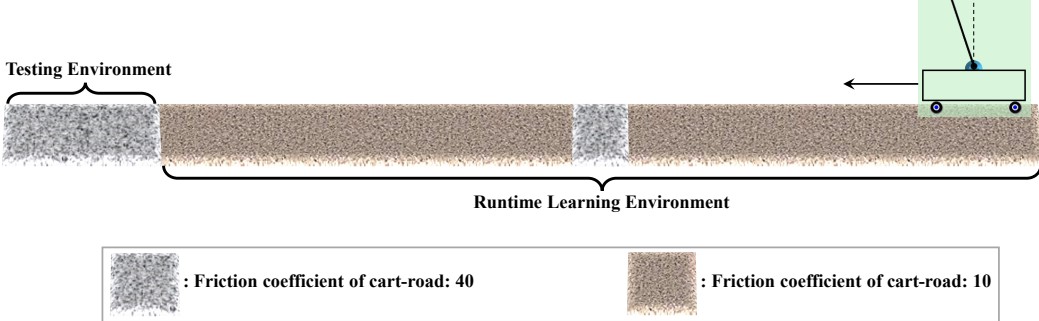

Figure 10: Learning and testing environments. The learning environment has a corner case (pink area), which is also the testing environments.

### K.5.1 Safety-Informed Batch Sampling

For the safety-informed batch sampling in Equation (14), we let $\rho_1 = 10$ and $\rho_2 = 0.1$. To demonstrate the proposed safety-informed batch sampling method for addressing experience imbalance, we create a learning environment characterized by this imbalance, as shown in Figure 10. Specifically, the brown area closely resembles the pre-training environment of the DRL-Student, representing the majority of the continual learning environment for Real-DRL, accounting for 90% of the cases. In contrast, the gray area indicates a corner case that significantly differs from the DRL-Student's pre-training conditions, comprising only 10% of the total runtime learning cases. This corner-case environment also serves as the testing environment for the DRL-Student after it has completed continual learning.

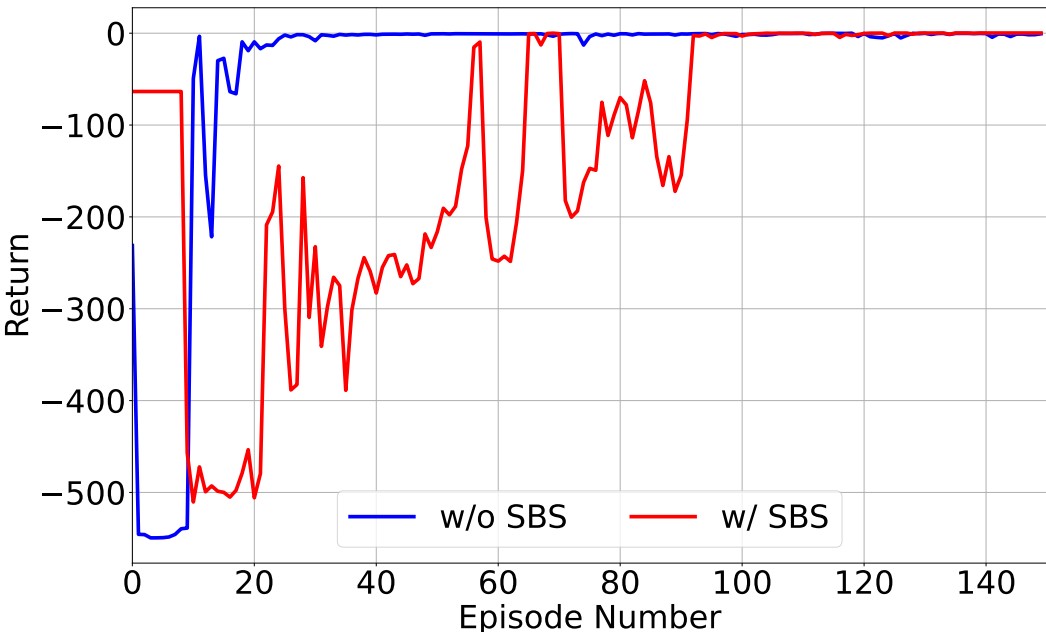

Figure 11: Return trajectories of DRL-Student.

Figure 11 shows the return trajectories of the DRL-Student, which were learned using the Real-DRL framework. It compares the results obtained with our proposed safety-informed batch sampling method (referred to as 'w/ SBS') to those obtained without this method (denoted as 'w/o SBS'), which utilizes only a single united reply buffer. Figure 11 clearly shows that the tested DRL-Students, which were obtained by disabling PHY-Teacher, are selected from both Real-DRL models once their learning processes have reached convergence. This can ensure fair comparisons.

### K.5.2 Teaching-to-Learn Paradigm

To evaluate the effect of the teaching-to-learn paradigm on policy learning, we introduce the metric of DRL-Student's episode-average reward, defined below.

$$\text{Episode-Average Reward} = \frac{\text{Return (i.e., cumulative reward) in one episode}}{\text{DRL-Student's total activation time in one episode}}. \tag{84}$$

### K.5.3 Automatic Hierarchy Learning

To further demonstrate PHY-Teacher's contributions to the claimed automatic hierarchy learning mechanism, we define the metric of PHY-Teacher's activation ratio as follows.

$$\text{PHY-Teacher's activation ratio} = \frac{\text{PHY-Teacher's total activation times in one episode}}{\text{one episode length}}, \tag{85}$$

which has a value range of $[0, 1]$. A ratio of 0 means PHY-Teacher is never activated throughout the entire episode of learning, while a ratio of 1 means PHY-Teacher completely dominates DRL-Student for the entire episode.

Given five random initial conditions, the phase plots, each running for 1000 steps, are displayed in Figure Figure 12. Observing Figure 12, we conclude that the DRL-Student has successfully learned from the DRL-Teacher how to ensure safety in episode 5: its action policy effectively confines the system states to the safety set (indicated by the red area). In episode 50, the PHY-Teacher is rarely activated by the condition in Equation (6), as illustrated in Figure Figure 6 (b). Meanwhile, the action policy of the DRL-Student demonstrates improved mission performance, achieving faster clustering and a much closer proximity to the mission goal, which is located at the center of the safety set.

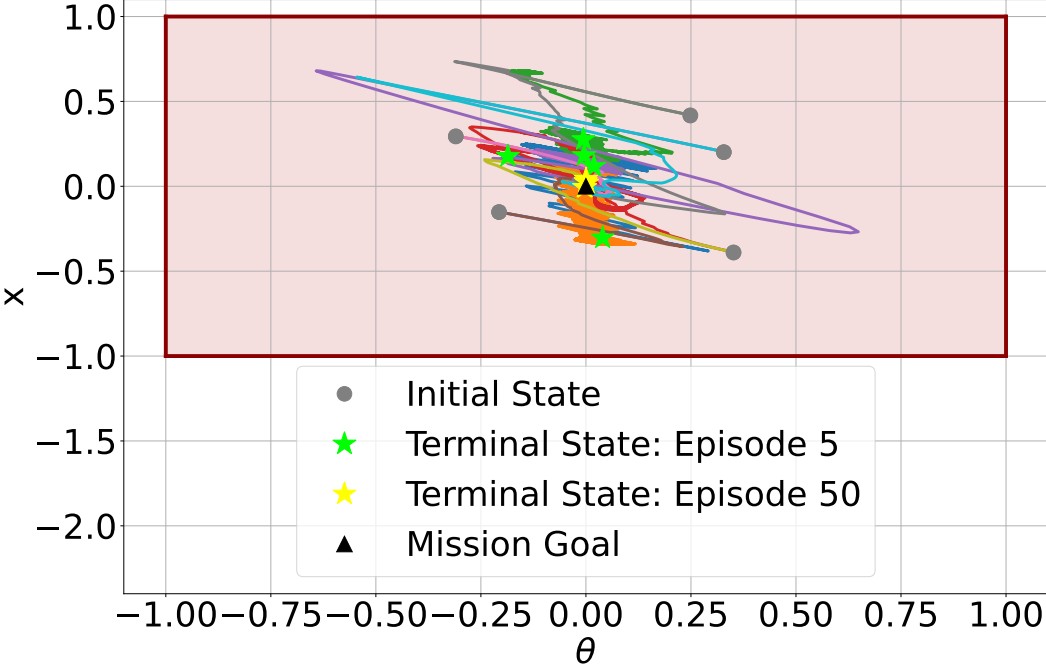

Figure 12: Demonstrating automatic hierarchy learning by phase plots, running over five random initial states within the safety set.

