# OpenReview forum: "Real-DRL: Teach and Learn at Runtime"
_NeurIPS.cc/2025/Conference — NeurIPS 2025 poster_

### Official Review · Reviewer_zVT6 · 2025-07-01

**Clarity:** 3
**Significance:** 2
**Originality:** 3
**Rating:** 4
**Confidence:** 4

**Summary:**

The Real-DRL framework enables safe deep reinforcement learning in physical autonomous systems by integrating three components: a DRL-Student that learns action policies through self-learning (for performance) and teaching-to-learn (safety lessons from interventions); a PHY-Teacher using physics models to provide real-time safety patches for unknown unknowns and Sim2Real gaps; and a Trigger that monitors state boundaries to switch control. It features automatic safety-first hierarchy learning and safety-status-dependent batch sampling to address corner-case imbalances. Validated on a real quadruped robot, NVIDIA Isaac Gym wild environments, and cart-pole, Real-DRL guarantees runtime safety while outperforming baselines.

**Questions:**

1. How long does it take to collect sufficient data for training in the real world? In Section 6.1, fixed vel commands are used, which is a simpler setup compared to the quadruped robot algorithms widely adopted nowadays. If real-world training is conducted using continuous commands, similar to state-of-the-art quadruped robot algorithms, would the time cost incurred be acceptable? This is the point I am most concerned.
2. How are the marginal safety boundaries determined? How does the setting of marginal safety boundaries affect the results of the algorithm?
3. Can these marginal safety boundaries be gradually widened as the capability of the DRL-Student improves?

**Ethical Concerns:**

["NO or VERY MINOR ethics concerns only"]

**Final Justification:**

The algorithm proposed in this paper can serve as an "Evolving Brain" for real-world control policy. The experiments in the paper are solid, and the innovations are novel.

The only drawback, in my opinion, is that the experiments on real-world deployment are rather simplistic. Therefore, I maintain my original score of 4.

**Limitations:**

yes

**Paper Formatting Concerns:**

No paper formatting concerns.

**Quality:**

3

**Strengths And Weaknesses:**

Strengths
1. Formal Safety Guarantees: Addresses unknown unknowns and Sim2Real gaps through PHY-Teacher's physics-based interventions, providing verifiable safety.
2. Runtime Adaptability: Teaching-to-learn enables the DRL-Student to acquire safety knowledge from real-time failures. Safety-status-dependent sampling mitigates data imbalance in corner cases.

Weekness
1. Even model-based approaches cannot fully guarantee safety, especially when encountering unknown unknowns.
2. In the field of quadruped robots, existing sim2real training frameworks can already achieve highly robust real-world performance. Yet the authors did not adopt state-of-the-art quadruped robot training strategies.
3. Training a sufficiently good RL strategy requires a huge amount of data, and real-world data collection is much more time-consuming compared to data collection in simulations, even with the addition of safety guarantees.

---

> ### Author Rebuttal · Authors · 2025-07-29
>
> ---
>
> **Weakness 1: Even model-based approaches cannot fully guarantee safety, especially when encountering unknown unknowns.**
>
> **Response:** No technique, including ours, can guarantee safety against `arbitrarily large` unknown unknowns, such as unpredictable engine failures. This acknowledgment is formally reflected in  Assumption 5.1 in the paper, which imposes an upper bound on model mismatch.  The model mismatch can arise from various sources, including unknown uncertainties, unpredictable platform faults, unmodeled dynamics, and external disturbances not captured in the nominal model. PHY-Teacher's theoretical safety guarantee (formally stated in Theorem 5.2) through tolerating unknown unknowns is valid under Assumption 5.1. When this assumption is violated, PHY-Teacher can still maintain safety, but cannot provide a theoretical guarantee (i.e., Theorem 5.2 does not hold).
>
> In reality, particularly when facing large, unpredictable, real-time unknown unknowns, it is unrealistic to expect Assumption 5.1 to hold consistently. To address this challenge, we introduce a real-time adaptive mechanism (detailed in Appendix F) that employs the real-time state sampling to estimate real-time model mismatches to enable continuous monitoring of Assumption 5.1's validity. When violations are detected, the mechanism computes compensatory actions in real time to mitigate model mismatch effects and restore the assumption's validity.
>
> ---
>
> **Weakness 2: In the field of quadruped robots, existing sim2real training frameworks can already achieve highly robust real-world performance. Yet the authors did not adopt state-of-the-art quadruped robot training strategies.**
>
> **Response:** Many state-of-the-art frameworks leverage GPU-accelerated simulators (e.g., Nvidia Isaac Gym) to train robust locomotion policies through large-scale parallel sampling. They often require millions to billions of training steps per instance to enable the deployment of sufficiently robust policies directly onto real hardware. However, such policies are typically static once deployed, lacking the flexibility to adapt or continue learning in real-world environments. This limitation reduces their biological plausibility and adaptability, which are critical for achieving general-purpose intelligence in the physical world.
>
> Moreover, we provide the motivation and vision of our research to address the reviewers' comment.
>
> * **Motivation: Dynamics and Environments: Complexity and Unpredictability.** Deep reinforcement learning (DRL) possesses vast parameters, intractable activation functions, and stochastic learning dynamics. Meanwhile, many safety-critical autonomous systems (e.g., legged robots, drones, autonomous vehicles, and UAVs) have complex dynamics-environment interactions. Their operating environments (e.g., wildfires, freezing rain, snow squalls) are often unpredictable and hard to simulate accurately. Consequently, the dynamics of DRL-enabled autonomous systems exhibit inherent complexity and unpredictability, creating substantial Sim2Real and domain gaps that threaten system safety. While numerous approaches have been developed to address these gaps, these methods can only mitigate the gaps to varying degrees. Undisclosed gaps continue to hinder the safety assurance of real systems, arising from the fundamentally complex and unpredictable nature of real-world dynamics and operating environments.
>
> * **Vision: "Evolving Brain" Architecture.**  To thoroughly address Sim2Real and domain gaps caused by complex and unpredictable dynamics and operating environments, learning-enabled autonomous systems should possess an "evolving brain"—similar to human cognitive adaptation—that enables continuous learning in the real physical world, using real-time sensor data generated from real-time physical operating environments. This approach allows action policies to evolve dynamically in real-time, achieving adaptive responses to complex and unpredictable operating conditions. Our proposed Real-DRL framework represents an essential milestone toward delivering this ``evolving brain" capability, enabling autonomous systems to maintain safety while continuously learning to adapt to novel environmental challenges.
>
> ---
>
> **Weakness 3: Training a sufficiently good RL strategy requires a huge amount of data, and real-world data collection is much more time-consuming compared to data collection in simulations, even with the addition of safety guarantees.**
>
> **Response:**  We would like to clarify that our primary goal in conducting the experiments and designing our Real-DRL is not to replace a high-fidelity simulator or to advocate for starting real-world learning from scratch. Instead, our Real-DRL aims to enable the system to continue learning after deployment, while assuring safety through addressing Sim2Real gaps and unknown unknowns that inevitably arise in the complex real world.
>
> Our framework positions the simulator as a crucial component for pre-training suboptimal policies, providing reasonable starting points. However, a simulator cannot capture all real-world complexities—including unmodeled dynamics, sensor noise, and unexpected disturbances. Our Real-DRL addresses this gap by enabling safe and real-time learning in real physical environments.
>
> The significance of hardware implementation demonstrates a key advantage over conventional Sim2Real pipelines: while traditional approaches halt learning after deployment, our framework maintains the robot's plasticity, allowing continuous improvement through real-world experience within certified safety bounds. This capability is essential for long-term autonomy and adaptability in open-ended and non-stationary environments.
>
>
>
> ---
>
> **Question 1: How long does it take to collect sufficient data for training in the real world? In Section 6.1, fixed vel commands are used, which is a simpler setup compared to the quadruped robot algorithms widely adopted nowadays. If real-world training is conducted using continuous commands, similar to state-of-the-art quadruped robot algorithms, would the time cost incurred be acceptable? This is the point I am most concerned.**
>
> **Response:** The duration and cost of real-world training are influenced by the Sim2Real gap and the specific task configuration.
>
> * **Learning in the Real World: Safety First:** The time cost incurred during real-world learning is not our primary concern, as our application systems are `safety-critical`, where learning must adhere to the safety-first principle. Our Real-DRL prioritizes and assures safety, meaning that even without learning, Real-DRL can still succesfully perform the safety-critical functions, but with `low` task performance. The learning objective is to achieve high task performance, such as minimizing travel time, average power consumption, and total energy expenditure. While high task performance is desired, it is secondary to assuring safety.
>
> * **Static Command:**  In Section 6.1 of the paper, we deliberately employ a fixed velocity command as a controlled and simplified test case to validate our framework's real-world learning capability. The policy is pre-trained in a simulator, then deployed and continuously refined in the real world on the real robot at a different speed. Notably, the robot successfully learned a safe locomotion policy `< 20 episodes`, with each episode lasting approximately 15 seconds, resulting in `< 5 minutes` of total physical interaction time.
>
> * **Dynamic Commands:** Section 6.2 utilizes NVIDIA Isaac Gym to create a wild environment, `assumed` to be the real one. The architecture of Real-DRL, integrated with a navigation module, is illustrated in `Figure 10` of the paper. The navigation module outputs dynamic planned motions for our Real-DRL. We compared Real-DRL with safe DRLs such as CLF-DRL and Phy-DRL, and fault-tolerant DRLs like neural Simplex and runtime assurance. The trajectories of episode returns in `Figure 4`  demonstrate that Real-DRL exhibits significantly faster and more efficient learning in identical environments.
>
> ---
>
> **Question 2: How are the marginal safety boundaries determined? How does the setting of marginal safety boundaries affect the results of the algorithm?**
>
> **Response:**  The marginal-safety boundaries are the ones of the self-learning space $\mathbb{L}$. So, the reviewer's first question transforms to: how is the $\mathbb{L}$ determined? The $\mathbb{L}$ is to intentionally create a buffer zone $\mathbb{S} \setminus \mathbb{L}$. This configuration provides adequate margins for system response time and decision-making latency. Therefore, the $\mathbb{L}$ in our work is determined offline through analyzing a worst-case dynamics model, obtained by considering the worst-case model mismatch, CPU computation time, sensor sampling period, and other relevant system constraints.
>
> For the second question, the extensive safety boundaries would result in a minimal buffer zone, which would reduce decision-making time and space, making it difficult for the PHY-Teacher to implement a feasible backup action policy for safety. Conversely, smaller safety boundaries do not pose safety concerns because they yield a larger buffer zone, but they result in a reduced self-learning space. This limitation makes it slow for DRL-Student to develop a robust and high-performance action policy.
>
> ---
>
> **Question 3: Can these marginal safety boundaries be gradually widened as the capability of the DRL-Student improves?**
>
>
> **Response:** Yes. DRL-Student engages in a dual teaching-to-learn and self-learning paradigm. So, its safe learning space is $\mathbb{L}  \bigcup \mathbb{P}_{k}, k \in \mathbb{N}$, where $\mathbb{L}$ represents the self-learning space, the other is the teaching-to-learn space, and $k$ represents the activation time of the teacher. As the learning deepens, the total activation number of the teacher increases; consequently, the safe learning space expands.
>
> ---

---

> > ### Comment · Reviewer_zVT6 · 2025-08-02
> > **Official Comment by Reviewer zVT6**
> >
> > Thank you for the response.
> >
> > The rebuttal has addressed my concerns. I also strongly agree with the "Evolving Brain" Architecture. Carrying out incremental learning in real-world scenarios is indeed an approach to tackle the "Complexity and Unpredictability" of  "Dynamics and Environments".  In the future, I'd glad to see more demos that can better demonstrate the algorithm's performance based on this work, especially in scenarios that can break through the upper limits of traditional RL's sim2real methods.

---

> ### Author Response · Authors · 2025-08-02
>
> We thank the reviewer for the positive feedback. Our current real-world experimental setting in Section 6.1 has demonstrated Real-DRL's capabilities in addressing the Sim2Real gap, as well as tolerating complex external disturbances. Enabling Real-DRL to operate on a complete edge-AI device, such as an NVIDIA Jetson, will realize the "evolving brain" concept as well as the edge learning. We would appreciate it if the reviewer could adjust the rating accordingly if our rebuttal has addressed the concerns.

---

### Official Review · Reviewer_BbAN · 2025-07-02

**Clarity:** 3
**Significance:** 3
**Originality:** 3
**Rating:** 4
**Confidence:** 3

**Summary:**

This work introduces Real-DRL, a hybrid runtime learning architecture for safety-critical systems composed of a DRL-Student (learning high-performance policies), a PHY-Teacher (guaranteeing safety via physics-based backup control), and a Trigger that switches control based on safety conditions. The goal is to deal with the unknown unknowns and the sim2real gap.

The manuscript seems to be a mix of describing the proposal in the context of a controlling a plant, and some examples in other domains, which is confusing. If the focus is on safe controlling of plants, examples in this domain should be provided. Otherwise, remove the excessive "plant" indication.

**Questions:**

- Why does the DRL-Student adopt an actor-critic architecture? Are other architectures possible?
- Why mentioning so many times "plants" but provide no example of real application to a plant?
- Cannot the proposal be applied in general and remove the mention to plants in places where it is not necessary?
- How robust is the PHY-Teacher under model mismatch or poorly characterized environments? This is address in Annex F, but would be good to mention it in the main text.
- Could the PHY-Teacher be replaced by a simpler rule-system? Or is the "physic-model" a generalization or any system that can provide control actions?
- Is the PHY-Teacher ever trained? If so, how? If not, does it need to be formally described for each problem?

**Ethical Concerns:**

["NO or VERY MINOR ethics concerns only"]

**Final Justification:**

It’s clear that this is a solid piece of work, and I appreciate the effort the authors have put into identifying and addressing potential drawbacks in the annexes and in the rebuttals.

The only drawback that I can currently see in the proposal is that lack of generality as the PHY-Teacher needs to be defined/tweaked for each domain (cannot be done automatically). Said that, if you are focused in just one domain, you define it once and forget about it.

In any case, I am increasing my score due to the thorough responses to me and my fellow reviewers.

**Limitations:**

The manuscirpt does not highlight any limitation of the proposal, but I have listed several ones (or at least, questions for which I cannot find a proper answer) in my previous comments.

**Paper Formatting Concerns:**

no concerns

**Quality:**

3

**Strengths And Weaknesses:**

STRENGHTS:
- Verifiable safety of DRL systems is a must these days, and this is a work in that direction.
- The proposal is formally described.
- Experiments are solid and they are very well described in the Annex.


WEAKNESSES:
- The manuscripts mentions, since the very beginning, "plants" (which?). But given examples have nothing to do with them. Why such a focus on plants? Cannot the proposal be described in general without that focus?
- The proposal has a heavy reliance on PHY-Teacher performing ok at all times. This might not always be possible to guarantee in real scenarios.
- The novelty, besides the concrete implementation details, is limited compared to existing Fault-tolerant DRL that uses HPM and HAM.

---

> ### Author Rebuttal · Authors · 2025-07-29
>
> ---
>
> **Weakness 1: The manuscripts mentions, since the very beginning, "plants" (which?). But given examples have nothing to do with them. Why such a focus on plants? Cannot the proposal be described in general without that focus?**
>
> **Response:** In the communities of control, robotics, and cyber-physical systems, the term "real plant" (often referred to as "plant") refers to the actual physical system, process, or device that is being controlled or manipulated. The "real plant" can be anything from a DC motor, robotic arm, or chemical process to a vehicl. It is a generic term that signifies "the thing to be controlled." So, in our real-world experiments, the real quadruped robot serves as the "real plant." In our mimicked real-world experiments, the quadruped robot from Nvidia Isaac Gym and the cart-pole system from OpenAI Gym represent the "real plants."
>
> To address the reviewer's concern and avoid potential confusion, we have replaced the "real plant" with "real system" in the revised paper.
>
> ---
>
> **Weakness 2: The proposal has a heavy reliance on PHY-Teacher performing ok at all times. This might not always be possible to guarantee in real scenarios.**
>
> **Response:** Our Real-DRL framework does not always have heavy reliance on PHY-Teacher. We provide the following summaries of the learning journal and teaching-to-learn process of the DRL-Student to clarify this point.
>
> * **Learning journal of DRL-Student.**  PHY-Teacher is activated only when DRL-Student behaves unsafely, leading to a safety violation of the system. When the system returns to normal, PHY-Teacher returns the control role to DRL-Student. The Trigger in our Real-DRL framework orchestrates their interactions, assuring system safety. Under the management of Trigger, the DRL-Student's learning journey involves acquiring knowledge from the PHY-Teacher (i.e., teaching-to-learn), who possesses verifiable domain expertise in safety protocols. As learning deepens, the DRL-Student gradually becomes `independent` of the PHY-Teacher, ultimately achieving `self-mastery` upon acquiring safe and high-performance action policies.
>
> * **Continuous teaching-to-learn, independent of PHY-Teacher activation!** The DRL-Student continuously acquires safety knowledge from PHY-Teacher through our proposed `dual teaching-to-learn and self-learning replay buffers + safety-status-dependent batch sampling`. This design ensures  DRL-Student has continuous access to PHY-Teacher's safety experiences stored in the teaching-to-learn replay buffer, allowing the DRL-Student to refine its policy while maintaining safety guarantees. The teaching-to-learn mechanism thus operates independently of PHY-Teacher activation status, providing persistent safety learning.
>
> ---
>
> **Weakness 3: The novelty, besides the concrete implementation details, is limited compared to existing Fault-tolerant DRL that uses HPM and HAM.**
>
> **Response:** Our research aims to advance verifiably safe Deep Reinforcement Learning (DRL) for `safety-critical` autonomous systems, including legged robots, drones, UAVs, autonomous vehicles, and aircraft. This topic encompasses two primary directions: 1) safe DRL and 2) fault-tolerant DRL. Our paper provides a comprehensive comparison of these approaches, examining their motivations, underlying assumptions, design principles, and experimental implementations.
> Section 1.2 summarizes the fundamental differences in motivations and assumptions between these approaches. For implementation and experimentation, Sections 6.1 and 6.2 present comparisons with recently published state-of-the-art safe DRL frameworks, including Phy-DRL (which claims both theoretical and experimental fast training towards safety guarantee) and CLF-DRL (a seminal CLF-like reward function towards verifiable stability and safety). Additionally, Section 6.2 compares the notable fault-tolerant DRL frameworks, including neural Simplex and runtime assurance frameworks.
>
> ---
>
> **Question 1: Why does the DRL-Student adopt an actor-critic architecture? Are other architectures possible?**
>
> **Response:** There are several key motivations behind our adoption of the actor-critic architecture.
>
> First, the actor-critic architecture provides natural support for continuous action spaces, which is essential for the safety-critical autonomous systems we consider, such as legged robots, drones, UAVs, autonomous vehicles, and many others. This capability is crucial given the continuous control requirements inherent in these applications.
>
> Second, one of our core contributions—the conjunctive safety-status-dependent batch sampling mechanism and dual replay buffers—integrates seamlessly with the actor-critic framework's modular structure. This integration significantly enhances learning efficiency in task performance and safety assurance.
>
> Third, the actor-critic architecture represents one of the most widely adopted paradigms in deep reinforcement learning. Its balanced approach to policy optimization and value estimation forms the foundation for numerous state-of-the-art algorithms, including both on-policy methods (e.g., PPO, TRPO) and off-policy methods (e.g., SAC, DDPG, TD3).
>
> While we focus on actor-critic architectures in the paper, we note that the proposed mechanism is not architecturally constrained and could, in principle, be adapted to other deep reinforcement learning frameworks that employ similar modularity and replay-based training paradigms.
>
>  ---
>
> **Question 2: Why mentioning so many times "plants" but provide no example of real application to a plant?**
>
> **Response:** In the communities of control, robotics, cyber-physical systems, and IoT, the "real plant" is a standard terminology signifying "the system to be controlled." In our real-world experiments, the real quadruped robot serves as the real plant. In our simulated experiments, the quadruped robot from Nvidia Isaac Gym and the cart-pole system from OpenAI Gym represent the real plants within their respective environments.
>
> ---
>
> **Question 3: Cannot the proposal be applied in general and remove the mention to plants in places where it is not necessary?**
>
> **Response:** We appreciate the reviewer's suggestion. To address the reviewer's concern and avoid potential confusion, we have replaced the terminology "real plant" with "real system" in the revised paper.
>
> ---
>
> **Question 4: How robust is the PHY-Teacher under model mismatch or poorly characterized environments? This is address in Annex F, but would be good to mention it in the main text.**
>
> **Response:** PHY-Teacher is a physics-model-based design that focuses exclusively on safety-critical functions. Its theoretical safety guarantee is valid under Assumption 5.1, which provides an upper bound on model mismatch. The model mismatch arises from unknown uncertainties, unmodeled dynamics, and external disturbances that are not accounted for in the nominal model used during the design process.  In real-world environments, particularly in complex, dynamic, and unpredictable situations (such as wildfires, freezing rain, and snow squalls), we cannot always guarantee that Assumption 5.1 will hold. To address this challenge, we propose a real-time adaptive mechanism that employs real-time state sampling to estimate the current model mismatch, thereby monitoring whether Assumption 5.1 is satisfied. If the assumption is violated, compensatory actions are computed in real-time to counteract the model mismatch and restore the validity of Assumption 5.1.
>
> Logically and at a high level, the goal of the proposed adaptive mechanism is only to ensure that Assumption 5.1 remains valid by compensating for model mismatches in real-time, thereby ensuring that Theorem 5.2 holds.  Also, due to the nine-page limit of the submitted paper, we have moved the details of this proposed mechanism to Appendix F. The accepted paper is permitted to include one additional content page for the camera-ready version. If our paper is accepted, we will transfer this content from Appendix F to the main text.
>
> ---
>
> **Question 5: Could the PHY-Teacher be replaced by a simpler rule-system? Or is the "physic-model" a generalization or any system that can provide control actions?**
>
> **Response:**  Safety-critical autonomous systems typically operate within continuous action and state spaces. Consequently, the actions of our PHY-Teacher are continuous. If a rule-based system can deliver verifiable safety, it could potentially serve as a replacement for the PHY-Teacher. To our knowledge, no research efforts have explored this direction.
>
> The "physics model" in our paper refers to general mathematical models derived from fundamental physics laws (e.g., Newton's laws and conservation of energy) that describe system dynamics. Therefore, our PHY-Teacher design framework can be generalized to all physical engineering systems that have available physics models describing their dynamics.  This generalizability is demonstrated in our experiments, where we evaluated our framework on two distinctly different systems: a quadruped robot and a cart-pole system.
>
> ---
>
> **Question 6: Is the PHY-Teacher ever trained? If so, how? If not, does it need to be formally described for each problem?**
>
> **Response:** PHY-Teacher is a physics-model-based design with assured safety, focusing exclusively on safety-critical functions. Unlike data-driven learning approaches, it does not require training. PHY-Teacher satisfies our three key requirements: (i) assured safety with formal theoretical guarantee, (ii) real-time action policies that can respond to complex, dynamic, and unpredictable operating environments, and (iii) rapid and automatic computation of action policies in real time. All of these expected capabilities are formally described in Section 5. Additionally, Appendix E provides pseudocode for rapid and automatic computation of PHY-Teacher's real-time action policies via different toolboxes: Python CVXPY and Matlab LMI.
>
> ---

---

> > ### Comment · Reviewer_BbAN · 2025-08-03
> >
> > Thank you for the additional information provided in the revision.
> > It’s clear that this is a solid piece of work, and I appreciate the effort the authors have put into identifying and addressing potential drawbacks, both in the annexes and in the rebuttal.
> >
> > I still feel that my last question was not adequately answered. Based on the response, I understand that a new PHY-Teacher needs to be formally defined for each new domain/problem.
> >
> > In any case, and considering your responses to both my comments and those of the other reviewers, I am increasing my score.
> >
> > PS: Apologies for my lack of knowledge on “plants”, and thank you for your explanation.

---

> ### Author Response · Authors · 2025-08-03
>
> Dear Reviewer,
>
> We appreciate your positive feedback, recognition of our contributions, and valuable comments.
>
> Your initial understanding of "real plant" serves as a valuable reminder for us. We must ensure that our terminology accommodates the diverse backgrounds of our readers across various research domains. Therefore, in our revised paper, we have replaced "real plant" with "real system."
>
> Finally, we would like to provide additional information regarding your last question (Perhaps, we also misunderstood the question in the rebuttal).
>
> * **Generalizability:** Our proposed PHY-Teacher framework can be applied to diverse safety-critical autonomous systems. Different autonomous systems typically have different safety problems. For example, the navigation of legged robots in the wild focuses on balance maintenance and collision avoidance with obstacles such as trees and large stones, while the self-driving vehicles prioritize safe adherence to traffic regulations. Application of the PHY-Teacher to these different safety problems only requires the formal description of these safety constraints or regulations in the mathematical format of the conditions for defining the safe set in Eq. (2) of the paper.
>
> * **Physical AI:** The PHY-Teacher within our Real-DRL is a physics-model-based design (no training needed) that focuses exclusively on safety-critical functions. We note that in today's world, there are two primary approaches to achieve the same control tasks of physical engineering systems: a `high-dimensional, data-driven ML` approach (e.g., DRL and DNN), and a `low-dimensional, physics-model-based` approach (e.g., adaptive control and CBF). The ML approach often delivers superior task performance but lacks analyzable and verifiable behavior. In contrast, the physics-model-based approach enables analyzable and verifiable stability and safety, but has limited task performance due to model mismatch. This dichotomy motivates our Real-DRL framework: integrating these two approaches (i.e., `the PHY-Teacher and DRL-Student`) to overcome their respective limitations while preserving their advantages. We can also conclude that our work belongs to the emerging topic of Physical AI (physics for AI, not AI for Physics).
>
> * **Domain Knowledge for PHY-Teacher Design:** The design of PHY-Teacher relies on "physics models," which refer to general mathematical models derived from fundamental physics laws (e.g., Newton's laws and conservation of energy) that describe system dynamics. According to our knowledge, the required domain knowledge in the physics models of physical engineering systems is readily available for use, as system dynamics has been well-studied for decades. Such physical engineering systems include quadruped robots [1], drones [2], autonomous vehicles [3], UAVs [4], humanoid robots [5], power systems [6], and many others.
>
> Best regards,
>
> Authors
>
> ---
>
> **References:**
>
> [1] Di Carlo, J., Wensing, P. M., Katz, B., Bledt, G., & Kim, S. (2018, October). Dynamic locomotion in the MIT Cheetah 3 through convex model-predictive control. In 2018 IEEE/RSJ international conference on intelligent robots and systems.
>
> [2] Beard, R. W. (2008). Quadrotor dynamics and control. Brigham Young University, 19(3), 46-56.
>
> [3] Rajamani, R. (2011). Vehicle dynamics and control. Springer Science & Business Media.
>
> [4] Goodarzi, F. A., & Lee, T. Dynamics and control of quadrotor UAVs transporting a rigid body connected via flexible cables. In the 2015 American Control Conference.
>
> [5] Nenchev, D. N., Konno, A., & Tsujita, T. (2018). Humanoid robots: Modeling and control. Butterworth-Heinemann.
>
> [6] Machowski, J., Lubosny, Z., Bialek, J. W., & Bumby, J. R. (2020). Power system dynamics: stability and control. John Wiley & Sons.

---

### Official Review · Reviewer_gyUQ · 2025-07-03

**Clarity:** 3
**Significance:** 3
**Originality:** 3
**Rating:** 5
**Confidence:** 4

**Summary:**

This paper proposes Real-DRL, a framework enabling runtime learning of a deep reinforcement learning (DRL) agent with safety assurance and high-performance learning, which improves on existing DRL frameworks' abilities to provide safety guarantees especially in the face of corner cases and the sim2real gap. The framework has 3 components: DRL-Student, PHY-Teacher, and Trigger. DRL-Student is the agent being trained and only acts within a predefined subset of the "safe" state space. PHY-Teacher is responsible for taking control of the actions when DRL-Student navigates too close to the "unsafe" space and provides samples of its own experience returning the plant back to the "safe" region of the state space to DRL-Student to learn from. Trigger is the monitoring component that determines when to switch between DRL-Student and PHY-Teacher.

**Questions:**

Why does the framework perform so poorly without the safety-status-dependent batch sampling method? I would expect safety performance to be worse without it, but every trajectory in the phase plot appears to enter the unsafe region. Are none of these samples from PHY-teacher being used at all when evaluating without the safety-status-dependent batch sampling method?

**Ethical Concerns:**

["NO or VERY MINOR ethics concerns only"]

**Final Justification:**

Based on the response and discussion, assuming the authors clarify what safety means in this context and given how the discussion has focused much on safety/verification, I am in favor of acceptance, along with the condition the authors release their code that will help further clarify what safety means in this context.

**Limitations:**

Yes

**Quality:**

3

**Strengths And Weaknesses:**

Strengths:
- The general problem studied is significant and interesting, with overall a rigorous development of the results presented for the developed framework
- Most writing of the paper is clear and they provide ample intuition for how their framework works (albeit there are some caveats noted below).
- Experiments are interesting and broad, and the results appear to beat SOTA in some cases.

Weaknesses:
- Some equations are missing references to the variables in them and therefore are not clear or intuitive at all (e.g., the Real-Time Patch in equation 15 or equation 21 and 22).

- The main motivation for combining learning and control is to avoid the strong assumption of fully known Markovian dynamics. In general, I am not fully convinced learning-based control techniques while still assuming the dynamics are known is reasonable. Assuming a known model significantly limits the novelty and practical impact of the contribution.

- Unlike standard reinforcement learning algorithms that do not assume model access, the authors rely on having access to the model to design PHY-Teacher. This makes comparisons with other techniques challenging.

- Regarding the definition of the trigger’s role: How can you guarantee that the student policy will be able to bring the system back into the learning-enabled region when its actions are restricted to the set A? This is a critical challenge that must be addressed. Encountering unseen data may lead to large errors that push the system far outside the learning-enabled region, making recovery infeasible or requiring prohibitively large control inputs. This issue is especially likely if the system dynamics are highly nonlinear.

- The tracking error dynamics in Equation 17 are assumed to be linear, which is a very strong and potentially unrealistic assumption.

- While the authors present strong and compelling plans for addressing unknown unknowns, they do not provide equally convincing arguments that their controllers—PHY-Teacher and DRL-Student—will reliably behave as intended in practice. Although the experimental results are promising, they depend on having access to the model during training.

- "reply" should be "replay" on line 213 (pg. 6).

---

> ### Author Rebuttal · Authors · 2025-07-29
>
> ---
> **Weakness 1: Some equations are missing references to the variables in them and therefore are not clear or intuitive at all (e.g., the Real-Time Patch in equation 15 or equation 21 and 22).**
>
> **Response:** We thank the reviewer for bringing this issue to our attention. The references to the variables in Equations (15), (19), and (20) are summarized in Lines 254-256 and Remark 5.3 of the paper. Due to the 9-page limit, more detailed descriptions of these variables, along with their physical representations, are provided in Appendix E. Since the accepted paper allows for one additional content page in the camera-ready version, we will place the variables' physical interpretations directly after Equations (15), (19), and (20), if our paper is accepted.
>
> ---
>
> **Weakness 2: The main motivation for combining learning and control is to avoid the strong assumption of fully known Markovian dynamics. In general, I am not fully convinced learning-based control techniques while still assuming the dynamics are known is reasonable. Assuming a known model significantly limits the novelty and practical impact of the contribution.**
>
> **Response:**  The aims of this paper extend beyond merely eliminating the assumption of fully known Markovian dynamics.
> * **Motivation:** In today's world, there are two approaches to achieve the same control tasks: a high-dimensional ML approach (e.g., DRL), and a low-dimensional, physics-model-based approach (e.g., adaptive control).  The ML approach often delivers superior task performance but lacks analyzable and verifiable behavior. In contrast, the physics-model-based approach enables analyzable and verifiable stability and safety, but has limited task performance due to model mismatch.  This dichotomy motivates our Real-DRL: integrating these two approaches to overcome their respective limitations while preserving their advantages.
>
> * **Novelty and Practicality:** The physics model in our paper refers to a general mathematical model derived from physics laws (e.g., Newton's laws and the conservation of energy) for describing the dynamics of a system. Using this physics model does not detract from the novelty of the proposed Real-DRL; rather, it enhances Real-DRL by featuring: i) verifiable safety, ii) automatic hierarchy learning, and iii) safety-status-dependent batch sampling to address the experience imbalance. Moreover, this requirement does not limit the practical impact of Real-DRL, because the physics models for safety-critical autonomous systems are readily accessible.
>
> ---
>
> **Weakness 3: Unlike standard reinforcement learning algorithms that do not assume model access, the authors rely on having access to the model to design PHY-Teacher. This makes comparisons with other techniques challenging.**
>
> **Response:** At a high level, our proposed Real-DRL belongs to the emerging topic of Physical AI, infusing domain knowledge in physics into data-driven ML. As a result, our RL is inherently different from standard RL frameworks. Comparing our Real-DRL with other techniques is straightforward (not challenging). This is because recent efforts have been devoted to physical RL, e.g.,  Phy-DRL, neural Simplex, and runtime assurance. This paper includes such comparisons.
>
> ---
>
> **Weakness 4: Regarding the definition of the trigger’s role: How can you guarantee that the student policy will be able to bring the system back into the learning-enabled region when its actions are restricted to the set A? This is a critical challenge that must be addressed. Encountering unseen data may lead to large errors that push the system far outside the learning-enabled region, making recovery infeasible or requiring prohibitively large control inputs. This issue is especially likely if the system dynamics are highly nonlinear.**
>
> **Response:** Our design and theoretical foundation have taken the issue of large unforeseen errors or faults into account.
>
> * **Theoretical Guarantee and Adaptive Mechanism:** PHY-Teacher is designed to have verifiable expertise in safety protocol, as formally stated in **Theorem 5.2**. When the system controlled by the DRL-Student approaches a marginal-safety boundary $\partial\mathbb{L}$, the Trigger activates the PHY-Teacher to take over the DRL-Student to steer the system back to $\mathbb{L}$.  This safety protocol has a theoretical safety guarantee and is valid under Assumption 5.1, which establishes an upper bound on model mismatch. The model mismatch therein can arise from unknown uncertainties, errors, unmodeled dynamics, and disturbances not captured in the nominal physics model. In practice, especially when confronted with unpredictable significant errors, faults, or disturbances, it is unrealistic to expect Assumption 5.1 always to hold. To address this challenge, the paper introduces an adaptive mechanism (detailed in Appendix F) that utilizes real-time state samplings to estimate the real-time model mismatches, enabling the monitoring of the validity of Assumption 5.1. When the assumption is violated, the mechanism computes compensatory actions in real-time to mitigate the effects of model mismatch and restore the validity of the assumption.
>
> * **Buffer Zone:** In our Real-DRL, the Trigger monitors the system's real-time safety status to manage interactions between DRL-Student and PHY-Teacher. However, the safety set $\mathbb{S}$ cannot be directly utilized to indicate the safety status. This is because when the system approaches its boundaries, the PHY-Teacher cannot instantaneously steer the system back into the safety set due to inherent system delays, especially in the face of large errors, faults, or disturbances.  These delays are caused by factors such as communication latency, sampling periods, actuator response times, etc. To address this issue, we introduce the self-learning space $\mathbb{L}$ that ensures $\mathbb{L} \subset \mathbb{S}$, creating a buffer zone: $\mathbb{S} \setminus \mathbb{L}$. Meanwhile, the Trigger refers to the boundary $\partial\mathbb{L}$ to proactively manage interactions between the DRL-Student and PHY-Teacher. The configuration provides adequate margins for system response time and decision-making latency.
>
>
>
> ---
>
> **Weakness 5: The tracking error dynamics in Equation 17 are assumed to be linear, which is a very strong and potentially unrealistic assumption.**
>
> **Response:** Perhaps the formula format in Eq. (19) impressed the reviewer that it is linear, which is not true. Demonstration examples of highly nonlinear dynamics are presented in Appendix H.3 and Appendix J.3 of the paper.
>
> ---
>
> **Weakness 6: While the authors present strong and compelling plans for addressing unknown unknowns, they do not provide equally convincing arguments that their controllers—PHY-Teacher and DRL-Student—will reliably behave as intended in practice. Although the experimental results are promising, they depend on having access to the model during training.**
>
> **Response:**  Our `Response to Weakness 2` explains that the dependence on physics models enhances rather than limits its novelty and practical impact. Our `Response to Weakness 4` explains that our Real-DRL, integrating PHY-Teacher and DRL-Student, explicitly addresses the safety challenge due to unknown unknowns, from theory to real-world implementation.
>
> ---
>
> **Weakness 7: "reply" should be "replay" on line 213 (pg. 6).**
>
> **Response:**  We have corrected it.
>
> ---
>
> **Questions: "Why does the framework perform so poorly without the safety-status-dependent batch sampling method? I would expect safety performance to be worse without it, but every trajectory in the phase plot appears to enter the unsafe region. Are none of these samples from PHY-teacher being used at all when evaluating without the safety-status-dependent batch sampling method?**
>
> **Response:**  In the paper,  Figure 6 includes 10 trajectories: 5 for learning with the safety-status-dependent batch sampling scheme (denoted as `w/ SBS` ) and 5 for learning without this scheme (denoted as `w/o SBS`). Perhaps due to the color scheme, the phase plots impressed the reviewer with a contradictory conclusion. Actually, the results meet our expectations entirely: the trajectories denoted by `w/o SBS-1 - w/o SBS-5` all enter the unsafe region. In contrast, the remaining `w/ SBS-1 - w/ SBS-5` represent ours, and none of them enter the unsafe area.
>
> The integrated architecture `Dual Replay Buffers + Safety-Status-Dependent Batch Sampling` achieves enhanced learning performance—its absence results in significantly degraded performance. In addition to facilitating a dual teaching-to-learn and self-learning paradigm, the architecture also significantly addresses uneven experience distribution (due to corner but safety-critical cases) in learning.
>
> * **Dual Replay Buffers:** `Figure 1` in the paper shows that teaching-to-learn experience data is acquired in the marginally safe space, capturing corner-case experiences. However, existing DRLs typically employ a single replay buffer and random batch sampling. As a result, corner-case data is often omitted entirely, due to the lower probability of selection. Our dual replay buffers address this limitation by storing the corner-case and normal data separately and specifically. This design ensures that batch samples consistently incorporate corner-case data, thereby maintaining their representation during learning.
>
> * **Safety-Status-Dependent Batch Sampling:**  In the paper, `Figure 2`, `Theorem 4.1`, and `Eq. (14)` reveal that the contribution of corner-case experience data increases proportionally with the real-time safety status. For instance, the $V(s(t)) = 1$ indicates that the real-time states are approaching safety boundaries. Consequently, all batch samples are drawn from the marginally safe space (of the corner cases) stored in the teaching-to-learn replay buffer. This focuses the learning on corner-case experiences, precisely when the real system is marginally safe.
>
> ---

---

> > ### Comment · Reviewer_gyUQ · 2025-08-06
> >
> > Thank you for the response and clarifications. I have gone over the other reviews and discussions. Overall while I was generally positive already, based on the clarifications and pointers, I am in favor of acceptance. I would suggest the following further updates in a final version, if accepted, which seem to not fully be addressed in the paper, appendices, or discussion so far.
> >
> > 1: The authors indicate they will release the code/models, this is crucial if the paper is accepted.
> > 2: The definition of safety, which appears to a degree, must be clarified, especially given much of the discussion has focused around verification and safety, and the presentation of Table 1 is difficult to fully understand what safety means (as has arisen in some confusion from reviewers).
> >
> > For example, collisions and fall are to a degree defined in I.2. However, the fall aspect in particular is problematic and does not appear fully defined, and collision is not clearly defined at all (why is it in R^10? dimensionality of the constraints is not clear as there are many fewer terms, perhaps they are all eg in R^2 and about 5 components make it R^10). The authors must clarify what safety means in this work and clearly present it mathematically, given the importance of this angle of the paper. Finally, this highlights a limitation, in that what is presented in the paper for safety (and action constraints), all appear linear or as intervals, so the authors may wish to discuss this and clarify if it can work in more general settings. The other (eg Lyapunov) aspects (I.4.1) make it appear this may in the end necessitate perhaps ellipsoidal constraints due to what looks like quadratic terms, which have problems in higher dimensions in actual constraints. There is prior work in the literature on Simplex architecture addressing this (eg, in the context of quadratic Lyapunov functions) and reducing such conservatism due to ellipsoidal representations in high dimensions.
> >
> > For other examples, there are some constraints (eg H.2 and J.2), but given this, I would strongly encourage checking all this and making sure everything is precise, especially to make the paper more self-contained. As these are complex, perhaps pointers to where these are defined in the codebase will make things clearer, once that is available.

---

> ### Author Response · Authors · 2025-08-07
>
> Dear Reviewer,
>
> Thank you sincerely for your positive feedback and valuable suggestions. If accepted, we will revise the final paper accordingly. Below, we provide additional responses, and the necessary information will be included in the final paper.
>
> ---
>
> **Code Release**: All experimental code and models are now included in the supplementary materials. Upon acceptance, we will make the following enhancements to the repository: standardize naming conventions, add comprehensive inline documentation, and provide a detailed README.md with setup instructions and dependency requirements. The complete codebase will be made publicly available through GitHub.
>
> **Safety Definition**: Appendix I.2 provides an example safety problem where we define the safety set using empirically determined thresholds tailored to the system's physical characteristics. For a quadruped robot operating on uneven terrain, safe operation requires the center of mass (CoM) height to remain between 0.18 m and 0.4 m, while body roll and pitch angles must stay below 0.8 rad and 1.0 rad (approximately ±60°), respectively. Violations of these constraints constitute a fall event. In the demonstration video, the robot flips upside down after stepping on a movable stone, resulting in a CoM height below 0.15 m and roll and pitch angles exceeding 0.8 rad, as measured by the onboard IMU. These physically meaningful and safety-critical state variables are explicitly encoded into the model, yielding a 10-dimensional safety state representation.
>
> Beyond the experimental examples, certain safety constraints originate from regulatory requirements, such as statutorily mandated speed limits for autonomous vehicles in designated school zones.
>
>
> **Generalizability: Linear Inequalities**: The conditions for defining the safety set are linear inequalities of system states. These linear inequalities are sufficiently generic to describe many safety conditions. Examples include a quadruped robot's safe navigation and autonomous vehicles' lane keeping, lane tracking, velocity regulation, collision avoidance, and many others.
>
> In reality, we also have safety conditions expressed as nonlinear inequalities, but they can be equivalently transformed into (or approximated by) linear inequalities. A typical example is autonomous vehicles's safe control of the (nonlinear) slip ratio $: = \frac{\omega \cdot r - v}{v}$, where $\omega$ is the angular velocity of the wheel, $r$ is the effective radius of the tire in free-rolling condition, and $v$ is the forward velocity of the vehicle. To avoid slipping and sliding, we need to keep the slip ratio within a small value $\delta > 0$. This safety requirement can be expressed with the nonlinear inequality: $| \frac{\omega \cdot r - v}{v} | \leq \delta$. Given that $v > 0$ and $\delta > 0$, we can transform this nonlinear inequality into the equivalent form: $\left| \omega \cdot r - v \right| \leq \delta \cdot v$. It can be further equivalently transformed to $(1 - \delta) \cdot v \leq \omega \cdot r \leq (1 + \delta) \cdot v$, representing the linear inequalities of system states to formulate safety conditions.
>
> **Safety-Embedded Reward:** The reward described in Eq. (76) of Appendix I.4.1 is referred to as a safety-embedded reward. This type of reward can automatically convert complex, high-dimensional safety conditions into a one-dimensional scalar condition by utilizing a concept from the Simplex design. This contribution belongs to the comparison model called Phy-DRL (published in ICLR 2024).
>
> ---
>
> Best regards,
>
> Authors

---

### Official Review · Reviewer_Pf5q · 2025-07-04

**Clarity:** 3
**Significance:** 2
**Originality:** 3
**Rating:** 5
**Confidence:** 3

**Summary:**

The methodpresented in this paper, Real-DRL, is focused on don safety-critical learning for robotics applicaitons. Themethod is hierarchical with a "teacher" that guides a "student" policy to achieve safety, and a "student" focused on the control task.
A Safety-Status-Dependent Batch Sampling technique  oversamples safety-critical situations during training, such that safety constraints are prioritized before optimizing for performance (by keeping the teacher in a larger set than the student. The method is validated experimentally on quadruped robots, and cart-pole systems with comparison to existing safe RL approaches and shows a superior performance.

**Questions:**

Currently, safety is maintained by addressing nearly unsafe, or safety-critical situation during learning. What would happen when the system encounters a disturbance not included as safety-critical (i.e., unforeseen disturbance)? Can it be detected as potntially unsafe - in this case, can the policy adapt or restrict itself of acting in the presence of that disturbance?

The physics-guided teacher relies on a model of the real system. What happens in the presence of uncertainties, mismatch, or drifts in this model with respect to the real system? How important is the model's fidelity? A study on parameter uncertainties could be useful to clarify this, in particular with respect to safety constraint violations.

**Ethical Concerns:**

["NO or VERY MINOR ethics concerns only"]

**Final Justification:**

The real-time adaptive mechanism in Appendix F addresses the question about unforeseen deisturbances. I have previously misunderstood that there is no safety guarantee, therefore I am increasing the rating to 5.

**Limitations:**

yes

**Quality:**

3

**Strengths And Weaknesses:**

Strengths: The proposed framework is supported by solid experimental validation on simulated and real platforms, also in the presence of disturbances, and achieves superior performance in the demonstrated cases.
The paper is clearly written.
The method provides runtime assurance for safety.

Weaknesses: The implementation complexity as compared to other methods is higher, due to the hierarchical nature of the method. Computational costs during training are not analysed.
The design an appropriate PHY-Teacher for each new task or robot requires domain expertise.
The approach lacks formal mathematical safety guarantees or proofs. This means that for safety critical systems, there is still a probability of violating safety.

---

> ### Author Rebuttal · Authors · 2025-07-29
>
> ---
> **Weaknesses: The implementation complexity as compared to other methods is higher, due to the hierarchical nature of the method. Computational costs during training are not analysed. The design an appropriate PHY-Teacher for each new task or robot requires domain expertise. The approach lacks formal mathematical safety guarantees or proofs. This means that for safety critical systems, there is still a probability of violating safety.**
>
> **Response:** We thank the reviewer for this thoughtful and multifaceted critique. We address each point below.
> * **Implementation Complexity:** The proposed framework introduces a hierarchical architecture that increases implementation complexity compared to end-to-end learning pipelines. This hierarchical design is intentional and offers several benefits. The hierarchical learning architecture enables clear separation between safety-critical control (managed by the PHY-Teacher) and task-specific learning (handled by the DRL-Student), providing enhanced modularity, flexibility, and safety supervision during real-world deployment. In practice, this separation significantly simplifies debugging processes and improves system interpretability—both critical requirements for safety-critical robotic systems, such as legged robots, UAVs, autonomous vehicles, etc.
>
> * **Computational Cost:** The PHY-Teacher and DRL-Student are designed to operate in parallel. The DRL-Student utilizes GPU acceleration and can be deployed on embedded hardware such as the NVIDIA Jetson series, while the PHY-Teacher handles real-time safety-critical tasks efficiently on the CPU. To minimize runtime overhead, we implement the `LMI-based solver (for automatic and rapid computation of PHY-Teacher's real-time action policies)` in C, which significantly improves both memory efficiency and computational speed. This implementation strategy makes the framework well-suited for deployment on edge devices with limited computational resources.
> We provide a summary of computational resource usage across two platforms with varying CPU frequencies in the following table. Empirically, the C-wrapped PHY-Teacher achieves control update rates exceeding 50 Hz on desktop-class CPUs and over 20 Hz on embedded platforms with constrained computational resources. These frequencies are sufficient for high-level policy updates in a time-critical operating environment. Furthermore, the ability to run the PHY-Teacher efficiently across diverse CPU architectures demonstrates the portability and scalability of our design.
> | Platform | CPU Architecture | CPU Core | Memory Size | Runtime Memory Usage: Python | Runtime Memory Usage: C | LMI Solver: Python | LMI Solver: C |
> |---|---|---|---|---|---|---|---|
> | Dell XPS 8960  Desktop  | x86/64 | 32 | 64 GB |  52 MB |  `10 MB` |  44.6 ms | `18.2 ms` |
> | NVIDIA Jetson AGX Orin           | ARM    | 12 | 64 GB | 113 MB | `8.3 MB` |  116.7 ms | `42.4 ms` |
>
>
> * **Domain Expertise and Formal Mathematical Poof:**  The design of PHY-Teacher relies on the "physics model," which refers to a general mathematical model derived from fundamental physics laws (e.g., Newton's laws and conservation of energy) that describe system dynamics. Building on the physics model, the PHY-Teacher is designed to foster the teaching-to-learn mechanism for the DRL-Student and provide safety backup when the DRL-Student behaves unsafely or unexpectedly. To achieve these objectives, we establish three fundamental requirements for the PHY-Teacher: (i) assured safety with a `rigorous theoretical guarantee (mathematical poof is in Appendix D)`, (ii) real-time action policies that can adapt to complex, dynamic, and unpredictable operating environments, and (iii) rapid and automatic computation of action policies in real time. These requirements can only be met when the PHY-Teacher possesses domain expertise in the physics model of system dynamics. According to our knowledge, the physics models of physical engineering systems are well-studied and are readily available. Examples include quadruped robots, drones, on-road and off-road vehicles, UAVs, humanoid robots, power systems, and aircraft.
>
> ---
>
> **Question 1: Currently, safety is maintained by addressing nearly unsafe, or safety-critical situation during learning. What would happen when the system encounters a disturbance not included as safety-critical (i.e., unforeseen disturbance)? Can it be detected as potntially unsafe - in this case, can the policy adapt or restrict itself of acting in the presence of that disturbance?**
>
> **Response:** Our studies have addressed unforeseen disturbances, which belong to the category of unknown unknowns.
> PHY-Teacher is a physics-model-based design with verifiable safety. Its theoretical safety guarantee holds under Assumption 5.1, which establishes an upper bound on model mismatch. The model mismatch arises from unknown uncertainties, unmodeled dynamics, software and hardware faults, unforeseen external disturbances, and other factors that are not accounted for in the nominal model used by PHY-Teacher. This means that our PHY-Teacher can deliver rigorous safety guarantee as long as Assumption 5.1 holds.
>
> In real-world applications, particularly in hard-to-predict and hard-to-simulate environments (e.g., natural disasters and snow squalls), we cannot always guarantee that Assumption 5.1 will hold. To address this challenge, we propose a real-time adaptive mechanism in Appendix F that employs real-time state samplings to estimate the real-time model mismatch, thereby monitoring the validity status of Assumption 5.1. If the assumption is violated, compensatory actions are computed in real-time to counteract the model mismatch and restore the validity of Assumption 5.1.
>
> Within our framework, any disturbances, faults, or unknowns that lead to the detection of approaching the marginal-safety boundaries are deemed a safety violation. This safety violation triggers the PHY-Teacher to take over DRL-Student to steer the system back toward the safe self-learning space.
>
> Our paper validated the robustness of PHY-Teacher through two comprehensive experiments:
>
> * **Simulator:** In IsaacGym, we evaluate the quadruped robot under external disturbances such as random pushes while traversing uneven, unstructured terrain populated with movable stones (see Section I.5).
>
> * **Real World:** In real-world experiments on the real quadruped robot, we introduce various common unforeseen disturbances, including lateral pushes, kicks, random payload drops, and DDoS faults to emulate system or communication delays  (see Section 6.1).
>
> In the experiments, PHY-Teacher successfully maintains system safety, demonstrating its ability to tolerate the environmental uncertainties, model imperfections, unforeseen disturbances, and platform faults.
>
> ---
>
> **Question 2: The physics-guided teacher relies on a model of the real system. What happens in the presence of uncertainties, mismatches, or drifts in this model with respect to the real system? How important is the model's fidelity? A study on parameter uncertainties could be useful to clarify this, in particular with respect to safety constraint violations.**
>
> **Response:** The design of the physics-guided teacher utilizes the physics model:
> $\mathbf{s}(t+1) = \mathbf{A}(\mathbf{s}(t)) \cdot \mathbf{e}(t) + \mathbf{B}(\mathbf{s}(t)) \cdot \mathbf{a}(t) + \mathbf{h}(\mathbf{s}(t))$, where $\{\mathbf{A}(\mathbf{s}), \mathbf{B}(\mathbf{s})\}$ represents the available knowledge from the physics model used by the teacher. Additionally, $\mathbf{h}(\mathbf{s})$ denotes unknown model uncertainties, parameter uncertainties, unmodeled dynamics, and external disturbances that are not captured by the nominal model knowledge. Consequently, the design and its theoretical foundation explicitly account for uncertainties, mismatches, and drifts in the physics model that relate to the real system, summarized below.
>
> * **Theoretical Safety Guarantee:** The teacher's theoretical safety guarantee is valid under Assumption 5.1, which establishes an upper bound on model mismatch: $\mathbf{h}^\top \cdot \mathbf{P} \cdot \mathbf{h} \le \kappa$. This indicates that the physics-guided teacher can deliver rigorous safety assurance as long as Assumption 5.1 is satisfied. However, if Assumption 5.1 does not hold, the teacher can still ensure safety, though it cannot provide a theoretical proof for this. Therefore, a high-fidelity model is always preferred, as it guarantees the validity of Assumption 5.1 and, consequently, the theoretical safety guarantee outlined in Theorem 5.2.
>
> * **Practical Challenge and Approach:**  In reality, the dynamics of learning-enabled autonomous systems are governed by three categories of knowledge: known knowns (e.g., Newtonian physics), known unknowns (e.g., Gaussian noise without identified mean and variance), and unknown unknowns. One significant example of an unknown unknown arises from DNN's colossal parameter space, intractable activation functions, and random factors. Moreover, many autonomous systems, such as quadruped robots and autonomous vehicles, exhibit complex dynamics-environment interactions. Unpredictable and complex environments, such as freezing rain conditions, can introduce new types of unknown unknowns. Given these challenges, it is not practical to expect that a high-fidelity model exists that can guarantee Assumption 5.1 always holds in real-world scenarios. To address this challenge, the paper introduces a real-time adaptive mechanism that utilizes real-time state sampling to estimate model mismatches in real-time. This allows for monitoring the satisfaction of Assumption 5.1. When this assumption is violated, the mechanism calculates compensatory actions in real-time to counteract the model mismatch and restore the validity of Assumption 5.1. Due to the nine-page limit of the submitted paper, the details of this proposed mechanism are provided in Appendix F.
>
> ---

---

> > ### Comment · Reviewer_Pf5q · 2025-08-02
> > **Answer to the authors' rebuttal**
> >
> > Thank youfor adressing my questions. The real-time adaptive mechanism in Appendix F addresses the question about unforeseen deisturbances. I have previously misunderstood that there is no safety guarantee, therefore I am increasing the rating to 5.

---

> > > ### Author Response · Authors · 2025-08-02
> > >
> > > Dear Reviewer,
> > >
> > > Thank you so much. Your positive feedback is really encouraging for us.
> > >
> > > Best regards,
> > >
> > > The Authors

---

### Note · Authors · 2025-08-13

In our final remarks, we would like to express our sincere gratitude to all reviewers for their thorough evaluations. We greatly appreciate their positive feedback and valuable suggestions, as well as the Area Chair's effective management of the review process.

We have carefully addressed each reviewer's comment in our detailed rebuttals. Based on the reviewers' subsequent responses, we believe we have successfully addressed their concerns. If our paper is accepted, we will enhance both the final manuscript and the accompanying open-source package on GitHub according to the reviewers' essential suggestions.

Best regards,

The Authors

---

### Decision · Program_Chairs · 2025-09-17

**Decision:**

Accept (poster)

**Comment:**

This paper introduces Real-DRL, a safe reinforcement learning framework composed of three components: a DRL-Student that learns task policies, a PHY-Teacher that intervenes in safety-critical states, and a Trigger that switches control. Safety-status-dependent batch sampling is used to emphasize critical scenarios. The framework is evaluated in cart-pole, quadruped robots, and both in simulation and on real platforms.

**Strengths:** The approach is well-motivated and addresses an important challenge in safe RL. The teacher-student-trigger design and safety-status sampling are novel and effective. Experiments are thorough, covering both simulation and real robots, and demonstrate improvements over baselines. The rebuttal clarified important points regarding safety, including robustness to unforeseen disturbances and the scope of the formal guarantees, which apply to certain systems.

**Weaknesses:** The method relies on designing a task-specific PHY-Teacher, which reduces generality and requires domain expertise. While some formal guarantees are provided, they depend on the fidelity of the PHY-Teacher’s model relative to the real system. Since the PHY-Teacher itself does not adapt or improve with data, this reliance represents an important limitation. Real-world validation, though present, remains limited compared to state-of-the-art quadruped RL work. Some clarity issues (notation, terminology) were also noted.

**Final Recommendation:** In line with reviewers' consensus, I recommend acceptance (as a poster). The contribution is solid and relevant for the safe RL community.